# A dominant role of transcriptional regulation during the evolution of C4 photosynthesis in *Flaveria* species

Ming-Ju Amy Lyu[1,9] ✉, Huilong Du[2,3,9], Hongyan Yao[4,9], Zhiguo Zhang[5,9], Genyun Chen[1], Yuhui Huang[1,6], Xiaoxiang Ni[1,6], Faming Chen[1,6], Yong-Yao Zhao[1,6], Qiming Tang[1,6], Fenfen Miao[1,6], Yanjie Wang[1,6], Yuhui Zhao[2], Hongwei Lu[2], Lu Fang[2], Qiang Gao[2], Yiying Qi[7], Qing Zhang[7], Jisen Zhang[7], Tao Yang[8], Xuean Cui[5], Chengzhi Liang[2] ✉, Tiegang Lu[5] ✉ & Xin-Guang Zhu[1] ✉

C4 photosynthesis exemplifies convergent evolution of complex traits. Herein, we construct chromosome-scale genome assemblies and perform multi-omics analysis for five *Flaveria* species, which represent evolutionary stages from C3 to C4 photosynthesis. Chromosome-scale genome sequence analyses reveal a gradual increase in genome size during the evolution of C4 photosynthesis attributed to the expansion of transposable elements. Systematic annotation of genes encoding C4 enzymes and transporters identify additional copies of three C4 enzyme genes through retrotranspositions in C4 species. C4 genes exhibit elevated mRNA and protein abundances, reduced protein-to-RNA ratios, and comparable translation efficiencies in C4 species, highlighting a critical role of transcriptional regulation in C4 evolution. Furthermore, we observe an increased abundance of ethylene response factor (ERF) transcription factors and cognate *cis*-regulatory elements associated with C4 genes regulation. Altogether, our study provides valuable genomic resources for the *Flaveria* genus and sheds lights on evolutionary and regulatory mechanisms underlying C4 photosynthesis.

C4 photosynthesis is a complex trait that evolved from ancestral C3 types approximately 35 million years ago[1,2]. Due to its high efficiencies in light, water, and nitrogen use[3,4], C4 photosynthesis has been proposed for integration into C3 crops to enhance crop yield[5-7]. In contrast to C3 photosynthesis, C4 photosynthesis allocates more enzymes to carbon fixation, with these enzymes compartmentalized in mesophyll cells (MCs) or bundle sheath cells (BSCs)[8,9]. All known genes that involved in C4 photosynthesis have orthologs in C3 species[10-13]. The same C4 orthologous genes, which exhibit relatively high transcript abundances, are employed to support C4 metabolism across different C4 lineages in parallel[12,14]. However, it remains largely unclear how C4 genes evolved to

[1]State Key Laboratory of Plant Molecular Genetics, Center of Excellence for Molecular Plant Sciences, Chinese Academy of Sciences, Shanghai 200032, China. [2]State Key Laboratory of Plant Genomics, Institute of Genetics and Developmental Biology, Innovation Academy for Seed Design, Chinese Academy of Sciences, Beijing, China. [3]School of Life Sciences, Institute of Life Sciences and Green Development, Hebei University, Baoding, China. [4]State Key Laboratory of Genetic Engineering, School of Life Sciences, Fudan University, Shanghai 200438, China. [5]Biotechnology Research Institute/National Key Facility for Gene Resources and Gene Improvement, Chinese Academy of Agricultural Sciences, 100081 Beijing, China. [6]University of Chinese Academy of Sciences, 100049 Beijing, China. [7]Center for Genomics and Biotechnology, Fujian Provincial Key Laboratory of Haixia Applied Plant Systems Biology, Key Laboratory of Sugarcane Biology and Genetic Breeding, National Engineering Research Center for Sugarcane, College of Life Sciences, Fujian Agriculture and Forestry University, Fuzhou, China. [8]China National GeneBank, Shenzhen 518120, China. [9]These authors contributed equally: Ming-Ju Amy Lyu, Huilong Du, Hongyan Yao, Zhiguo Zhang. ✉e-mail: lvmj@cemps.ac.cn; cliang@genetics.ac.cn; lutiegang@caas.cn; zhuxg@cemps.ac.cn

acquire increased transcript abundances necessary for $C_4$ photosynthesis.

Among dicotyledonous model systems for $C_4$ photosynthesis, the genus *Flaveria* is notable for encompassing $C_3$, $C_4$, and many intermediate species[15]. *Flaveria* intermediate species are classified into $C_3$–$C_4$ and $C_4$-like types, with the latter performing a full $C_4$ metabolic pathway alongside $C_3$ metabolic pathway. $C_3$–$C_4$ species are featured with decreased $CO_2$ compensation points as a result of performing photorespiratory glycine shuttle, which was acquired by localizing glycine decarboxylase activity into BSCs and thus resulting in high $CO_2$ concentration in BSCs[2]. $C_3$–$C_4$ species are further classified into type I $C_3$–$C_4$ species (with no or minimal $C_4$ metabolism) and type II $C_3$–$C_4$ (with moderate $C_4$ metabolism)[16]. Note that the term intermediate species does not necessarily refer to transitional forms, but may instead represent alternative evolutionary outcomes within the spectrum of photosynthetic strategies[17]. The *Flaveria* genus serves as an ideal model for investigating how $C_4$ genes evolved from non-photosynthetic genes and adapted to function in $C_4$ photosynthesis.

Decades of research on the genus of *Flaveria* have significantly advanced our understanding of the evolution of $C_4$ photosynthesis[15,18–20]. Recently published short-read assembly-based reference genomes of four *Flaveria* specie provide valuable resources for protein-coding gene annotation in *Flaveria* genus[21]. Analyses of gene regulatory networks (GRNs) using long-read transcriptomic sequencing have provided critical insights into the evolution of $C_4$ photosynthesis in the genus *Flaveria*[20], emphasizing the pivotal role of transcriptional regulatory mechanisms in shaping the $C_4$ pathway across different *Flaveria* species. However, due to a lack of high-quality reference genomes, our understanding of the regulation of $C_4$ photosynthetic genes in the *Flaveria* genus is still incomplete. Existing short-read assembly-based *Flaveria* reference genomes are fragmented, potentially limiting their utility in identifying *cis*-regulatory elements (CREs) crucial for $C_4$ photosynthesis. In this study, we present chromosome-scale reference genomes of five *Flaveria* species, generated using long-read genome sequencing technology. Based on high-quality *Flaveria* genomes generated, we further conducted an integrated multi-omics study focusing on gene duplications and transcriptional and translational regulations during the evolution of $C_4$ photosynthesis, aiming to uncover the mechanisms underlying the elevated mRNA and protein levels of $C_4$ genes in $C_4$ photosynthesis.

## Results
### Chromosome-scale genome assemblies of five *Flaveria* species
The genome sequences of five *Flaveria* species, *i.e., F. robusta* (Frob, $C_3$), *F. sonorensis* (Fson, type I $C_3$–$C_4$), *F. linearis* (Flin, type I $C_3$–$C_4$), *F. ramosissima* (Fram, type II $C_3$–$C_4$), and *F. trinervia* (Ftri, $C_4$)[22,23] were constructed using PacBio RSII single-molecule real-time (SMRT) sequencing technology (Fig. 1a). The assembled genome size gradually increased during the evolution of $C_4$ photosynthesis in this genus, ranging from 0.55 Gb in the $C_3$ species Frob to 1.26–1.66 Gb in the $C_3$–$C_4$ species, and reaching 1.8 Gb in the $C_4$ species Ftri (Supplementary Fig. 1a). These findings were corroborated by flow cytometry analysis (Supplementary Data 1). Chromatin conformation capture (Hi-C sequencing) analysis revealed that 98% to 99% of the assembled genome sequences were anchored to 18 pseudo-chromosomes (Fig. 1b, Supplementary Fig. 1b and Supplementary Data 2). To verify the chromosome number of sequenced species, we examined the chromosome number of Frob, Flin, and Ftri using fluorescence in situ hybridization (FISH). All three analyzed species exhibited a chromosome number of $2 \times 18$ (Fig. 1c), consistent with the reported diploid chromosome number of 36 (2n) for all five *Flaveria* species[15].

The genome completeness of the *Flaveria* species sequenced in this study was assessed using Benchmarking Universal Single-Copy Orthologues (BUSCO) genes and showed coverage ranging from 92.5%

to 99.2%. In line with the high genome completeness, the average genome mapping rate of RNA-seq reads across the five *Flaveria* species was 94.8%, ranging from 86.7% to 97.6% (Supplementary Data 3). A relatively lower RNA-seq genome mapping rate (86%) was observed in Flin compared to the other four species, likely due to the use of different accessions of Flin for RNA-seq (Sugarloaf Key population) and genome assembly (Yucatan population)[22].

A recent whole genome duplication event (WGD), referred to as WGD2, occurred in Asteraceae species, including in *Helianthus annuus* (sunflower) approximately 29 million years ago (mya)[24]. *Flaveria* and sunflower were estimated to have diverged approximately 31.7 mya based on our calibrations (Fig. 1a). To determine whether *Flaveria* shared the WGD2 with sunflower, we analyzed the Ks distribution of paralogs in both Ftri and sunflower. Two distinct peaks were identified in sunflower, with the higher Ks peak corresponding to WGD2. Similarly, *Flaveria* species exhibited a peak within the same Ks range (Fig. 1d and Supplementary Fig. 2), suggesting that *Flaveria* shared the WGD2 with sunflower.

Although the genome size of the $C_4$ species Ftri was tripled compared to the $C_3$ species Frob, the number of protein-coding genes remained comparable, with 35,875 protein-coding genes in Frob ($C_3$), 32,915 in Ftri ($C_4$), and 37,028 to 38,652 predicted in the $C_3$–$C_4$ species (Supplementary Fig. 1a). The synteny of 18 chromosomes was conserved across the five *Flaveria* species, with 50% to 75% of protein-coding genes being colinear between Frob and the other species (Supplementary Data 2). We compared the predicted protein-coding genes from our assembly with those from Taniguchi's assembly for the shared species *F. robusta* (Frob). The results showed that approximately 93.1% of the genes with protein-coding regions of at least 100 amino acids from Taniguchi's assembly[21] were readily covered by our assembly (Blastp, *E*-value < 0.001, coverage ≥80%) (Supplementary Data 4). Subsequently, we cross-referenced the annotated $C_4$ genes in both assemblies and identified several crucial $C_4$ genes, including *CA1*, *PEPC1*, and *NADP-ME4*, in our assemblies that were absent in earlier assemblies (Supplementary Data 4). This finding underscores the importance of our high-quality chromosome-scale genome assembly in improving the annotation of protein-coding genes in *Flaveria* species.

### Annotating the genes encoding $C_4$ enzymes and transporters
The chromosome-scale assembly of genome sequences and improved gene annotations of five *Flaveria* species enabled the identification of functional $C_4$ gene copies and provided insights into the evolutionary trajectory by which non-photosynthetic genes evolved into photosynthetic genes. We identified eight enzymes and seven transporters as functional copies of $C_4$ genes by integrating both phylogenetic analysis and transcript abundance data (Fig. 2a and Supplementary Data 5, also see "Methods"), These enzymes include *carbonic anhydrase 1* (*CA1*), *phosphoenolpyruvate carboxylase 1* (*PEPC1*), *PEPC kinase 1* (*PEPC-k1*), *NADP-dependent malate dehydrogenase* (*NADP-MDH*), *Aspartate aminotransferase* (*AspAT*), *alanine aminotransferase* (*AlaAT*), *NADP-dependent malic enzyme 4* (*NADP-ME4*), *pyruvate orthophosphate dikinase* (*PPDK*), and *PPDK regulatory protein* (*PPDK-RP*). The transporters identified were *dicarboxylate transporter 2.1* (*DiT2.1* or *DCT*)[25,26], *bile acid sodium symporter 2* (*BASS2*)[27], *sodium: hydrogen antiporter 1* (*NHD1*)[27], *oxaloacetate/malate transporter* (*OMT* or *DiT1*)[28], and *phosphate/phosphoenolpyruvate translocator 1* (*PPT1*)[29–31]. In addition to these verified transporters, we included *BASS4*, which has been proposed as a pyruvate transporter in bundle sheath chloroplast[32,33]. Higher transcript abundance of the gene encoding BASS4 was observed in the $C_4$ species compared to the $C_3$ and $C_3$-$C_4$ species (Supplementary Data 5). Our data showed that *PEPC-k1* and was absent in the Fram plant sequenced in this study (Supplementary Data 6). Notably, the chromosomal locations of all 15 $C_4$ versions of $C_4$ genes were conserved throughout evolution (Fig. 2b).

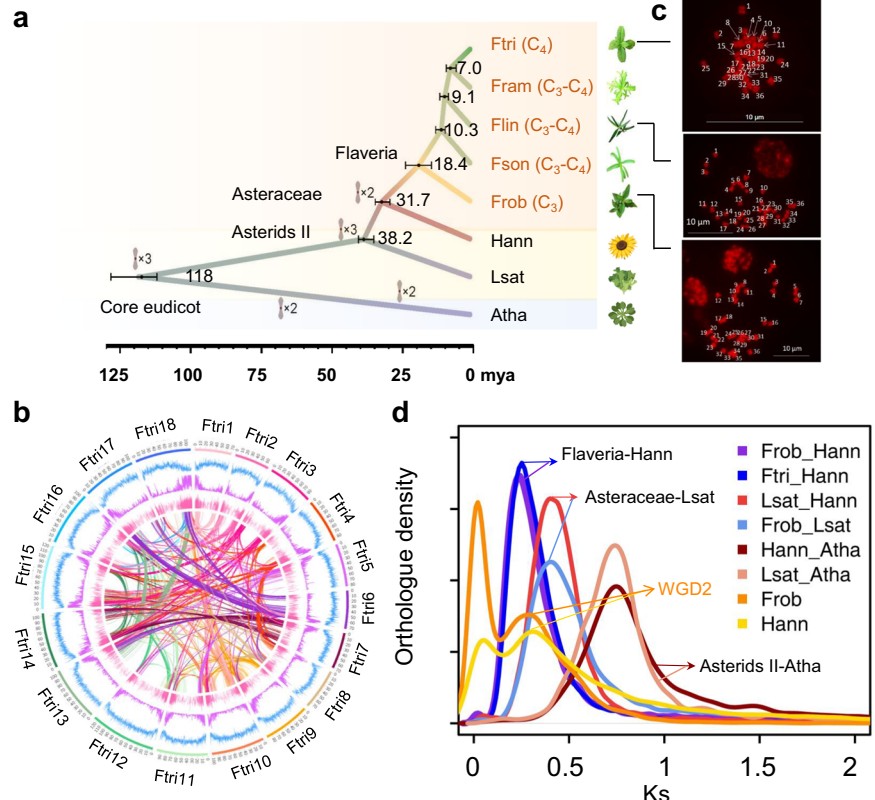

**Fig. 1 | The assemblies and evolution of five *Flaveria* genomes. a** Summary of phylogeny and timescale of the five *Flaveria* species and the three indicated outgroup species, i.e., *Arabidopsis thaliana* (Atha), *Helianthus annuus* (Hann, sunflower), *Lactuca sativa* (Lsat, lettuce). Bars represent 95% confidence intervals of the estimated divergence time. Whole genome duplications are shown at the corresponding node/branch. The plant image of Atha, Hann and Lsat were created in BioRender. **b** The circular representation of pseudochromosomes. From outer to inner side: blue: long terminal repeats density per million base pair (Mb), purple: exon density per Mb, pink: transcript abundance per gene in log10 TPM (transcript per kilobase per million mapped reads). Lines in the inner circle represent links between synteny-selected paralogs. **c** Fluorescence in situ hybridization images to assess the chromosome numbers in Ftri, Flin, and Frob. This experiment was conducted in three biological replicates and one representative result (**c**) was shown. **d** Ks between different species and within species. Ks of Frob vs. Hann and Ftri vs. Hann are shown to represent a speciation between *Flaveria* and Hann. Ks of Lsat vs. Hann, Frob vs. Lsat are shown to represent the speciation of Asteraceae and Lsat, and Ks of Hann vs. Atha, Lsat vs. Atha are shown to represent the speciation of Asterids II with Atha. Ks of paralogs in Frob and Hann are shown representing a whole genome duplication event (WGD). Frob and Hann shared the WGD2 indicated as the second peak in orange "WGD2". The first peak for Ks curves for either Frob or Hann represents tandem duplication found in each species. (Frob *F. robusta*, Fson *F. sonorensis*, Flin *F. linearis*, Fram *F. ramosissima*, Ftri *F. trinervia*).

Since functional copies of $C_4$ enzymes, but not their orthologs in $C_3$ species, have been reported to be induced by light in $C_4$ species[34], we verified the predicted functional copies of seven $C_4$ enzymes by examining their responsiveness to light. *PEPC* and *PPDK* showed light induction in both $C_3$ and $C_4$ species, but the light induction occurred more rapidly in $C_4$ species than in $C_3$ species, i.e., 2 h upon illumination in the $C_4$ species Ftri compared to 4 h upon illumination in the $C_3$ species Frob. Light induction of *CA* was observed in $C_4$ (Ftri) and type II $C_3$–$C_4$ species (Fram and Flin) but not in $C_3$ species (Frob) or type I $C_3$–$C_4$ species (Fson). Light induction of *PEPC-k* and *NADP-ME* was observed in Fram (type II $C_3$–$C_4$) and Ftri ($C_4$), whereas induction of *NADP-MDH* and *PPDK-RP* was restricted in Ftri ($C_4$). Therefore, the light induction of $C_4$ enzymes was most pronounced in $C_4$ species and largely intermediate in $C_3$–$C_4$ species (Supplementary Fig. 3a). These findings align with previous reports[30,35], indicating that $C_4$ genes have evolved to become light-responsive over time. Given that orthologs of these $C_4$ genes play roles in primary metabolism within $C_3$ species[36], the acquisition of light responsiveness during the evolution of $C_4$ photosynthesis enables these genes to better synchronize their activities with those of other photosynthetic genes, which predominantly exhibit light responsiveness. This supports the accuracy of identifying functional copies of $C_4$ genes and highlights the gradual acquisition of

light responsiveness during $C_4$ evolution. Our results also indicate that light responsiveness may serve as a potential criterion for identifying novel $C_4$-related genes.

We also measured enzyme activity for PEPC, NADP-MDH, NADP-ME, and PPDK. The $C_4$ species showed significantly higher enzyme activities compared to all four enzymes than the $C_3$ and $C_3$–$C_4$ species (two-tailed Wilcoxon rank sum tests, BH-adjusted $p < 0.05$, the exact adjusted $p$ values were provided in the Supplementary Fig. 3b). Specifically, the $C_4$ species displayed approximately 10-fold higher enzyme activities for PEPC and NADP-MDH compared to the $C_3$ species. For NADP-ME, the increase was even more pronounced, aligning with recent observations of enzyme activity in *Flaveria* species[22]. Remarkably, Fram ($C_3$–$C_4$) exhibited enzyme activities that were comparable to those of $C_3$ species (PEPC and NADP-MDH) or intermediate between $C_3$ and $C_4$ species (NADP-ME and PPDK) (Supplementary Fig. 3b). Western blot experiment further demonstrated that NADP-ME exhibited intermediate levels between $C_3$ and $C_4$ species in Fram. The other two $C_3$–$C_4$ species, Fson and Flin, showed enzyme activities comparable to those of the $C_3$ species Frob (Supplementary Fig. 3c).

*CA*, *PEPC*, and *PEPC-k* showed additional copies in the $C_4$ species Ftri (Fig. 2c and Supplementary Fig. 4). For example, the $C_4$ version of *PEPC*, identified as *PEPC1* due to its highest transcript abundance

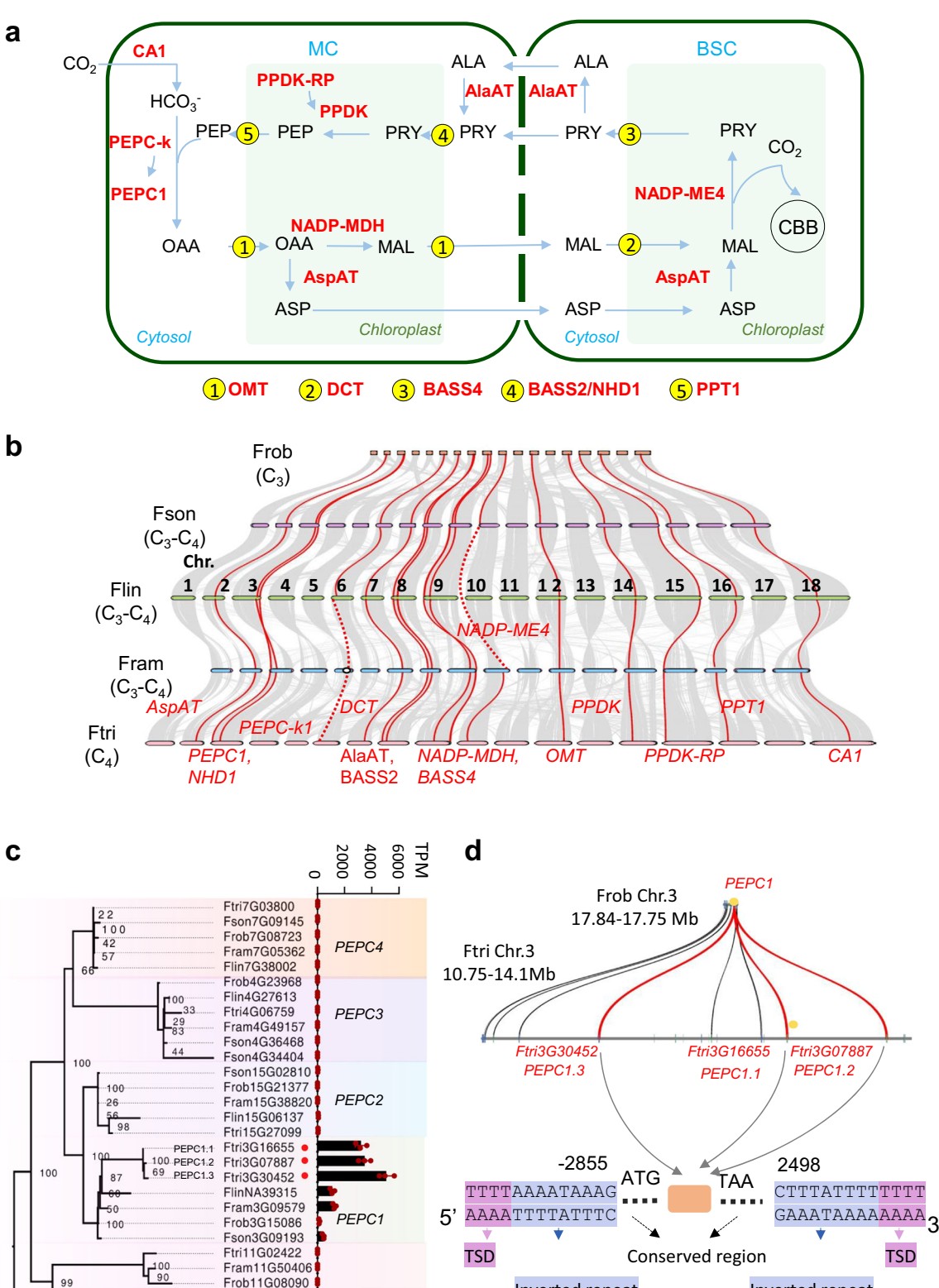

among paralogs in C$_4$ species, contained three copies in the C$_4$ species Ftri but only one copy in the other four *Flaveria* species. The three paralogs of *PEPC1* in Ftri, designated as Ftri*PEPC1.1*, Ftri*PEPC1.2*, and Ftri*PEPC1.3*, were located on the same chromosome (Chr3). The presence of the three Ftri*PEPC1s* paralogs on the chromosome was verified by PCR (Supplementary Data 7). Such duplication events of *CA*, *PEPC*, or *PEPC-k* were not observed in other C$_4$ species, including *Zea*

*mays* (corn; herein Zmay), *Setaria italica* (foxtail millet), or *Sorghum bicolor* (sorghum) (Supplementary Data 8), suggesting that these C$_4$-specific gene duplications were not universal.

Retrotransposons are important mediators for gene duplications through retroposition[37–42]. To determine whether the observed C$_4$ gene duplications were associated with retrotransposons in Ftri, we closely examined the evolution and sequences of Ftri*PEPC1s*. Among

**Fig. 2 | The evolution of C₄ genes in the *Flaveria* genus. a** The diagram of the core C₄ pathway in *Flaveria* C₄ species, C₄ enzymes, and transports are labeled in red. **b** Collinearity of chromosomes among *Flaveria* species. Genes encoding C₄ enzymes and transporters are drawn in red line. Dashed lines represent either failure in anchoring to chromosome (*NADP-ME4* in Flin) or deletion from the genome (*PEPC-k1* in Fram). **c** Gene tree of *PEPC* orthologs, *PEPCs* from *Arabidopsis thaliana* (Atha) are used as outgroups. *PEPC1* (indicated with red circles) is the functional version according to the highest expression levels among all *PEPCs*. Bars on right of tree show gene expression in transcript per kilobase per million mapped reads (TPM). The bars show mean values ± SD. (*n* = 3 biological replicates). **d** Comparative of *PEPC1* in Frob (C₃) and Ftri (C₄). Inverted repeats (blue background) were observed adjacent to the conserved region of *FtriPEPC1.3*. A 4-bp motif (purple background) flanks the inverted repeats, resembling a target site

duplication (TSD) in a transposition event mediated by retrotransposons. (MC mesophyll cell, BSC bundle sheath cell, ALA Alanine, ASP aspartate, CBB Calvin-Benson-Bassham cycle, MAL malate, OAA oxaloacetate, PEP phosphoenolpyruvate, PRY pyruvate, *AspAT aspartate aminotransferase, PEPC1 phosphoenolpyruvate carboxylase 1, NHD1 sodium: hydrogen antiporter 1, PEPC-k1 PEPC kinase 1, DCT dicarboxylate transport 2.1* (or *DiT2.1*), *AlaAT alanine aminotransferase, BASS2 bile acid sodium symporter 2, NADP-MDH NADP-dependent malate dehydrogenase, BASS4 bile acid sodium symporter 4, NADP-ME4 NADP-dependent malic enzyme 4, OMT oxaloacetate/malate transporter or dicarboxylate transporter 1* (*DiT1*), *PPDK pyruvate orthophosphate dikinase, PPDK-RP PPDK regulatory protein, PPT1 phosphate/phosphoenolpyruvate translocator 1, CA1 carbonic anhydrase 1.*). Source data are provided as a Source Data file.

the three Ftri*PEPC1* paralogs, Ftri*PEPC1.1* was predicted to be the ancestral copy, as its mesophyll expression module 1 (MEM1) in the promoter of Ftri*PEPC1.1* was conserved with that of *PEPC1* from the other four *Flaveria* species (Frob, Fson, Fram, and Flin). In contrast, the MEM1 of Ftri*PEPC1.2* and Ftri*PEPC1.3* contained a 109-bp deletion (Supplementary Data 9). Beyond the coding region, sequences approximately 2500 bp upstream and 2000 bp downstream of the coding sequences were also conserved among three Ftri*PEPC1* paralogs (Supplementary Data 9). Closely examining the sequences near the conserved region showed 9-bp inverted repeat sequences, i.e., 5′-AAAATAAAG-3′. Besides, a 4-bp motif, i.e., 5′-TTTT-3′ (Fig. 2d), immediately flanked the invert repeats, resembling a target site duplication (TSD) characteristic of retrotransposon-mediated transposition events. In line with the observation that all three Ftri*PEPC1* paralogs shared conserved gene flanking sequences, particularly the MEM1 motif[43] (Supplementary Data 9), their transcript abundances were similar and higher than those of *PEPC1*s in the other four species (Fig. 2c).

### A major role of transcriptional regulation in elevated protein levels of C₄ genes in Ftri

The increased transcript abundance of C₄ genes in C₄ species is well-documented[44,45]. Here, we investigated how protein abundances were modified during evolution using proteomics. We performed proteomics measurements for the five species with six biological replicates for each (Supplementary Fig. 5). To compare protein and transcript levels of paralogous genes across different *Flaveria* species, we incorporated RNA-seq data of five species from our previous study, which included six replicates for each species[20]. We found that correlations between samples from the same species were higher than those between different species based on either transcript abundances of detected 27,684 genes or protein abundances of 4908 detected proteins (Supplementary Fig. 6a, b), implying the reliability of RNA and protein quantifications.

Transcript and protein abundances of C₄ genes were generally higher in C₄ species compared to C₃ and C₃-C₄ species (Fig. 3a). To investigate whether transcriptional or translational regulation is primarily responsible for the observed differences in protein abundance among species with different photosynthetic types, we compared the protein-to-mRNA ratios (PTR) between genes in five *Flaveria* species. Low PTR genes were defined as those with PTR values less than the mean PTR minus one standard deviation (SD), while high PTR genes had PTR values exceeding the mean PTR plus one SD. The remaining genes were classified as moderate PTR genes (Fig. 3b). An average, 181 low PTR genes (ranging from 138 to 238) and 418 high PTR genes (ranging from 372 to 469) were obtained across the five species (Supplementary Fig. 6c–e and Supplementary Data 10–12). In general, a positive correlation was observed between mRNA and protein levels, with Pearson correlations ranging from 0.36 to 0.53, and most genes exhibited moderate PTRs (Fig. 3c). In C₄ species, seven C₄ genes were identified as low PTR genes, whereas three or fewer C₄ orthologous

genes were classified as low PTR genes in the C₃ and C₃–C₄ species (Fig. 3c).

The low PTR genes were enriched in gene ontology (GO) categories related to photosynthesis, including chloroplast, light harvesting, and PSII (Supplementary Fig. 6f). This result aligns with a previous study in *Arabidopsis*, which reported that photosynthesis-related genes exhibited significantly lower PTRs than other genes in photosynthetic leaf tissues[46] (Supplementary Data 12). C₄ genes showed significantly lower PTRs in C₄ species compared to their orthologs in C₃ and C₃–C₄ species (two-tailed Wilcoxon rank sum test, BH-adjusted $p < 0.05$, the exact adjusted $p$ values were provided in Fig. 3d). In contrast, photorespiratory genes and photosynthesis genes (excluding C₄ genes) showed comparable PTRs between C₃ and C₄ *Flaveria* species (Fig. 3d). These results suggest that the elevated protein levels of C₄ genes in C₄ species during the evolution of C₄ photosynthesis might be primarily attributed to increased transcriptional abundances.

### Translation efficiency of C₄ genes between C₃ and C₄ *Flaveria* species

In addition to transcriptional regulation, factors such as RNA stability, translation efficiency, and protein stability contribute to increased protein abundance. One noteworthy aspect influencing both transcription and translation is the frequencies of G + C at the third positions of codons (GC₃)[47,48]. We compared the GC₃ values of C₄ orthologous genes in the five *Flaveria* species and found no significant differences in GC₃ among them (Fig. 4a).

To further examine whether significant differences in translation efficiency arose during evolution, we performed ribosome profiling (Ribo-seq) on two representative species: the C₃ species Frob and the C₄ species Ftri, with two biological replicates for each. As Ribo-seq captures the positions of ribosomes on mRNAs, it provides a direct measure of translational activity. In parallel, RNA-seq was conducted on the same samples for Frob and Ftri to measure transcript abundances for further translational efficiency estimation (Supplementary Fig. 7a). After filtering out rRNA sequences, approximately 35% and 25% of the reads mapped to the genomes of Frob and Ftri, respectively (Supplementary Fig. 7a), which was comparable to those reported in other species, such as 12% in maize[49] and 16% in Saccharomyces cerevisiae[50]. The Ribo-seq data demonstrated clear triplet periodicity on codons in the reference transcriptome (Supplementary Fig. 7b). The read length distribution peaked at 27 to 32 nucleotides, with 94% of fragments mapping to the gene region (UTR exons, coding exons, and introns) originating from the coding exons (Supplementary Fig. 7c, d).

Principal component analysis (PCA) based on transcript per kilobase per million mapped reads (TPM) of either RNA-seq or Ribo-seq data showed that samples from Frob were well separated from those of Ftri, with the first principal component explaining 63% and 65% of the total variance, respectively (Supplementary Fig. 7e). Consistent with RNA-seq results, C₄ genes from Ftri exhibited higher transcript abundances compared to their counterparts in Frob based on Ribo-seq

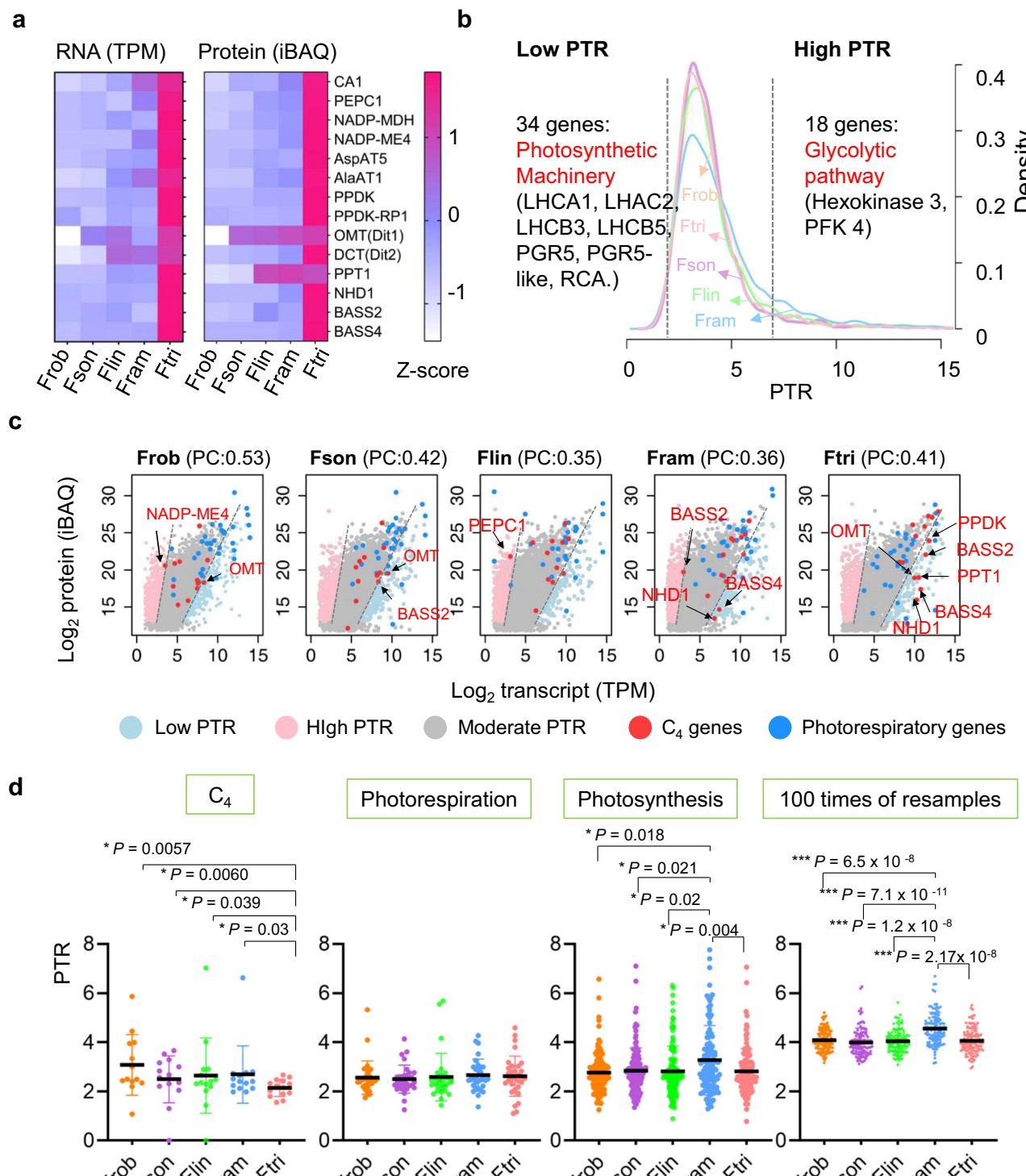

**Fig. 3 | The C4 species showed increased transcript abundances of C4 genes.** **a** Heatmaps show relative transcript and protein abundances of C4 genes in the five *Flaveria* species. Transcript abundance in TPM and protein abundance in iBAQ were normalized using *Z*-score normalization. *PEPC-k1* was excluded as the protein level of C4 version of *PEPC-k1* could not be detected in any of these species. **b** The protein-to-mRNA ratio (PTR) distribution of genes across the five *Flaveria* species. High PTR and low PTR genes are defined as genes with PTR higher than the mean plus one standard deviation (SD) and with PTR values lower than the mean minus one SD respectively. Enriched function of conserved high PTR and low PTR genes across the five *Flaveria* species and their enriched function were shown. **c** Scatter plots of protein vs. transcript abundance of the five *Flaveria* species. Low PTR and high PRT C4 genes were indicated with arrows. Pearson correlation (PC) between protein abundance and transcript abundance is shown in the parentheses on top of each panel. **d** PTR values for the C4 gene set in the five *Flaveria* species, showing that C4 genes have significantly lower PTRs in C4 species Ftri than in the non-C4 species. Data are presented as mean values ± SD. Note that no such decrease is shown for photorespiratory genes, photosynthesis genes, or 100 times of resampling dataset (randomly choosing 14 genes from each species for each resampling). The statistical significance was determined by a one-way ANOVA procedure followed by a two-tailed Wilcoxon rank sum test, *p* values were adjusted with "BH" (*$p < 0.05$, **$p < 0.01$, ***$p < 0.001$). (Abbreviations for the C4 gene are the same as Fig. 2). Source data are provided as a Source Data file.

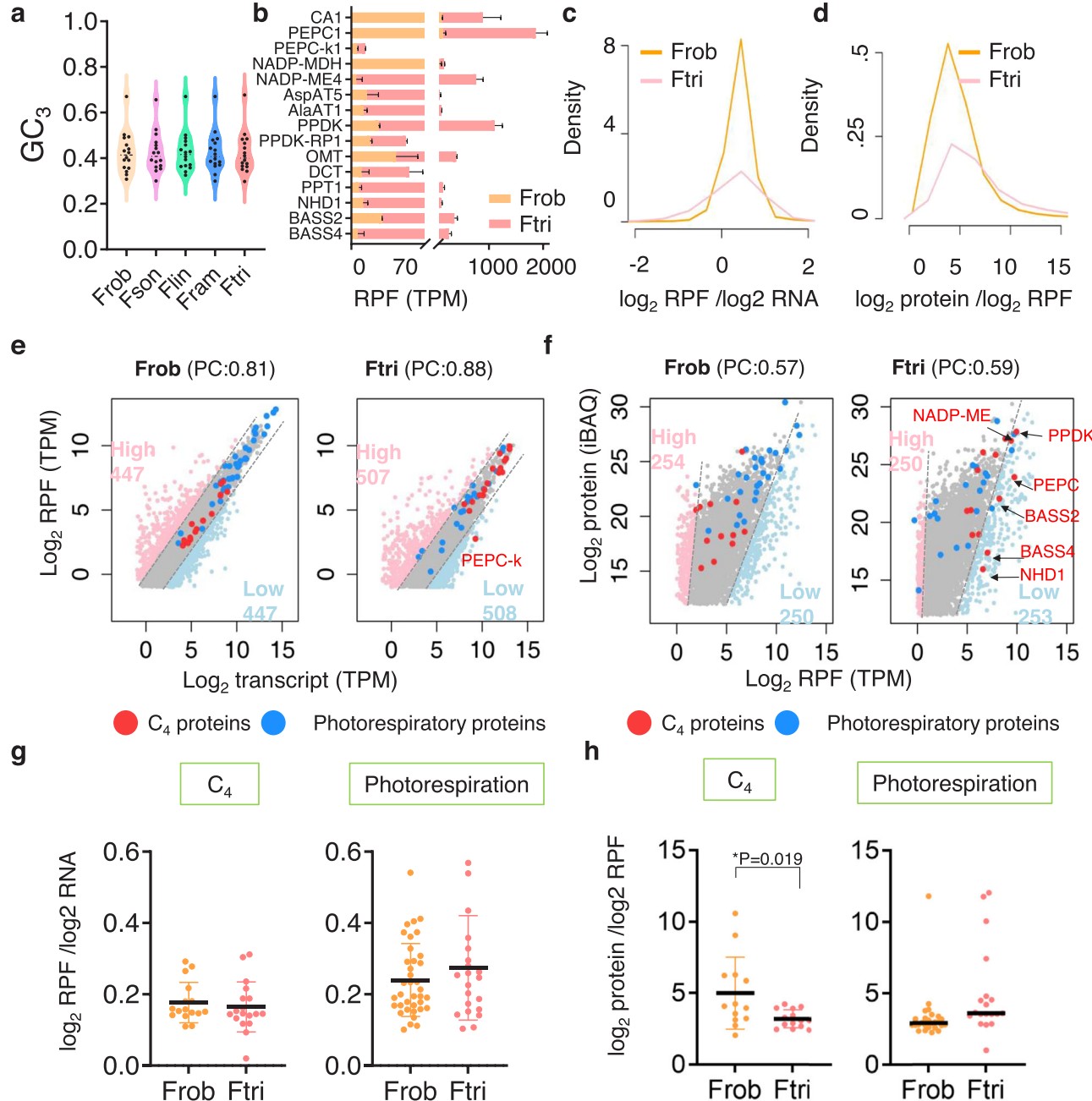

**Fig. 4 | Translation efficiency comparison between Frob and Ftri. a** GC₃ comparisons of C₄ genes across five *Flaveria* species. As Fram*PEPC-k1* is missing, its paralog (Fram*PEPC-k2*, Fram*NAO3444*, see Supplementary Fig. 4) was utilized as a comparison for Fram in this context. **b** Abundances of ribosome protected fragment (RPF) of C₄ genes. Data are presented as mean values ± SD. ($n$ = 2 biological replicates). **c, d** Distribution of RPF-to-RNA ratio and protein-to-RPF ratio. **e** Scatter plots of RPF vs. RNA of Frob and Ftri. Low/high/moderate RPF-to-RNA genes are labeled in pink/light blue/gray. **f** Scatter plots of protein vs. RPF of Frob and Ftri.

Low/high/moderate protein-to-RPF genes were labeled in pink/light blue/gray. Low protein-to-RPF C₄ genes were indicated with red arrows. C₄ and photorespiratory genes are labeled in red and blue, and Pearson correlation (PC) between RPF vs. RNA, protein vs. RPF is shown in the parentheses on top of each panel in (**e**) and (**f**). **g**, **h** RPF-to-RNA ratio and protein-RPF ratio for C₄ genes and photorespiratory genes. Data are presented as mean values ± SD. Statistical significance was determined by two-tailed Wilcoxon rank sum test (*$p < 0.05$, **$p < 0.01$, ***$p < 0.001$). Source data are provided as a Source Data file.

(Fig. 4b). We then estimated the translation efficiency of each gene as a ratio of ribosome-protected fragment (RPF) abundance to RNA abundance and further compared the transcriptional efficiency between species. To ensure comparability, the translation efficiency was normalized by the mean translation efficiency of all photosynthesis genes, excluding C₄ genes. The distribution of translation efficiencies was comparable between Frob and Ftri after normalization (Fig. 4c). Similarly, the distribution of protein-to-RPF abundance ratios was also comparable between Frob and Ftri (Fig. 4d).

A relatively high correlation between RPF and RNA abundances was observed, with Pearson correlation coefficients of 0.81 and 0.88 in Frob and Ftri, respectively (Fig. 4e). The correlation between protein and RPF abundances was intermediate to the RPF vs. RNA and protein vs. RNA correlations, with Pearson correlation coefficients of 0.57 in Frob and 0.58 in Ftri (Fig. 4f). Given that translation efficiency varied over a wide range due to the detection of a relatively large number of genes with both RNA and RPF (~17,000 genes), we defined high and low translation efficiency genes as the top 5% and bottom 5% of genes

ranked by translation efficiency, resulting in 447 and 507 high-translation-efficiency genes and 447 and 508 low-translation-efficiency genes in Frob and Ftri, respectively. Notably, all $C_4$ genes, except PEPC-k in Ftri, and a large proportion of photorespiratory genes were classified as having intermediate translation efficiency in both species (Fig. 4e). Consistently, translation efficiency did not differ significantly between Frob and Ftri for either $C_4$ genes or photorespiratory genes (Fig. 4g).

We also defined high and low protein-RPF-ratio genes the same way as high and low translation-efficiency genes, 254 and 250 high-ratio genes and 250 and 253 low-ratio genes in Frob and Ftri, respectively (Fig. 4f). In line with the protein and RNA comparisons, more $C_4$ genes fell into the category of low protein-RPF-ratio genes, and $C_4$ proteins showed significantly lower protein-RPF ratios in Ftri compared to Frob (two-tailed Wilcoxon rank sum test, $p = 0.019$, Fig. 4h). In contrast, photorespiratory proteins exhibited comparable protein-RPF ratios between these two species. These findings suggest that the observed decreased protein-RPF ratios for $C_4$ genes may primarily attributable to the increased transcriptional abundances rather than changes in translation efficiency.

### Predicted *cis*-regulatory elements and transcription factors associated with the regulation of $C_4$ genes in Ftri

Having established a major role of the transcriptional regulation in $C_4$ genes during evolution, we then explored how *cis*-regulatory elements (CREs) were modified along the evolution of $C_4$ genes. We first investigated the enriched CREs within the promoter regions of $C_4$ genes in the $C_4$ species Ftri. The results revealed that $C_4$ genes in Ftri were enriched with nine known CREs, three of which were identified as ethylene response factor (ERF) CREs. In contrast, the other four species showed at most one enriched CRE of their $C_4$ orthologous genes in ERF (Supplementary Fig. 8). To ascertain whether the ERF CREs were localized within accessible chromatin regions (ACRs) of $C_4$ genes in the $C_4$ species Ftri, we analyzed the enriched CREs within ACRs (ACR-CREs) using data from two biological replicates of transposase-accessible chromatin sequencing (ATAC-seq) experiments (Supplementary Fig. 9).

During ATAC-seq experiments, the Tn5 transposase enzyme shows strong preferential binding to nucleosome-free DNA regions, generating sequencing tags that correspond to open chromatin. Consequently, Tn5 transposase-sensitive sites often exhibit peaks at gene transcription start sites. Our data also showed that Tn5 transposase-sensitive sites in the ATAC-seq reads showed a prominent peak upstream of gene transcription start sites (Supplementary Fig. 9a, b). We obtained 14,443 conserved peaks from the two replicates after applying Irreducible Discovery Rate (IDR) less than 0.05 (Supplementary Data 13), with 48% of Tn5 peaks mapping to the gene promoter region (3k bp upstream of the start codon) (Supplementary Fig. 9b). Tn5 peaks were evident in the promoter regions of photosynthetic genes and $C_4$ genes (Supplementary Fig. 9c), including *Rubisco small subunit 1b* (*RUBSC1b*), *Light-harvesting complex a 1b* (*Lhca1b*), and *proton gradient regulation 5-like* (*PGR5-like*). Due to the complete sequence identity in the upstream regions of the three *Ftri-PEPC1* paralogs, chromatin accessibility showed consistent patterns across these genes (Fig. 5a).

We categorized gene-associated ACRs-CREs into three types according to their distance from the nearest gene: genic (gACR-CREs; overlapping a gene), upstream (upACR-CREs; within 3 kb upstream of a gene's start codon), and downstream (downACR-CREs; within 3 kb downstream of a gene's stop codon). We then calculated enriched CREs in ACR-CREs ("Methods"). Among all three types of ACR-CREs, ERF CREs were the most abundant enriched CREs (Supplementary Fig. 9d). Moreover, ERF CREs dominated the enriched ACR-CREs of $C_4$ genes (Fig. 5b) and were also prevalent in photosynthetic and photorespiratory genes (Supplementary Fig. 9e). Notably, ERF CREs were

abundant in photosynthesis-related genes of other $C_4$ species, including maize, foxtail millet, and sorghum (Supplementary Data 14).

We have constructed gene regulatory networks (GRNs) for Frob, Fson, Fram, and Ftri based on at least 22 RNA-seq datasets previously[20]. In this study, the GRNs for these four species were further refined by incorporating species-specific gene annotations. Additionally, we developed a GRN for Flin, leveraging data from 18 RNA-seq datasets explicitly generated for this study (Supplementary Data 15). Besides, transcription factors (TFs) without predicted cognate CRE families within 3 kb upstream of the start codon were filtered out. Sub-GRNs comprising $C_4$ genes and their regulating TFs were constructed for each species and were termed $C_4$GRNs. The $C_4$GRNs of Frob, Fson, Fram, and Ftri reconstructed in this study were largely consistent with previously constructed GRNs annotated using transcriptomic data of Fram[20]. However, the number of regulated TFs was increased due to improved TF annotations based on our high-quality genome assemblies (Supplementary Data 15).

Regarding the total annotated TFs, the number of TFs within each TF family remained comparable across all five *Flaveria* species. However, notable distinctions were observed for TFs within the $C_4$GRN (Fig. 5c, d and Supplementary Fig. 10). The ERF, bHLH, MYB, NAC, and C2H2 emerged as the top five most abundant TF families in the $C_4$GRN of the five *Flaveria* species (Fig. 5d). In the $C_4$ species Ftri, 323 TFs were predicted to regulate $C_4$ genes (Fig. 5e). Notably, ERF TFs were much more prevalent in the $C_4$GRN of the $C_4$ species compared to other species, either considering the total number of ERF TFs in $C_4$GRN or the number of ERF TFs per $C_4$ genes (Supplementary Data 15). In contrast, the number of predicted ERF TFs were comparable across all five *Flaveria* species (Fig. 5c). These findings suggest a preferential recruitment of ERF TFs for regulating $C_4$ genes during the evolution of the genus *Flaveria*.

We performed an electrophoretic mobility-shift assay (EMSA) to verify the predicted regulation of $C_4$ genes by ERF TFs in the $C_4$ species Ftri. Seven ERF TFs predicted to regulate Ftri*CA1*, *PEPC1.1* and *PEPC-k1.1* were selected for the EMSA experiment. Cognate ERF TF binding site (TFBS) located within 2 kb upstream of their (*CA1*, *PEPC1.1*, or *PEPC-k1.1*) start codon were selected to perform EMSA. EMSA experiments verified the binding of ERF12 to the promoter of Ftri *PEPC1.1* and *PEPC-k1.1*, as well as the binding of ERF61, ERF51, and ERF1 to the promoter of *CA1.1* (Fig. 5f). We then performed transient transcription assay to further verify the regulation of ERF12 and ERF61 on $C_4$ genes. We used dual-luciferase reporter plasmids, containing the firefly luciferase (*LUC*) gene driven by *CA1.1* promoter (200 bp from the start codon) and *PEPC1.1* promoter (250 bp from the start codon) and the Renilla luciferase (*REN*) gene driven by the constitutive 35S promoter, in the analysis (Fig. 5g). The results showed that ERF12 and ERF61 displayed significantly higher LUC/REN ratio compared with Flag tag (Fig. 5h, i), suggesting the activation role of ERF12 and ERF61 on *CA1.1 and PEPC1.1*.

Taken together, these results indicate that ERF CREs and ERF TFs are involved in the regulation of $C_4$ genes in the $C_4$ species Ftri.

## Discussion

The high-quality chromosome-level genome assemblies and multi-omics data from five *Flaveria* species provide valuable resources for investigating the evolutionary and regulatory mechanisms of $C_4$ photosynthesis. Our comprehensive study revealed that tandem duplication, rather than whole-genome duplication (WGD), likely contributed to the increased transcript abundance of $C_4$ genes. Mechanisms controlling the up-regulation of $C_4$ genes' protein abundances during evolution were also explored, revealing that increased RNA abundances may predominantly drive the observed increased ribosome-protected fragments (RPFs) and protein abundances of $C_4$ genes. Additionally, ethylene response factor (ERF) transcription factors (TFs) and their cognate *cis*-regulatory elements (CREs) were identified as being associated with the regulation of $C_4$ genes in the $C_4$ species Ftri.

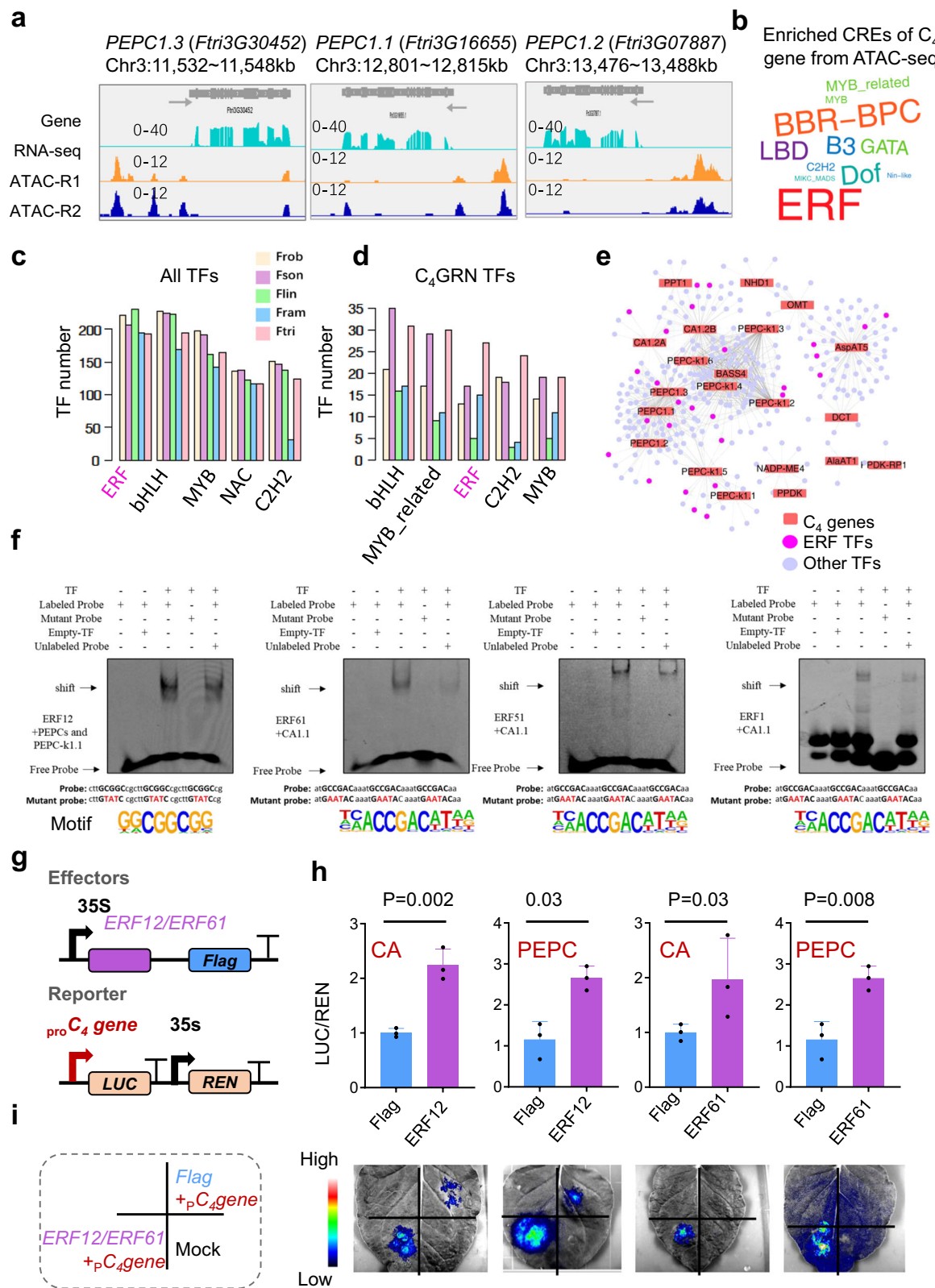

Transposable elements have contributed to the expansion of genome size in *Flaveria* species during evolution (Supplementary Fig. 1). In this study, we found that three C4 genes (*CA1*, *PEPC1*, and *PEPC-k1*) gained additional copies through retrotransposons, contributing to the elevated transcript abundances of these genes in the C4 species Ftri (Fig. 2c and Supplementary Fig. 4). While the duplication is intriguing, its specific role in C4 evolution remains speculative.

We propose that it might be a beneficial event facilitating the progression from the C3–C4 intermediate to a full C4 state, without implying an inevitable progression towards a full C4 photosynthetic pathway. Given that the chromosome number remains consistently at 2 × 18 for known *Flaveria* species, with the exception of two *F. pringlei* collections, the only known polyploids in this genus[15], WGD events appear to be rare in the *Flaveria* genus. This contrasts with

**Fig. 5 | Identifying *cis*-regulatory elements and TFs that regulate C4 genes in Ftri. a** Integrated Genome Viewer (IGV) of RNA-seq reads and two-biological replicates of ATAC-seq reads of three *PEPC1s* in Ftri (C4). **b** Word cloud shows the log2-transformed frequency of enriched *cis*-regulatory elements (CREs) identified through Monte Carlo permutation testing (FDR < 0.05) in accessible chromatin regions (ACR)-CREs associated with C4 genes based on ATAC-seq in Ftri. **c, d** Bar plots show the top five most abundant TF families of all annotated TFs and TFs that are from the C4 gene-TF regulatory network, respectively, with the latter were termed as C4GRN TFs. **e** C4 gene-TF regulatory network of the C4 species Ftri. Lines represent predicted regulatory interactions between TFs and C4 genes. ERFs are highlighted in pink. **f** Electrophoretic mobility shift assay (EMSA) was performed with ERF TFs and cy5-labled partial DNA sequence (probe) of their regulated C4 genes' promoter from Ftri. Labeled probes were incubated with GST-TF-10xHis protein. GST represents GST−10xHis protein without TF. For competition analysis,

the binding reaction was performed with addition of 200-fold of corresponding unlabeled probe. Bands corresponding to DNA-protein complexes (shift) or free probes are indicated by arrows. Predicted ERF motif was shown on the bottom of each panel. The EMSA experiment was conducted in three biological replicates and one representative result (**f**) was shown. **g** Structure of reporter and effector plasmids for transient transcription assay. For the reporter constructs, 35S promoter, C4 gene (*CA1.1* or *PEPC1.1*) promoter, firefly luciferase (*LUC*) and Renilla luciferase (*REN*) are indicated. For the effector construct, ERF TF (ERF12 or ERF61) was driven by the 35S promoter. **h, i** ERF12 and ERF61 activates the promoters of *CA1.1* and *PEPC1.1* (Significance was calculated with two tailed *T*-test, *n* = 3 biological replicates). Data are presented as mean values ± SD. Leaf epidermal cells of *Nicotiana benthamiana* were transfected with infiltration buffer (Mock), reporter DNA (*pCA1.1* or *pPEPC1.1*) with *Flag*, reporter DNA with *ERF-Flag* (ERF12 or ERF61). Source data are provided as a Source Data file.

*Gynandropsis gynandra*[51,52], where WGD has been implicated as a major factor contributing to the increased gene copy number during the evolution of C4 photosynthesis. The discovery of retrotransposon-mediated gene duplications offers critical insights into the genomic complexity and adaptability that facilitated the evolution of C4 photosynthesis, highlighting the diverse evolutionary strategies underlying C4 photosynthesis.

It has been known that RNA and protein levels of C4 genes are elevated in C4 species compared to C3 species[32,44,45], but the dominant mechanism underlying these changes remains unclear. Transcriptional regulation has been identified as a key mechanism in the evolution of C4 photosynthesis[53,54]. In *Flaveria*, notably in the C4 species Ftri, C4 genes exhibited elevated RNA, RPFs, protein levels compared to non-C4 species (Figs. 3a, 4b and Supplementary Fig. 3). Despite overall comparable protein-to-RNA (PTR) values across the five *Flaveria* species, C4 genes exhibited lower PTR ratios in C4 species compared to non-C4 species (Fig. 3c, d). Moreover, the translation efficiency (defined as the ratio of RPF to RNA) of C4 genes in C4 species and their counterparts in non-C4 species is comparable (Fig. 4e, g), further suggesting that increased RNA abundances predominantly drive the increased RPF and protein abundances of C4 genes. This is consistent with our finding that $GC_3$, which is widely recognized for its impact on translation efficiency[47,48], is comparable among these C4 orthologs across the five *Flaveria* species (Fig. 4a). Nevertheless, it is important to acknowledge that other factors, such as epigenetic regulation and post-transcriptional/post-translational processes, could also influence protein abundance, warranting further exploration.

The acquisition of new *cis*-regulatory elements and new transcription factors has been crucial in enhancing the transcript levels of C4 genes[53,54]. Our research revealed a pronounced association between ERF CREs and ERF TFs with C4 genes in C4 species compared to non-C4 species (Fig. 5). Notably, ERF CREs were highly abundant in genes related to photosynthesis in C4 species, such as maize, foxtail millet, and sorghum (Supplementary Data 14), suggesting the widespread presence of ERF *cis*-regulatory elements in photosynthesis-related genes across various plants. ERF transcription factors are widely recognized for their roles in plant stress responses[55–57], and their heightened association with C4 genes suggests a contributions to the evolution of C4 photosynthesis, possibly as an adaptation to environmental stressors such as low $CO_2$, drought, high light, and high-temperature conditions[58,59]. Transposable elements have also been linked to plant stress responses[60,61] and are capable to drive rapid phenotypic changes[62]. The observed associations between transposable elements, ERF transcription factors, and the evolution of C4 photosynthesis highlight C4 photosynthesis as a strategic adaptation of plants to environmental stresses.

ATAC-seq is an important genomic approach for facilitating the genome-wide identification of *cis*-regulatory elements[63–65]. However, obtaining high-quality ATAC-seq data remains challenging, especially

for non-model species, including those in *Flaveria* genus. In this study, we obtained ATAC-seq data only from C4 species (Ftri). Although considerable effort has been devoted to ATAC-seq experiments in other *Flaveria* species, we were unable to obtain ATAC-seq data of comparable quality to that of the C4 species (Ftri). Based on ATAC-seq data from C4 species (Ftri), we provided evidence that ERF CREs were enriched in the open chromatin regions associated with C4 genes (Fig. 5b). Importantly, the electrophoretic mobility-shift assay (EMSA) and transient transcription assay further verified the regulation of ERF TFs on the expression of C4 genes. Nevertheless, high-quality ATAC-seq data from species of *Flaveria* genus other than C4 species (Ftri) are critical for further deepening our understanding of the regulatory and evolutionary mechanisms underlying the formation of C4-specific photosynthesis, which requires further exploration.

The *Flaveria* genus has historically been used as a model system to study the evolution of C4 photosynthesis, leading a substantial body of knowledge on this topic. With the high-quality chromosome-level reference genomes and comprehensive genomic resources provided in this study, we anticipate that the *Flaveria* genus will accelerate the investigation into the genetic basis of C4 photosynthesis evolution. Initial analyses supported by these data highlight the critical role of transcriptional control, particularly the ERF TFs, in the regulation of C4 gene expression. By leveraging the comprehensive genomic data generated in this study, researchers can further explore the genetic and regulatory elements that drive the development of C4 photosynthesis in *Flaveria* and other plant species. Moreover, this study provides a wealth of data that can serve as a foundation to explore the genomic features and evolutionary stages of different intermediate species within the *Flaveria* genus. For instance, our comprehensive dataset allows detailed comparisons between C3−C4 species from clade A (Fson and Fram) and clade B (Flin) of this genus. Such analyses may uncover the mechanisms underlying the absence of true C4 photosynthesis in clade B, thereby providing deeper insights into the evolutionary dynamics and genetic factors that influence photosynthetic pathway development.

## Methods

### Plant materials and fluorescence in situ hybridization assay

*F. robusta* (Frob, C3) and *F. ramosissima* (Fram, C3−C4) were provided by Professor Peter Westhoff from Heinrich Heine University, Germany. *F. sonorensis* (Fson, C3−C4), *F. linearis* (Flin, C3−C4), and *F. trinervia* (Ftri, C4) were obtained from Professor Rowan F. Sage at the University of Toronto, Canada. The plants were grown in soil in a greenhouse as depicted in ref. 30.

The chromosome numbers of Frob, Flin, and Ftri were determined using fluorescence in situ hybridization assay (FISH). Mitotic metaphase spreads of meristem root tip cells were prepared following[66]. FISH was performed following[67] with slight modifications, which were depicted in Supplementary Data 2.

## Genome sequencing

Genomic DNA was extracted from young leaves. PacBio sequencing libraries were constructed following the guidelines of Pacific Biosciences (USA). DNA fragments of 0.5–18 kb were selected using BluePippin electrophoresis (Sage Science, USA). Libraries were then sequenced on the PacBio Sequel platform (PacBio, USA). The N50 of PacBio reads ranged from 16.4 to 21.9 kbp. Approximately 120 GB of data were produced for each species on average. Genome coverage ranged from 66.9-fold (Ftri) to 232.2-fold (Frob). Besides, short reads were sequenced using the Illumina X Ten platform in paired-end 150 bp mode. Approximately 200 million short reads were obtained for each species and used for genome assembly polishing and completeness estimation. Hi-C libraries were constructed following[68]. Two Hi-C libraries were constructed for each species with an insert size of ~350 bp, and sequenced on the Illumina X Ten platform. Between 291 Gb and 325 Gb of 150-bp paired-ended clean data were generated for each species.

## De novo assembly

*Flaveria* nuclear genome sequences were assembled into 18 pseudo-chromosomes in a step-wise way. Sequencing adapters were removed, and low-quality or short reads were filtered using PacBio SMRT Analysis package with the following parameters: readScore = 0.75; min-SubReadLength = 50. The remaining high-quality PacBio subreads were then corrected and contigs were assembled using Canu (v1.8)[69] with the following parameters: useGrid = true, minThreads = 4, genomeSize = 1200 m, minOverlapLength = 500, minReadLength = 1000. For contig polishing, the Illumina paired-end reads were mapped to assembled contigs applying bwa mem (bwa v0.7.17)[70], low qualified mapped reads were filtered off applying samtools (v1.11)[71] with q30 setting. Pilon (v1.22)[72] was used for polishing with the following parameters: –mindepth 10 –changes –fix bases.

For Fram specifically, the BioNano next-generation mapping system was used to facilitate high-quality genome assembly. DNA was labeled at Nt.BspQI sites using the IrysPrep kit (BioNano Genomics, USA). Molecules collected from BioNano chips (BioNano Genomics, USA) were de novo assembled using RefAligne and Assembler from the BioNano[73] using the following parameters: -U -d -T 20 -j 4 -N 10 -i 5, resulting in the optical genome maps. Next, genome assembly generated from Pilon (v1.22)[72] mentioned above was then evaluated and corrected by aligning with the optical genome maps. Corrected contigs and optical genome maps were aligned and merged using hybridScaffold.pl[73], resulting in hybrid scaffolds. Next, HERA[74] was used to fill gaps in the obtained hybrid scaffold in the following parameters: InterIncluded_Side = 30000, InterIncluded_Identity = 99, InterIncluded_Coverage = 99, MinIdentity = 97, MinCoverage = 90, MinLength = 5000, MinIdentity_Overlap = 97, MinOverlap_Overlap = 1000, MaxOverhang_Overlap = 100, MinExtend_Overlap = 500. Obtained hybrid scaffolds were then used for the following assembly.

Subsequently, assembled genome sequences were improved using Hi-C data in two steps. First, contigs were corrected using Hi-C data. Briefly, low-quality Hi-C data (over 10% N base pairs or Q10 < 50%) were removed, and remaining reads were mapped to assembled contigs applying bwa (v0.7.17)[70] with "aln" settings and other parameters were in default (https://bio-bwa.sourceforge.net/bwa.shtml). Only uniquely mapped reads were used for re-assembly. Invalid mapping was filtered using HiC-Pro (v2.11.1)[75] with the following settings: mapped_2hic_fragments.py -v -S -s 100 -l 1000 -a -f -r -o. Next, corrected contigs were re-assembled into a scaffold using LACHESIS[76] with the following parameters: CLUSTER MIN RE SITES = 770, CLUSTER MAX LINK DENSITY = 2, CLUSTER NON-INFORMATIVE RATIO = 2, ORDER MIN N RES IN TRUNK = 578, ORDER MIN N RES IN SHREDS = 593.

## Annotation of transposable elements

To predict transposable elements (TEs), whole genome sequences of the five *Flaveria* species were searched for repetitive sequences individually. A de novo repeat sequence library was constructed using RepeatModeler (RepeatModeler-Open-1.0.5) with the following parameters: RepeatModeler -database database_name -engine ncbi -pa [int]. RepeatMasker (RepeatMasker-Open-4.1.0) was then used to search for similar TEs against the de novo library with the following parameters: RepeatMasker genome. fa -lib de_novo_library -nolow -no_is -q -engine rmblast -pa [int] –norna. Intact long terminal repeat retrotransposons (LTR-RTs) were identified using LTR_FINDER (v1.07)[77] and LTRharvest (v1.5.10)[78]. Then LTR_Retriever (v2.9.0)[79] was used to merge the above results with the parameters: LTR_retriever -genome genome.fa -inharvest species.harvest.scn -infinder species.finder.scn –nonTGCA species.harvest.nonTGCA.scn. The insertion time of intact LTR-RT was extracted from LTR-Retriever analysis.

## Annotation of protein-coding genes

Gene models were predicted using a combination of de novo prediction, homology-based, and transcriptome-based strategies. Briefly, Augustus (v2.4)[80], GlimmerHMM (v3.0.4)[81], GeneID (v1.4)[82], and Genscan (http://genes.mit.edu/GENSCAN.html) were used in combination for de novo prediction. GeMoMa (v1.3.1)[83] was used for homology-based prediction. To facilitate gene annotation, 18 to 32 Illumina RNA-seq datasets were generated either in this study (for Flin, as depicted below) or generated in our previous work[20]. Clean RNA-seq reads were mapped to the genome using Hisat2 (v2.0.4)[84] with " -k 5", and genome-based transcript assembly was performed applying StringTie (v1.2.3)[85] with "-T 0 -F 0". Additionally, de novo transcript assembly was conducted using PASA (v2.0.2)[86] in default parameters based on RNA-seq data. All predicted gene structures were integrated into consensus gene models using EVidenceModeler (v1.1.1)[87], and pseudogenes were predicted applying GeneWise (v2.4.1)[88]. Coding sequence (CDS) failed to be translated either lacking an open reading frame (ORF) or having premature stop codons were removed.

The completeness of protein repertoire was estimated based on: (1) using BUSCO (v3.0.2)[89] against viridiplantae reference, (2) RNA-seq reads mapping to genome applying STAR (v2.7.3a)[90], and (3) 150-bp paired-ended DNA sequencing reads mapping to genome apply bowtie2 (v2.3.4.3)[91] (Supplementary Data 3).

Putative gene functions were assigned using the best matches to GO, KEGG, Swiss-Prot, TrEMBL, and a non-redundant protein database (NR) using BLASTP (v2.2.31+)[92] with the E value threshold of 1e-5.

Transcription factors were predicted using the online website PlantTFDB (v5.0)[93,94] (http://planttfdb.gao-lab.org/prediction.php). *Cis*-regulatory elements (CREs) in promoter regions (3 kb upstream of the start codon) were predicted using Plantpan (v3.0)[95] with a score threshold of 0.85.

## Orthologous genes prediction and gene evolution

To predict orthologous groups, protein-coding genes from the five *Flaveria* species, *Arabidopsis thaliana* (Atha), *Helianthus annuus* (Hann, sunflower), and *Lactuca sativa* (Lsat, lettuce) were processed using Orthofinder (v2.3.11)[96] "diamond" was used for sequence search, and "fasttree" was used for tree inference. The protein sequences of Atha (TAIR10), Hann (v1.0), and Lsat (v7) were downloaded from Phytozome (v13) (https://phytozome.jgi.doe.gov/pz/portal.html). For genes with multiple alternative transcripts, the longest one was kept to represent the protein-coding gene.

## Phylogeny and divergence time analysis

To construct the phylogenetic tree, CDS sequences of 1:1 orthologous genes were aligned using MUSCLE (v3.8.31)[97] with the options "-stable -quiet". Alignments of all the CDS were concatenated to create a supermatrix, and then RAxML (v7.9.3)[98] was applied to infer

phylogenetic tree using the following model: GTR (General Time Reversible nucleotide substitution model) + GAMMA (variations in sites follow GAMMA distribution) + I (a portion of Invariant sites in a sequence). To calibrate the evolutionary time, CDS were aligned codon-wisely guided by protein alignment using pal2nal (v14)[99]. The evolutionary time was calibrated using mcmctree in PAML package (v4.9)[100] using the following parameters: seqtype = 0 (nucleotides), clock = 2 (independent), model = 0 (JC69). The reported fossil divergence time between Hann and Lsat (34–40 million years), as inferred from timetree (http://timetree.org/), was used for calibration. The phylogenetic tree and calibrated evolutionary time were visualized using FigTree (http://tree.bio.ed.ac.uk/software/Figuretree/).

### Synteny analysis between *Flaveria* species
To identify syntenic gene blocks in each species and between Frob and other four species, all-against-all BLASTP (*E* value < 1e − 10, top five matches) (v2.2.31+)[92] was performed for protein coding genes for each genome pair. Syntenic blocks were determined according to the presence of at least five synteny gene pairs using MCScanX (-e 1e-10) (v0.8)[101]. The colinearity of the five species were visualized with JCVI (https://github.com/tanghaibao/jcvi). The circular graphic was plotted using Circos (v0.69-5).

### Estimation of genome duplications and speciation
To estimate the duration of whole genome duplication events and speciation events, pair-wise paralogs and orthologs were aligned in protein sequences, and CDS alignment codon-wise was generated based on protein alignment using pal2nal software (v14)[99]. Synonymous substitution (Ks) values were then calculated using the codeml program in the PAML package (v4.9)[100]. The following parameters were used: runmode = −2, seqtype = 1(codon sequences), codonFreq = 2 (F2X4) and alpha fixed to 0.

### Verification of functional copy of C$_4$ genes using qRT-PCR
As most C$_4$ genes belong to multiple-gene families[12]. The functional copy of C$_4$ genes was determined with the following criteria: (1) the highest transcript abundances within its paralogous group in C$_4$ species and (2) higher transcript abundance in C$_4$ species compared to its counterparts in C$_3$ species. Since the functional copy of C$_4$ genes exhibits faster light responsiveness in C$_4$ species but not in C$_3$ species[30,35], we verified the identified C$_4$ version of C$_4$ genes by investigating the changes of gene expression in response to light induction using quantitative real-time PCR (qRT-PCR). *Flaveria* species were placed in a dark room at 6:00 p.m. The dark-adapted plants were illuminated at 9:00 a.m. the next day. Fully expanded leaves, typically the 2nd or 3rd leaf pair counted from the top, were collected after illumination for 0, 2, or 4 h, and immediately flash-frozen in liquid nitrogen. Samples were stored at −80 °C before processing. RNA isolation and qRT-PCR were performed as previously described[20]. Relative transcript abundances were calculated using ACTIN7 as the reference gene, and the primers used were as described in our previous study[20].

### C$_4$ enzyme western blot and enzyme activity measurements
Western blots for PEPC, NADP-ME, and PPDK were performed using 0.6 g of fresh, fully expanded leaf tissue. Actin was used as a loading control. The antibody of PEPC and NADP-ME were custom-developed by Orizymes Biotechnologies Company (Shanghai). The antibody of PPDK was from Orizymes Biotechnologies Company (Shanghai) (catalog number: PAB07103). The antibody of Actin was from Yamei (Shanghai) (catalog number: LF208S). For all experiments, these antibodies were diluted to a working concentration of 1:5000.

PEPC activity was assayed following the method described in reference[102]. NADP-ME and NADP-MDH activities were determined

following the method described in reference[103]. The PPDK activity was assayed following the method described in reference[104]. A FlexA-200HT UV Spectrophotometer (VARIAN Co. Ltd., USA) was used to monitor the consumption or generation of NAD(P)H at 340 nm.

### RNA-seq and transcriptional quantification for *Flaveria* species
RNA-seq data of Flin were obtained from plants grown under low CO$_2$ (100 ppm) and normal CO$_2$ (380 ppm) for 2 weeks and 4 weeks, respectively. Additionally, plants grown under high light (with PPFD of 1400 µmol m$^{-2}$ s$^{-1}$) and control light condition (500 µmol m$^{-2}$ s$^{-1}$) were sequenced independently. Growth conditions were as described in ref. 20. For RNA extraction, the young fully expanded leaf typically situated on the 2nd or 3rd pair of leaves counting started from the top was used. The selected leaves were cut and immediately frozen into liquid nitrogen and stored thereafter at −80 °C until further processing. Total RNA was then isolated following the protocol of the PureLink$^{TM}$ RNA kit (Thermo Fisher Scientific, USA). The RNA sequencing was performed on the Illumina platform in paired-end mode with a read length of 150 bp. RNA-seq data of the other four species were obtained from our previous study[20].

RNA-seq data for Frob and Ftri were obtained from mature leaves from plants grown in the phytotron with a PPFD of 500 µmol m$^{-2}$ s$^{-1}$, a temperature of 25 °C ± 2 °C, 70% relative humidity, and a 16-h light/8-h dark photoperiod. Two biological replicates were used for each species. Following RNA extraction, mRNA was enriched using mRNA Capture Beads. The RNA sequencing was performed on the Illumina NovaSeq X Plus by Gene Denovo Biotechnology Co., Ltd (Guangzhou, China).

To quantify the expression level of *Flaveria* genes, raw reads were trimmed using fastp (v0.20.0)[105] in default parameters, filtering reads if 40% of bases were unqualified (phred quality < 15). Transcript abundance of genes were calculated by mapping RNA-seq reads to the assembly genome sequence of corresponding species using RSEM (v1.3.3)[106], with STAR (v2.7.3a)[90] as the mapping tool.

### Proteomics of five *Flaveria* species
Approximately 0.1 g of mature leaves were collected from 1-month-old plants and immediately frozen in liquid nitrogen. The plants were grown in the phytotron under the same conditions as those mentioned for the RNA-seq samples of Frob and Ftri. Six biological replicates were prepared for each species. Frozen leaf samples were finely ground and then incubated in 0.6 ml lysis buffer (100 mM Tris-Base, 100 mM EDTA, 50 mM Borax, 50 mM Ascorbic Acid, 30% (m/v) Sucrose, Triton X-100 (final concentration 1%), 10 mM TCEP, 1 mM PMSF, complete EDTA-free protease inhibitor cocktail (PIC) (Roche)). The lysis buffer was freshly prepared, and its pH was adjusted to 8.2 using ammonium hydroxide (NH4OH). After adding TCEP, the pH was readjusted to 8.0. The buffer was stored at −80 °C until needed and thawed at room temperature before use. Samples were centrifuged at 14,000 × *g* for 10 min at 4 °C. The supernatant was retained for total protein extraction. Total protein concentration was determined with a Bradford assay[107].

Details of protein digestion, HPLC fractionation, and LC-MS/MS analysis are provided in Supplementary Data 11. Briefly, peptides were pre-fractionated to generate data data-dependent acquisition (DDA) library. Fractionated peptides were mixed from all the 30 samples (a total of 200 µg). The mixture was separated using a linear gradient and 30 fractions were combined into 15 components. Raw data from each species were utilized to construct libraries based on their respective protein sequences. As a result, five peptide libraries were obtained, one for each species. Finally, data-independent acquisition (DIA) was performed using Spectronaut (version 14.7, Biognosys, Zurich, Switzerland). Default settings for MS1-level quantification were applied. The mass spectrometry proteomics data have been deposited in the PRteomics IDEntifications Database (PRIDE).

For the inter-species comparison among *Flaveria* species, orthologous gene pairs between the remaining four *Flaveria* species and *Frob* were predicted using blast (v2.2.31+)[92]. The top hits were identified with an *E*-value threshold of 1e−5 and a sequence identity requirement of at least 60%. A K-means clustering analysis was performed separately on the transcript abundance and protein abundance data, using the unified *Frob* annotation.

## Ribosome profiling of Frob and Ftri

For ribosome profiling (Ribo-seq), mature leaves of Frob and Ftri were collected from same plants used for RNA-seq, as described above. Two biological replicates were prepared for each species. The leaves were immediately frozen in liquid nitrogen and ground into fine powder.

The ribosome profiling was performed with slight modifications to a previously reported protocol[108]. Specifically, the powder was resuspended in 400 μL of lysis buffer (20 mM Tris-HCl, pH 7.4, 150 mM NaCl, 5 mM $MgCl_2$, 1 mM DTT, 100 μg/mL cycloheximide, and 1% [v/v] Triton X-100). The mixture was incubated on ice for 10 min and then centrifuged at 20,000 × *g* for 10 min at 4 °C. The supernatant was collected.

Ribosome footprints (RFs) were prepared by adding 10 μL of RNase I and 6 μL of DNase I (NEB, Ipswich, MA, USA) to 400 μL of the collected supernatant and incubated at room temperature for 45 min. The nuclease digestion was terminated by adding 10 μL of SUPERase-In RNase inhibitor (Ambion, Austin, TX, USA). Next, 100 μL of the digested RFs was loaded onto a pre-equilibrated size exclusion column (Illustra MicroSpin S-400 HR Columns; GE Healthcare) and eluted by centrifugation at 600 × *g* for 2 min. RFs longer than 17 nucleotides (nt) were isolated using an RNA Clean and Concentrator-25 kit (Zymo Research). Antisense DNA probes complementary to ribosomal RNA (rRNA) sequences were used to remove rRNA, and the RFs were further purified using magnetic beads (Vazyme, China). Ribo-seq libraries were prepared using the NEBNext Multiple Small RNA Library Prep Set for Illumina® (NEB, E7300S and E7300L). Sequencing was performed on the Illumina NovaSeq X Plus by Gene Denovo Biotechnology Co., Ltd. (Guangzhou, China).

For Ribo-seq data analysis, low-quality reads and adapter sequences were filtered and trimmed using fastp (v0.20.0)[105]. Reads with lengths ranging between 20 and 40 bp were retained for subsequent analysis. Remaining reads were mapped to the rRNA database, GenBank, and Rfam database using bowtie2 (v2.3.4.3)[91]. Reads aligned to rRNA, transfer RNAs, small nuclear RNAs, small nucleolar RNAs, and microRNAs were excluded. The remaining reads were aligned to the respective genome using STAR (v2.7.3a)[90] with 2-pass setting enabled. Gene expression levels were quantified with RSEM (v1.3.3)[106]. RFs were assigned to different genomic features (5′UTR, CDS, 3′UTR and intron) according to the position of the 5′ end of the alignments. Three-nucleotide periodicity was visualized using the riboWaltz package (v2.0)[109].

## ATAC-seq for the C4 species Ftri

To isolate nuclei from the $C_4$ species Ftri, fully expanded mature leaves were harvested at 1:00 pm. Approximately 3 g of fresh leaves from five plants were used for each of the two biological replicates. Leaf material was ground in ice in 10 ml 4xNE buffer (40 mM MES -KOH, PH5.4, 40 mM NaCl, 40 mM KCl, 10 mM EDTA, 1 M Sucrose, 0.1 mM spermidine, 0.5 mM spermine and 1 mM DTT). Next, the debris was removed by sieving the mixture through two layers of 70 μm nylon cell strainer into precooled flasks. The filtrate was then centrifuged at 200 × *g* at 4 °C for 3 min to further remove debris. The supernatant was centrifuged at 2000 × *g* at 4 °C for 5 min to spin down Nuclei. Nuclei were lysed by adding 1X NE buffer containing 0.1% (v/v) NP40, and 0.1 (v/v) Tween-20, followed by incubation on ice for 3 min. Nuclei were pelleted by centrifugation at 2000 × *g* at 4 °C for 5 min. Pellets incubated in RS buffer (Tn5 mix, 10 mM Tris-HCL, PH 7.4, 10 mM NaCl,

3 mM $MgCl_2$, 0.01% digitonin, 0.1% OM and 0.1% Tween-20) at 37 °C for 30 min. The Tn5 tagmentation was then terminated under 95 °C for 2 min. DNA was purified using a spin column (Qiagen, Germany) and amplified using index primers matching the Illumina Nextra adapter. The above protocol was provided by Orizymes Biotechnologies (Shanghai) Co., Ltd.

ATAC-seq libraries containing DNA inserts of 50–150 bp were gel-purified and sequenced in Illumina X Ten platform in paired-end 150 bp mode. Raw reads were trimmed using fastp (v0.20.0)[105] in default parameters. Sequencing reads were mapped to the genome sequence of Ftri ($C_4$) using bowtie2 (v2.3.4.3)[91] with the parameter "-k 10". Mapping results were sorted using the "sort" function in samtools (v1.11)[71], and low-quality was filtered off using "view" function in samtools with -q 10. We then used "samtools collate" to group reads with the same names, and "samtools fixmate -m" to fill in mate coordinates and add mate score tags, and "samtools markup -r" to remove duplicate reads. The start position of each read based on strand information was adjusted using alignmentSieve in deepTools (v3.5.0)[110] with −ATACshift. Peaks were identified using MACS2 (v2.2.7.1)[111] using the following parameters: -f BAMPE -g 1.7e9 -q 0.05 −nomodel −keep-dup all −nolambda −shift -100 −extsize 200. The consistency of the two biological replicates was assessed using the irreproducibility discovery rate (IDR) analysis with the IDR package (v2.0.4), following ENCODE guidelines. An IDR threshold of 0.05 was applied to filter irreproducible peaks.

Genes associated with peaks were identified using the "closest" function in bedtools (v2.29.2)[112] with the parameter "-k 2," considering the two nearest genes (upstream and downstream). The distribution of ATAC-seq reads relative to genome features were assessed using the "computeMatrix" function in deepTools (v3.5.0)[110] with the following parameters: −skipZeros −reference Point TSS -a 3000 -b 3000. The results were visualized using the "plotHeatmap" in the same tool. peaks was visualized with IGV (v2.16.0)[113].

Based on accessible chromatin regions (ACRs) from ATAC-seq data of $C_4$ species (Ftri), we employed both a permutation-based method and a Fisher's exact test-based method to predict the enriched CREs associated with $C_4$ genes (including the regions 3 kb upstream of start codons, 3 kb downstream of stop codons, and the gene bodies). FIMO of MEME suite (v5.0.2)[114] was used to identify the occurrences of known CREs of plants within the entire set of ACRs, applying a *q*-value threshold of 0.05. The CRE annotations were sourced from PlantPAN 3.0[95] (https://plantpan.itps.ncku.edu.tw/plantpan3/download/home.php). This analysis identified 1,471,751 occurrences of 277 distinct CREs, with 1858 occurrences of 117 CREs were associated with $C_4$ genes. To assess whether specific CREs were overrepresented near $C_4$ genes beyond random chance, we conducted a Monte Carlo permutation test. For each of the 117 CREs, observed occurrences were compared against a distribution of expected occurrences estimated from 1000 permutations. In each permutation, 1858 CRE occurrences were randomly selected from the total pool, and the frequency of each CRE was recorded. Following the completion of all permutations, the *p* value for each CRE was calculated as the proportion of permutations where CRE occurrences surpassed the observed value. To control for multiple testing, we applied the Benjamini−Hochberg procedure to adjust for the false discovery rate (FDR). For the Fisher's exact test-based method, we evaluated the enrichment of each CRE associated with $C_4$ genes against the background of 1,471,751 total CRE occurrences. The CRE enrichment results were largely consistent between these two methods (Supplementary Data 13).

Furthermore, to predict enriched CREs in ACRs from various genomic contexts, including within gene bodies, upstream and downstream of genes, as well as those associated with photosynthetic and photorespiratory genes, we employed the Monte Carlo permutation test as described above.

## Analysis of enriched CREs in the promoters of C4 genes in five Flaveria species

We employed the HOMER package[115] to identify enriched motifs within the promoters (3 kb upstream of the start codons) of $C_4$ genes and their orthologous counterparts in each *Flaveria* species. For each species, the promoter sequences (3 kb upstream of the start codon) of non-$C_4$ genes were used as the background to account for potential genomic distribution bias. Regarding sequence composition bias, the HOMER package automatically selects background regions from the promoter sequences of non-$C_4$ genes that match the GC-content distribution of the promoter sequences of $C_4$ genes (in 5% increments), as detailed in the HOMER manual ([http://homer.ucsd.edu/homer/ngs/peakMotifs.html](http://homer.ucsd.edu/homer/ngs/peakMotifs.html)). Specifically, if the promoter sequences of $C_4$ genes (input) are highly GC-rich, HOMER selects random regions from GC-rich regions of the promoter sequences of non-$C_4$ genes (background) as a control. In addition to accounting for GC-content bias, HOMER package also applies "autonormalization of sequence bias" to eliminate bias introduced by lower-order oligo sequences associated with the promoter sequences of $C_4$ genes. The HOMER package operates under the assumption that the promoter sequences of $C_4$ genes (input) and non-$C_4$ genes (background) should not exhibit imbalances in 1-mers, 2-mers, 3-mers, etc. After calculating these imbalances for each oligonucleotide, HOMER adjusts the weights of background sequences slightly to normalize the imbalances. This analytical procedure ensured that the enrichment analysis accounted for potential biases in sequence composition and genomic distribution between the promoters of $C_4$ and non-$C_4$ genes.

## Electrophoretic mobility shift assay

To construct plasmids for recombinant protein production, the coding sequences of target ERF proteins were PCR-amplified from cDNA and inserted into pGST (His-ERF) vector to create fusions with 3×Flag, 10×His and GST tags, respectively. The recombinant proteins were expressed in *E. coli* strain Rosetta (DE3) and induced with 0.5 mM isopropyl β-D-thiogalactoside (IPTG) for 2 h at 37 °C for ERF12 (Ftri15G25371, ortholog in Arabidopsis is AT1G28360), ERF11 (Ftri17G17198, ortholog in Arabidopsis is AT3G23240) and ERF57 (Ftri1G11197, ortholog in Arabidopsis is AT5G65130), 0.1 mM IPTG for 1.5 h at 37 °C for ERF61 (Ftri13G23465, ortholog in Arabidopsis is AT1G64380). Bacterial cells were collected and lysed in lysis buffer (20 mM Tris-HCl, 300 mM NaCl, 0.5 mM DTT, and 1× protease inhibitor cocktail). The proteins were released from the collected cells by sonication (100 V, 20 m) and purified with Ni column. DNA fragments were end-labeled with Cy5. The fluorescence-labeled DNA (20 nM) was incubated with purified protein in 5x EMSA/Gel-Shift bidding buffer according to manufactory's instructions (Beyotime, GS005, China) for 30 min at 25 °C. For competition assays, 400 nM unlabeled competitor DNA was also added in the reaction. For the empty-TF control, Glutathione-S-transferase (GST)-3×Flag-10×His without TF proteins was produced in *E. coli* Rosetta (DE3) as described above. The reaction mixture was electrophoresed at 4 °C on a 6% native polyacrylamide gel in 0.5×TBE for 50 min at 100 V. Fluorescence-labeled DNA on the gel was then detected with Typhoon (Typhoon™, Cytiva). All PCR primers are listed in Supplementary Data 15.

## Transient transcription assay

To construct an effector plasmid, the full-length CDS of *ERF12* and *ERF61* was cloned into pCambia1300, driven by the cauliflower mosaic virus (CaMV) 35S promoter to generate Pro35S::ERF12-Flag and Pro35S::ERF61-Flag. To construct a reporter plasmid, the *CA* promoter (200 bps upstream of the start codon) and *PEPC* promoter (250 bps upstream of the start codon) were cloned into pGreenII-0800 to generate pCA::LUC and pPEPC:LUC.

Transient transcription dual-LUC assays were performed using *Nicotiana Benthamiana*. The effector and report plasmids were transformed into *Agrobacterium* strain GV3101 carrying the helper plasmid pSOUP1P19. *Agrobacterium* cultures were cultured overnight and collected by centrifugation at 5000 × *g* for 3 min and resuspended in MES buffer (10 mM MgCl₂, 10 mM MES, 100 μm acetosyringone, PH = 6.0) to 1.5 OD600. Mixed *Agrobacterium* with effector and reporter were incubated at room temperature for 2 h. The *Agrobacterium* suspension was then gently press-infiltrated into healthy leaves of 3-week-old *N. benthamiana* plants with a 1-mL needleless syringe. The plants were grown under 25 °C with photoperiod of 16/8 h day/night for 2–3 days. Luciferase activity was imaged with a CCD camera or quantified with a luminometer (Promega 20/20) using LUC reaction reagents according to the manufacturer's instructions (Yeasen, China).

## Statistics and reproducibility

No statistical method was used to predetermine sample size. No data were excluded from the analyses except for those failing quality control checks. All statistical tests were performed in R (version 4.2.1) with a Benjamini–Hochberg correction applied where applicable. The statistical analysis for each experiment has been described in the main text and figure legends.

## Reporting summary

Further information on research design is available in the Nature Portfolio Reporting Summary linked to this article.

## Data availability

The genome assemblies, gene annotations, proteomics data, and raw reads of transcriptome data, Ribo-seq data and ATAC-seq data are available at the China National GeneBank (CNGB) [[https://db.cngb.org/codeplot/datasets/flaveria](https://db.cngb.org/codeplot/datasets/flaveria)] with project ID CPN0003058. The genome assemblies, gene annotations, transcriptome data, and proteomics data are also available at figshare [[https://doi.org/10.6084/m9.figshare.19918876.v4](https://doi.org/10.6084/m9.figshare.19918876.v4)]. The genome assemblies are also available at the National Center for Biotechnology Information (NCBI) with accession numbers: SAMN14943594 for Frob, SAMN14943595 for Fson, SAMN14943597 for Flin, SAMN14943596 for Fram, and SAMN14943598 for Ftri. The mass spectrometry proteomics data were submitted to the PRoteomics IDEntifications Database (PRIDE)[116] with accession number PXD024720. RNA-seq data of Flin were also submitted to Gene Expression Omnibus (GEO) in the NCBI database under accession number PRJNA827625. RNA-seq data of Frob, Fson, Fram, and Ftri were obtained from published data under project accession PRJNA600545. Source data are provided with this paper.

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

## Acknowledgements

We appreciate Prof. Rowan F. Sage, Prof. Peter Westhoff, Dr. Udo Gowik, and Dr. Matt Stata for sharing *Flaveria* materials, and Ms. Lin Huang (State Key Laboratory of Genetic Engineering, Fudan University) for help on proteomic analysis and Zhen Cao (Zhe Jiang Agricultural and Forestal University) for help on plastid construction. We also thank suggestions from Prof. Haiyang Hu. The work is funded by the National Key Research and Development Program of China (2020YFA0907600, X.G.Z.), the Strategic Priority Research Program of the Chinese Academy of Sciences (XDB0630101, M.J.A.L.; XDB0630301, X.G.Z.), the general program of the National Science Foundation of China (31870214, X.G.Z.).

## Author contributions

X.G.Z., T.L., C.L., and M.J.A.L. conceived and designed the study. H.D., Y.H., Z.Z., Y.Z., H.L., L.F., Q.G., Y.Q., and M.J.A.L. performed genome assembly and annotation. M.J.A.L., Y.H., and X.N. performed genome comparison analysis, qRT-PCR, and ATAC-seq analysis. M.J.A.L., H.Y., Z.Z., F.M., and Y.W. conducted RNA-seq and proteomics analysis. M.J.A.L. and F.C. performed gene regulatory network construction. Y.Y.Z. performed PCR verification of the three paralogs of PEPC1s in Ftri. Q.T. performed Ka/Ks analysis. X.C. and M.J.A.L. performed transcription factor prediction. Q.Z. and J.Z. performed syntenic analysis. T.Y. constructed *Flaveria* workspace in China National GeneBank (CNGB). M.J.A.L., X.G.Z., T.L., C.L., and G.C. drafted the manuscript. M.J.A.L. and X.G.Z. revised the manuscript. All authors reviewed and approved the final manuscript.

## Competing interests

The authors declare no competing interests.
