## [Transparent Peer Review file · Nature Communications]

A dominant role of transcriptional regulation during the evolution of C4 photosynthesis in *Flaveria* species

Corresponding Author: Professor Xinguang Zhu

Version 0:

Reviewer comments:

Reviewer #1

(Remarks to the Author)

Summary:

The authors present 5 high quality genome assemblies for the genus *Flaveria* with associated transcriptomic and proteomic datasets. They go on to conduct a fairly standard set of genomic analyses exploring genome size and gene arrangement, TE expansion, enrichment of cis regulatory elements alongside transcriptomic and proteomic analysis that highlight differences in gene expression, gene regulation by ERF transcription factors, ratios of transcript and protein abundance, and codon usage between species. There is no doubt that the publication of these genomes represents a significant asset in *Flaveria* and to C4 research in general. Furthermore, I do not see any problems with their methods or results. While I believe that some of their results will be of general interest to researchers studying plant genomics, I do not necessarily agree with their interpretations as presented in this manuscript. For example, the tandem duplication of PEPC in *F. trinervia* is interesting but I'm not sure it has anything to do with the transition to C4 in this species. All three copies of PEPC1 exhibit elevated expression in *F. trinervia* which could be interpreted as evidence that the increase in expression occurred prior to and independent of its duplication. That this duplication is not observed in any of the C3+C4 species or other distantly related C4 species render interpretation of its effect, independent of the simple increase in expression, highly speculative. In this case perhaps this could be reframed, if other papers have found evidence that WGD facilitated the transition to C4 and they find evidence of extensive TE-mediated duplication that gene duplication may play a role but that the source of those duplications can vary (i.e. WGD or TE-mediated tandem duplication). However, the duplication of PEPC, on its own is not particularly compelling. Likewise, the increase in light response in several well-known C4 genes does not seem consistent across the species presented here (see my more specific comments below), though in most cases there does appear to be a clear difference between the C3 and C4 species. Indeed, the majority of comparisons ultimately characterize differences between these two extremes with C3+C4 species contributing little to, and at times seemingly contradicting, the authors' overarching explanation of how C4 has arisen in *Flaveria*. This is evident in the discussion where instead of referring to C3, C3+C4, and C4 species individually, the authors merely make distinctions between C4 and "non-C4" species, which presumably encompass all species except *F. trinervia*. Despite this, the narrative seems to be that C4 was gradually acquired in four species following their divergence from *F. robusta*, though this is not explicitly stated in the discussion. To my eye, the explanation is necessarily much more complex and may simply not be captured by these results or even these type of broad -omics datasets.

Without specifically exploring the differences between C3+C4 species and *F. robusta* and *F. trinervia*, I don't see the added value of C3 + C4 species. I think these authors should explore these differences (between C3+C4 and others) more directly and/or make it clear that these species are not transitional between C3 and C4 but merely alternative outcomes.

These type of broad multiomics datasets are not always suited to testing specific hypotheses of evolution as it pertains to a given gene or pathway. Instead, their strength often lies in providing a (necessary) foundation for generating testable hypotheses and conducting more narrowly targeted functional studies.

In focusing on primary differences between well-characterized C4 genes between C3 and C4 species of *Flaveria*, I'm afraid that this paper contributes very little novel insight into the evolution of C4 in this or other plant lineages. Other authors have previously explored the role of gene duplication, differences in expression between C3 and C4 species and the role of various transcription factors. The only central and possibly novel finding of the paper involves ratios of transcript and protein

abundance. However, this finding is barely significant and the authors largely ignore substantial differences among C3+C4 species as well as between them and their C3 and C4 relatives. Without significant revision and clarification I cannot recommend this paper for publication in Nature Communications. That being said, I think it is possible that there are further insights that could really improve the impact of this research. As a single example (though I'm sure there are others), exploring gene regulation and coexpression outside of the canonical C4 pathway, might yield novel insights into downstream effects of duplication, modification and regulatory rewiring of C4 genes. I want to reiterate that I think this dataset is impressive and will represent a significant and foundational resource to further study. I think the analyses conducted are relatively standard but sound. The primary weakness of this paper is that it does not seem to generate any novel insights or testable hypotheses for future research that would utilize the basic genomic resource presented here.

Detailed review:

1. Are C4 lineages the same as "C4-like?"
2. Line 98 - Is Ftri C4 or C4-like?
3. Fig 1 – Ks peaks for WGD2 do not appear to align. Could this be explained by differences in evolutionary rates? You could easily use KsRates (or just compare Ka/Ks) to test this...
4. Lines 167-170: did transcripts have to meet both criteria what if a gene was most highly expressed but did not show increased abundance to C3 species? Which C3 species were used? Also, this could probably go in the methods.
5. Lines 187-192: this section makes it sound like there is a gradual increase in light responsiveness across all (or at least many) C4 genes. I do not see this in Extended Data Fig 2. While the difference between Frob and Ftri is clear in most comparisons, I think that this section overstates the significance of this trend and makes it sound like it is a smooth transition from C3 -> C3+C4 -> C4. Some examples:
 - a. PEPC1 – is significantly elevated in at least one comparison in all species. Though the magnitude/abundance does increase in Ftri, it doesn't in the C3+C4 species.
 - b. NADP-MDH shows no clear trend to my eye and is barely significant in Ftri comparisons.
 - c. PPKK is significantly higher at 4h in Frob, higher at 0h in two C3+C4 species and not significant in Ftri.
6. The duplicated copy of PEPC is interesting at first glance. However, it seems strange that you do not see this duplication in any of the C3-C4 species, since they also share elevated expression of C4 PEPC. Isn't it possible that elevated expression (and the transition to C4) preceded this duplication, especially since increases in copy number are not evident in other C4 species?
7. Fig 2 – some stylistic comments: the background for the chloroplast in a. is distracting and makes the text (especially arrows) difficult to read. This would read much better with a flat background color? Lines in d. look roughly drawn. They would be easier to read if they were straight or at least smoothed (similar to the synteny plot in b.).
8. Up to editor- should ERF be spelled out the first time?
9. Line 286-287 why were these TFs filtered out? More importantly, what proportion of TFs have predicted cognates in this species (or are you using predictions from Arabidopsis?). It seems that this approach might filter out important TFs without a predicted binding site which could significantly bias your results toward TFs based on functional screening in a distantly related species. Likewise, it might lead you to exclude potentially important ones that have no predicted binding site, which are presumably numerous.
10. This may just be my ignorance about ATACseq, but I'm a little confused by Fig 3 b. Is this showing THS within the coding sequences of genes? i.e. downstream of TSS? That seems opposite of what I would expect.
11. Based on your trends in TF abundance (Fig 3 f,g) I can sort of understand why you chose to focus on ERFs. However, the abundance plots seem to be more correlated to species than the presence of C4, not just in ERFs but other TFs as well. Instead of a gradual increase in ERF (or any other family) from C3 to C4, Ftri (C4) is almost always highest, followed by Fson (Type-I) and Frob(C3) with the other two C3+C4 lineages considerably lower. This seems to conflict with the species phylogenetic relatedness as well as their degree of C4. On a related note, I feel like almost as strong an argument could be made for focusing on C2H2 TFs as ERF. While there is no doubt that TFs play an important role in the evolution of C4 lineages, I think the role of expansion of large TF families is likely more nuanced than what is presented here.
12. Line 347 - "significantly lower" hmm... I suppose this all depends on your definition of "significant." $P=0.003$ is significant by most standards but $P=0.02$ may not be. While these cutoffs are entirely arbitrary, I think that at the very least you need to explicitly state your significance threshold in this statement (e.g. "C4 genes showed significantly lower PTRs ($P < 0.05$)...") and provide justification for this cutoff, considering this is one of the main findings of your research.
13. Lines 350-352 – I'm not sure I understand what you're saying here maybe you mean "... which can be attributed to increased transcriptional abundance"?
14. Fig 4 d –Why do P-values appear highly significant for the random resampling? Also, while you mention significant difference in photorespiration the difference between Ftr and Fram also appears significant in photosynthesis related genes. These p-values are barely significant and many of the random comparisons seem highly significant including one between Ftr and Fram. I'm not convinced, based on the results as presented in this figure, that there is evidence of a significant difference in PTR as stated here.
15. Lines 376 – 377 – why is this notable? Is there some reason you would expect codon bias to change in C4 species? Is there some other example of this in other C4 lineages? I'm just not seeing the link between codon usage and the transition to C4. Without more justification this could be greatly reduced or moved to the supplement. It seems that you are just verifying that this bias is highly conserved between plant lineages, which does not appear to be a novel finding.
16. Extended Data Fig 2 – I assume the Fram PEPC-k is the paralogous copy mentioned in Fig 5. If so it should be stated here
17. Extended Data Fig 3 – ERFs appear to be extremely common throughout the Ftri genome. While I'm not exactly sure how to interpret this plot in any quantitative sense, They appear to be dominant in both ACR categories as well as light responsive genes. I think that the expansion of ERF TFs is potentially an interesting difference between Ftri and the other species but I'm not sure how this links them to the transition to C4.

18. Lines 447-462 – As this result seems to confirm previous findings and no difference is found between C3 and C4 species I'm not sure it merits its own paragraph in the discussion (or its own results section for that matter).

Minor edits:

Line 46: 'both' is unnecessary.

Line 220: "franking" should be "flanking."

Fig 3d "Cut" or "CUT"?

Line 162 "gene" should be "genes."

Fig 5 – When you say: "FramPEPC-k2, FramNA03444, see Extended Data Fig. 3" I assume you mean Extended Data Fig. 4.

Reviewer #2

(Remarks to the Author)

The study by Lyu et al. provides a comprehensive analysis of the genomic and transcriptional changes associated with the evolution of C4 photosynthesis in the genus *Flaveria*. The integrated approach combining genomic sequencing, phylogenetic analysis, and transcript abundance profiling offers valuable insights into the molecular mechanisms underlying this complex trait. However, there are many mistakes, errors or unclear points in the manuscript. Some concerns/suggestions below need to be addressed to further improve the study.

1. Lines 124-125: Inferring that the other four *Flaveria* species and sunflower share the WGD2 event based solely on Ftri may be biased, considering the substantial genomic disparities observed between Frob and Ftri. Therefore, it is crucial to present additional data that supports this inference and provides a more robust basis for the proposed relationship.
2. Lines 194-195: The reference to "Fig. 2c" does not adequately demonstrate that CA has extra copies in C4 species, and "Extended Data Fig. 3" does not clearly show that either CA, PEPC, and PEPC-k have extra copies in C4 species. The authors should provide more direct evidence or a more detailed explanation for their claims.
3. Lines 89, 119, 261: What are the abbreviations of CREs, mya, ERF? Please use the full names the first time they appear and check for other abbreviations throughout the text.
4. Lines 260-261 How many known motifs were used in the analysis and where are these motifs from? The rationale for studying them cannot simply be based on the identification of three out of nine known CREs in C4 species. The presence of ERF CREs in C3 species is not demonstrated in the text, and it is unknown whether ERF CREs are specific to C4 species or if their existence affects C4 photosynthesis. The enrichment score should also be provided in Fig 3a.
5. Line 280: "Extended Data Fig. 3" does not depict the chromatin accessibility of CA1 and PEPC-k1. Please recheck the numbering sequence of the Extended Data Figures in the manuscript.
6. Line 282: there is some missing word after "previous".
7. Lines 277-278: The three copies of the FtriPEPC1 gene have identical sequences and, as described in lines 213-214, share the same sequences from approximately 2500 base pairs upstream to about 2000 base pairs downstream. Thus, assuming the preservation of multi-mapped reads, their RNA-Seq and ATAC-Seq results would necessarily be the same, which does not elucidate any biological issue.
8. Lines 327-328: How to compare ortholog transcript and protein abundances from different species? How are the data normalized for comparison? It's unclear from the text.
9. Line 376: The authors state that "The codon usage patterns of C4 genes were conserved," yet in line 383, the term used is "all protein-coding genes." Is there a conflict here?
10. Lines 391-395: The text attempts to illustrate nuanced codon usage variations between different species, but the consequences of these differences are not demonstrated in the manuscript.
11. Line 139: "*Flaveria*" should be italicized, and the formatting throughout the text should be checked for consistency.
12. Figure 1a: The time axis should not have negative values.
13. Figure 2d: The colinearity plot between genes should be presented in the same manner as Figure 2b, without manual connections.
14. The provided link (https://db.cngb.org/codeplot/datasets/public_dataset?id=flaveria) doesn't include gene annotation GTF files.

Reviewer #3

(Remarks to the Author)

The convergent evolution of C4 photosynthesis is an important question with clear applied implications in plant science. The authors generated five chromosome-scale assemblies of *Flaveria* species that represent the spectrum of C3-C4 photosynthetic biology. The authors find through analysis of RNA-seq and ATAC-seq data that cis-regulatory evolution played a major role in the acquisition of C4 photosynthesis in *F. trinervia*. The chromosome-assemblies appear to be of high-quality and will be useful to the plant science and photosynthesis community. The manuscript is well written, and the figures are nicely prepared. While I do think that this study will be of general interest, I have concerns regarding the ATAC-seq data quality and analysis that will require attention before publication can be justified. More details below.

Major comments:

I could not find how ATAC-seq replicates were assessed for reproducibility in the methods or main text. Therefore, I cannot evaluate the quality of these data (however, I do have concerns regarding data quality given the strange ATAC profile of Figure 3b, low signal/noise ratios observed in Figure 3d, and extremely low unique mapping rates). At a minimum, peaks should be called for each replicate, peaks merged by taking the union across all replicates, read counts scored against all peaks across replicates, and correlations taken between replicates. The fraction of reads in reproducible peaks (FRiPs) should also be reported for each sample. Although the methods section does not specify how correlations were estimated, please be aware that using counts taken from uniform bins across the genome represents a zero-inflated distribution and will result in highly inflated correlation estimates. Ideally, the authors should perform IDR analysis (from the ENCODE project) to assess reproducibility and keep peaks reliably identified in multiple samples. Note, please pay special attention to the parameters for peak calling with ATAC-seq data. More on this below.

Line 732: The parameters used for THS peak calling are inappropriate. I see this mistake all the time. MACS2 is a general usage peak caller that creates pseudofragments based on PE data when used in PE-mode (as is done here), which would be appropriate for ChIP-seq-like data. The result is that MACS2 is using the pile-up of the inferred fragment. For ATAC-seq data, you want the pile-up to happen on the integration sites – the ends of the fragments, not the center. As a result of using these parameters, the authors are likely calling THS that are in between accessible chromatin, which will obviously create artefacts/issues for downstream interpretation. Moreover, many of these THS peaks will be nucleosomal (the opposite of accessible chromatin), because Tn5 can insert adapters in linkers flanking nucleosomes. This is the basis of nucleosomal banding patterns seen in ATAC-seq libraries from a TapeStation or Fragment Analyzer. The correct way to call peaks with ATAC-seq data is to isolate the integration site (5' end of reads at single-bp resolution) and create pseudofragments that are centered on the integration site (rather than in the center of the PE coordinates) using the `-shift` and `-extsize` arguments. The larger issue is that because some proportion of THS represent in-accessible closed chromatin, it's hard to know how much of the observed CRE enrichments are reflecting biology versus technical artefacts (especially considering the ERF TFs bind highly degenerate GC-rich motifs). This analysis needs to be redone correctly before it is possible to evaluate the biological implications of this experiment.

As mentioned above, the meta ATAC-seq profile from Figure 3b looks strange to me. In every species I've seen, the peak of accessible chromatin is upstream of the TSS, not within the gene body. This suggests to me incorrect analysis parameters, an artefact of data processing, or bad data quality. If the authors can show the data is of good quality (see my first comment), I would take a careful look at how reads were processed by deeptools. Typically, you would want to use the Tn5 integration sites (again, not the PE reads), to create bigwigs for such analysis. MACS2 can output the coverages of Tn5 integration sites as bedgraphs if you give it a single-bp resolution Tn5 insertion site.

Line 222: "... all three FtriPEPC1's were upregulated ..." – Although the authors note a significant difference between low and normal CO2 conditions, the differences appear trivial. Moreover, in the methods, the authors report using a P-value threshold of 0.05. Did the authors perform multiple test correction? If not, then the false-discovery rate will be incredible high and many detected differences are likely to be technical rather than biological in origin.

Minor points:

Line 100: "... genome size has gradually increased...".

Figure 3d,e: How are the tracks scaled? There are no axes or scale bars.

Figure 3c: How does the ERF enrichment in C4 genes compare the genome background across all ACRs?

Why does the GRN analysis include CREs 3kb upstream of TSS instead of leveraging THSs?

Reviewer #4

(Remarks to the Author)

In their manuscript, Lyu et al. conducted genome assembly for five *Flaveria* species, demonstrating superior quality compared to prior genome annotations. By harnessing this genomic data, the authors identified genes encoding C4 enzymes and transporters, discovering duplication events in C4 genes within C4 species. Subsequently, ATAC-seq analysis revealed a prevalence of ERF cis-regulatory elements and ERF TFs in *Ftri*. Lastly, comparing protein-to-mRNA

ratios (PTR) among these species, the study suggests that transcriptional regulation, rather than translational regulation, drives the protein levels of C4 genes in Ftri species. Notably, the study highlights the significant role of transcriptional regulation in C4 photosynthesis across Flaveria species. However, additional evidence is required to bolster the conclusions drawn by the authors.

p293 - p305, the manuscript lists the five most abundant TF families across the five Flaveria species. Besides ERF, other TF types, such as bHLH and MYB, are notably prevalent in Ftri. However, it's challenging to definitively establish the direct correlation between these TFs and C4 genes solely from the C4GRNs. Additional experimental assays should be devised to thoroughly investigate this relationship.

p689 - p707, the authors did not provide a clear explanation of the proteomics assay. Details such as the quantity of samples (e.g., ?g) and protein extraction buffer (e.g., ?mL) utilized for total protein extraction, as well as the number of technical or biological replicates employed, remain unspecified. The observation of low PTR for certain C4 proteins in Ftri, as depicted in figures 4c and 4d, led to the conclusion that transcriptional regulation significantly impacts C4 photosynthesis. However, it's crucial to acknowledge that other factors, such as epigenetic regulation or post-transcriptional/translational processes, could influence protein abundance. Moreover, it's pertinent to assess the functionality of these C4 enzymes in these species. Have the authors conducted additional experiments to further validate the significance of transcriptional regulation in Ftri?

Other minors in the manuscript:

In Figure 1 and the method section (pages 150 and 521), the authors conducted FISH analysis to confirm the chromosome numbers of Flaveria species. However, they did not specify which DNA probes were used for this assay.

In Figure 2c, the authors labeled PEPC1-4 for the first four clades but omitted labels for their sub-clades. Additionally, in Figure 2d, it would be beneficial to include gene IDs for PEPC1.1 to 1.3. Additionally, the labels of PEPC1.1 to 1.3 could be removed from Figure 2c.

In Figure 4c, it should be noted that the red dots and blue dots represent proteins, not genes. Dotted lines should be added to segregate high PTR and low PTR in the scatter plot. Furthermore, the labeling of some C4 proteins by arrows does not align with some low and high PTR, necessitating double-checking in the figure. For Figure 4d, asterisks indicating the p-values should be added to the figure.

The conclusion regarding the conserved codon usage patterns in the five Flaveria species seems to have minimal contribution. It may be more appropriate to move this information to another supplementary figure.

Reviewer #5

(Remarks to the Author)

Lyu and co-authors describe an extensive genomics, transcriptomics and proteomics analysis of five Flaveria species to study C3 to C4 photosynthesis evolution. Draft genomes for four Flaveria species were published before in the context of C4 evolution (Taniguchi, Y. Y. et al. (2021). *Plant Genome*, 14:e20095), though higher-quality genomes at pseudo-chromosome scale were acquired in this work. As also indicated in the discussion, the increased mRNA and protein levels of C4 enzymes is known and confirmed in this work. As described more elaborately in the comments below, I have some questions regarding the protein quantification values used and the comparison of them between species. This is important as the calculation of the proposed translational efficiency depends on these. It should be noted that methods such as ribosome profiling are important in such studying mRNA translation, for instance within the field translation efficiency is defined as ribosome footprints divided by mRNA-seq expression. There is thus quite a leap from mRNA to protein intensities to conclude on translational efficiency, which warrants caution with the statements made from these data.

Comments

- The authors also performed long-read sequencing before in Flaveria species and studied C4 evolution, which I believe is a necessary work to be considered in the introduction. Also here GRN were made and association with TF families. Paper: Lyu, A. et al. (2023). Evolution of gene regulatory network of C4 photosynthesis in Flaveria. *Plant Commun.*, 4(1), 100426.

- There are five Flaveria species sequenced, it could be of interest to specifically mention which are compared to the published genome assemblies (from line 132). It is written in the supplementary notes that a BLASTP E-value threshold of 0.001 was used to determine proteins to be common or not. This is I believe not a very stringent criterion, i.e. would it not be better so to set a high % identity of your protein sequence of your assembly? For instance if your BLASTP alignment only covers 50% of your protein but fulfilling your E value criterion – you consider this as a common protein?

- From line 185 it was described how high light induction was used to further verify the identified C4 enzymes. Within Supplementary note 5, that was used to describe the high expression of these in the C4 species, there was also a CO₂ treatment whose rationale is not really properly explained in the manuscript itself (although also included in Figure 2c). Could you not integrate the high light expression together with the CO₂ treatment here in a single plot, and explain these results better within the results section? You could describe the high light induction as a third criterion, i.e. (i) highest expression among paralogues, (ii) higher transcript abundance in C4 species vs. C3, and (iii) HL induction.

- Line 256 and onwards: There is a sudden strong focus on the role of ERF transcription factors in this chapter. The introduction does not mention ERFs, is there any know role of ERF in regulating C4 enzymes? Did the expression of important ERF in the GRNs also increase from C3 to C4 species?

- Line 327: This jumps straight in a result. It would be beneficial to know that what sort of proteomics set-up was performed here in the text. It is in fact a thorough quantitative analysis, with six replicates per species – leading up to species-specific DDA spectral libraries and Spectronaut DIA quantitative analysis. It would be of value to showcase your experimental set-up, potentially with a small graphic included.

- It is highly advisable to use a more absolute quantitative measure for protein levels. Instead of protein LFQ values, it would be better to use iBAQ quantification values – see “Global quantification of mammalian gene expression control” (Schwanhäusser et al.). Spectronaut 14 contains iBAQ quantification values as a feature. Protein LFQ values are definitely suited for relative protein quantification, i.e. the same protein across different conditions (treatments/conditions). But you are here in fact comparing the levels between proteins and species. Hence, comparison of iBAQ values (proteomics) to TPM values (RNA-seq) seems more sensible.

- I am unsure how the final DIA quantifications were obtained. It mentions in the methods “As a result, five peptide libraries were obtained. Finally, data-independent acquisition (DIA) was performed using Spectronaut (version 14.7, Biognosys, Zurich, Switzerland).”. Does this mean the data of each species was analysed separately with Spectronaut (with its respective spectral library) and later on quantitative protein matrices were merged? This is not an easy task, as you would have to do this based on orthologous relationships between proteins and perhaps define certain orthologous protein groups.

- Related to this, how exactly did you generate Figure S29 as you are dealing with species-specific spectral libraries? Also for instance, in Fig S31 we see Frob protein identifiers (right of the heatmap) with quantifications in non-Frob species. Hence, this gives me the feeling a single species library was used for all species? If you search every species separately with Spectronaut (which seems the most correct to me), you could compare iBAQ values of the respective orthologs.

- The data was submitted to PRIDE but no accession, citation and reviewer access is provided. With such a data-rich article it would be good to have an accessible datafile providing TPM and protein levels across species. I did see there was a figshare link but I had no access, I apologize if I overlooked something.

- Figure 4a: Heatmaps of z-scored quantitative values are displayed, it is not mentioned what these were in the legend. Is this TPM for RNA-seq and LFQ protein intensity for proteins? Please see my comment above on iBAQ values.

- Figure 4b: I strongly advise against such complex Venn diagrams, a more elegant solution could be UpSet plots, e.g. see the paper entitled ‘UpSet: Visualization of Intersecting Sets’ by Lex et al., there are many interactive tools and R packages to do this.

Minor comments

Line 80: Missing point.

Line 89: CRE not spelled out within text at first mention.

Figure 1: Panel c text font is too small to be readable.

Supplementary Note 4: “exhibited no annotations sequences in the”: simply replace to were not annotated?

Supplementary Note 5: I deduce from the graphs that TPM values were used to estimate transcript abundance, which tool did you use to map/quantify? The methods do not state this. The figure quality of the alignments is highly variable in these Supplementary notes. The meaning of the statistics (*) that are not explained in the legend – these is differential expression according CO2 treatment as in Figure 2 I assume?

Line 265: 265 (ATAC-seq) experiments, also Tn5 hypersensitive site might need some more background for readers.

Figure 3d and e: Currently many IGV genome views are provided, from genome to chromosome scale to million to scale (panel d) and for three genes of interest (e). Instead of these many visualizations, I feel like it would be better to dedicate more space simply to the three genome views of the individual genes – preferably with their tracks at the same y-axis scale.

Figure 3h: This means the CRE of the C4 enzymes were connected to all possible matching TFs with ERFs in pink? The legend could give some more detailed explanation.

Overall many typos in the supplementary notes, e.g. ‘vacumm-dried’, ‘CAN’ instead of ‘ACN’,...

Version 1:

Reviewer comments:

Reviewer #1

(Remarks to the Author)

Overall, I find the revised manuscript to be much improved. The conclusions are clearer and have sufficient support from the data. Likewise, additional figures and modifications to existing figures make the results easier to understand to a broader range of readers. Specifically, my concerns regarding the role of gene duplication and Ks analysis have been dealt with, as have my concerns about claims regarding the effects of gene duplication. Revisions to results associated with light response in C4 and C3-C4 species is also satisfactory. The results for ATACseq analysis are much clearer as are methods relating to TF analysis and gene expression. In general, I believe that this version is suitable for publication in Nature Communications

pending some basic editing for grammar.

"The reason we do not delve deeply into C3-C4 species is that we have found that C3-C4 species may not necessarily be transitional between C3 and C4 species but rather alternative outcomes." I agree that viewing C3+C4 as a transitional state is problematic and that in most cases it more likely represents an alternative outcome to C4. I think now that my primary confusion emerged from the description of C3+C4 plants as intermediate but realize that "intermediate" is not necessarily equivalent to transitional. I appreciate the authors' well-considered reply to my comment in their response but I still feel that this could be made clearer in the introduction. Statements such as "facilitating the progression from C3-C4 intermediate to a full C4 state." And "Therefore, the transition from C4-like species to C4 ones is regarded as a fine-tuning process" still give the impression that extant C3+C4 lineages are potentially species in transition unless it is clearly stated otherwise. The authors state in their response: "The terms "C4 lineages" and "C4-like" are not synonymous. Within the *Flaveria* genus, *F. brownii* from Clade B is considered a C4-like species because approximately 12% of CO₂ is still fixed directly by Rubisco." I think my confusion arose from the fact while C4-like plants are described in the intro they are not included in this study.

Also, if the authors plan to conduct follow-up targeted studies on C3 + C4 species (or that further study is merited in these species) this could be indicated in the discussion.

Still some typos:

Line 83: cis-regulate should be cis-regulatory

Line 361: "Precited" should be "predicted"

Line 376 - 377 : "exhibited a peak at the upstream of gene transcription" can remove "at the"

Reviewer #2

(Remarks to the Author)

Thank you to the authors for thoroughly addressing my comments. However, I noticed an additional issue: the genome annotation GFF3 file is not available at https://db.cngb.org/codeplot/datasets/public_dataset?id=flaveria

Reviewer #3

(Remarks to the Author)

While the authors have addressed some of my concerns, the revised manuscript contains language that reveals a worrying lack of expertise in this field. For example, in line 373, the authors state "... the Tn5 transposase enzyme preferentially binds nucleosomes with accessible DNA...". This is patently incorrect. At the concentration used for most ATAC-seq experiments, Tn5 has an incredibly strong preference for nucleosome-free regions, and cannot bind nucleosomes or nucleosome-bound DNA. An additional concern is that the results are seemingly unchanged following the reanalysis of a totally new set of peaks. To confirm that the results in the present manuscript are from the new peak set, I ask that the authors contrast motif enrichment scores in (1) all ACRs and (2) ACRs near C4 genes compared to background/control sequences (non ACRs) for both the original peak set and the updated peak set for all tested motifs (highlighting the scores for ERF family TFs). Please indicate how the background/control sequences were selected, as well as their composition (GC content and genomic distribution) and include this information in the methods sections. For motifs that are enriched in ACRs close to C4 genes, the background for this analysis should be the remaining set of ACRs (non-C4 gene ACRs). Additionally, I could not find a description of how the enrichment scores were calculated in extended data fig 8 in the revised manuscript. What were the background sequences used to determine enrichment? ERF motifs are pretty degenerate (GGC repeats) and frequent in the genome by chance. Please include this information in the methods section. In general, background sequences should be matched by GC content, genome distribution (i.e. same fraction of intergenic, promoter, genic overlap as ACRs), and by length (same length distribution as ACRs).

Related to the above comments. I could not understand the statistical rationale for enrichment tests in the authors response (sounds like a Monte Carlo permutation test, but the way the description is worded lacks clarity and conciseness). The authors move between descriptions of motifs and ACRs as the focus of the test? It doesn't make any sense. The choice of the background is not justified and was not what I asked for (the specific enrichment test requested was ACRs near C4 genes versus the total set of ACRs). Something like Fisher's exact test or a Chi-square test would be sufficient for enrichment tests, provided the background/control regions are well reasoned. Also, how is FDR calculated? Which specific method? I did not see this information in the response or in the methods of the revised manuscript. The apparent lack of rigor is a concern. Although, this could be alleviated by more comprehensive reporting in the methods.

I wish to note that the addition of the EMS and transient transcription assays is a strength. However, I do wonder why these results weren't included in the original submission, given the quick turnaround time of the revised manuscript (seems unlikely that the review was the prompt for these new experiments). I hope that the computational analyses were not cherry-picked to support the experimental results. To enable reanalysis of the ATAC-seq data by other laboratories, please upload the raw fastq files from the ATAC-seq replicates to a public repository (NCBI GEO), similar to the other data sets.

Minor comments:

1. The manuscript is still littered with grammatical and nonsensical errors. For example, line 377-380 "... from the two replicates Apply Irreducible Discovery Rate..." should probably read "... from the two replicates after applying Irreducible Discovery Rate (IDR) analysis..."

2. For the correlation analysis between replicates (using the IDR peaks thresholded at 0.05), of course the peaks are highly correlated, this is the point of IDR. Please report the correlation between replicates using the union of the raw peaks identified in rep 1 and rep 2.

Reviewer #4

(Remarks to the Author)

The authors have addressed these questions and provided more experimental information for the conclusions. Lines 184-187, The C4 species showed significantly higher enzyme activity and light responsiveness, which needs more explanation and discussion.

Some minor points need to be modified in the methods.

Line 810 and 811: "0.1 IPTG" should be "0.1 mM IPTG"? "ortholog in is Arabidopsis" should be "ortholog in Arabidopsis is" ? Some spaces need to be added in the text for lines 711-712, 811.

Reviewer #5

(Remarks to the Author)

I thank the authors for addressing all my comments. I appreciate the inclusion of ribosome footprinting data in the study of translational efficiency. Questions/concerns regarding the orthologous protein relationships and spectral library searches are now clarified, also the proteomics experiment is now properly introduced. Taken together, the methodology and drawn conclusions from these data seem appropriate to me.

Reviewer #6

(Remarks to the Author)

Lyu and co-authors addressed all my comments. I appreciate the extensive effort to incorporate the iBAQ quantification and especially the inclusion of ribosome footprinting data to strengthen the claims on translational efficiency. Also the proteomics analyses were now clarified. All together, this present an extensive work, and I believe this to be of general interest to the plant genomics community.

Version 2:

Reviewer comments:

Reviewer #3

(Remarks to the Author)

I commend the authors on their comprehensive response and have no further comments.

Point-by-Point Responses

We are grateful to the reviewers for their insightful comments and constructive suggestions on our manuscript entitled “A dominant role of transcriptional regulation during the evolution of C₄ photosynthesis in *Flaveria* species” (Manuscript ID: NCOMMS-24-16595). We have addressed all questions and concerns raised by the reviewers with new experiments, analyses, and explanations, and believe the manuscript to be significantly improved.

Detailed responses to reviewers’ comments:

Reviewer #1: Comments and point-by-point responses (Pages 2~19)

Reviewer #2: Comments and point-by-point responses (Pages 20~32)

Reviewer #3: Comments and point-by-point responses (Pages 33~43)

Reviewer #4: Comments and point-by-point responses (Pages 44~52)

Reviewer #5: Comments and point-by-point responses (Pages 53~70)

Reviewer #1:

The authors present 5 high quality genome assemblies for the genus *Flaveria* with associated transcriptomic and proteomic datasets. They go on to conduct a fairly standard set of genomic analyses exploring genome size and gene arrangement, TE expansion, enrichment of cis regulatory elements alongside transcriptomic and proteomic analysis that highlight differences in gene expression, gene regulation by ERF transcription factors, ratios of transcript and protein abundance, and codon usage between species. There is no doubt that the publication of these genomes represents a significant asset in *Flaveria* and to C₄ research in general. Furthermore, I do not see any problems with their methods or results. While I believe that some of their results will be of general interest to researchers studying plant genomics, I do not necessarily agree with their interpretations as presented in this manuscript. For example, the tandem duplication of PEPC in *F. trinervia* is interesting but I'm not sure it has anything to do with the transition to C₄ in this species. All three copies of PEPC1 exhibit elevated expression in *F. trinervia* which could be interpreted as evidence that the increase in expression occurred prior to and independent of its duplication. That this duplication is not observed in any of the C₃+C₄ species or other distantly related C₄ species render interpretation of its effect, independent of the simple increase in expression, highly speculative. In this case perhaps this could be reframed, if other papers have found evidence that WGD facilitated the transition to C₄ and they find evidence of extensive TE-mediated duplication that gene duplication may play a role but that the source of those duplications can vary (i.e. WGD or TE-mediated tandem duplication). However, the duplication of PEPC, on its own is not particularly compelling. Likewise, the increase in light response in several well-known C₄ genes does not seem consistent across the species presented here (see my more specific comments below), though in most cases there does appear to be a clear difference between the C₃ and C₄ species. Indeed, the majority of comparisons ultimately characterize differences between these two extremes with C₃+C₄ species contributing little to, and at times seemingly contradicting, the authors' overarching explanation of how C₄ has arisen in *Flaveria*. This is evident in the discussion where instead of referring to C₃, C₃+C₄, and C₄ species individually, the authors merely make distinctions between C₄ and "non-C₄" species, which presumably encompass all species except *F. trinervia*. Despite this, the narrative seems to be that C₄ was gradually acquired in four species following their divergence from *F. robusta*, though this is not explicitly stated in the discussion. To my eye, the explanation is necessarily much more complex and may simply not be captured by these results or even these type of broad -omics datasets.

Without specifically exploring the differences between C₃+C₄ species and *F. robusta* and *F. trinervia*, I don't see the added value of C₃ + C₄ species. I think these authors should explore these differences (between C₃+C₄ and others) more directly and/or make it clear that these species are not transitional between C₃ and C₄ but merely alternative outcomes.

Response: We thank the reviewer for his/her thoughtful and favorable review, as well as critical questions and suggestions for revision. In this revision, we conducted new analyses and experiments to address the reviewer's concerns and revised the whole manuscript according to the reviewer's suggestions. The new results are fully consistent with and strengthen our findings.

Regarding the tandem duplication of PEPC in *Ftri* (C₄), we acknowledge that its connection to C₄ transition may not be straightforward. In our revision, we have rephrased our interpretation to reflect that while the duplication is intriguing, its specific role in C₄ evolution remains speculative. We propose that it could have been a beneficial event facilitating the progression from C₃-C₄ intermediate to a full C₄ state, enriching our understanding and prompting consideration of the potential adaptive advantages conferred by such duplications during the later stages of C₄ pathway refinement. We have included such information in lines 466-469 on pages 22-23, which was also shown in the following: "While the duplication is intriguing, its specific role in C₄ evolution remains speculative. We propose that it could have been a beneficial event facilitating the progression from C₃-C₄ intermediate to a full C₄ state."

The reason we do not delve deeply into C₃-C₄ species is that we have found that C₃-C₄ species may not necessarily be transitional between C₃ and C₄ species but rather alternative outcomes. For example, we found that *F. ramosissima* (C₃-C₄) exhibits very different features in its C₄ gene regulatory network compared to *F. trinervia* (C₄), which may represent an alternative metabolic strategy to deal with NH₃⁺ between mesophyll cells and bundle sheath (Lyu et al., Doi: 10.1016/j.xplc.2022.100426). This finding makes us hesitant to compare C₃-C₄ species in detail here. In fact, we are planning to delve into the detailed C₃-C₄ metabolic features in future work. The detailed features and characteristics of C₃-C₄ species will be systematically reported in our future work. We believe that this focused approach allows us to provide a more comprehensive and reliable analysis of the evolutionary transitions in *Flaveria* species.

These type of broad multiomics datasets are not always suited to testing specific hypotheses of evolution as it pertains to a given gene or pathway. Instead, their strength often lies in providing a

(necessary) foundation for generating testable hypotheses and conducting more narrowly targeted functional studies. In focusing on primary differences between well-characterized C₄ genes between C₃ and C₄ species of *Flaveria*, I'm afraid that this paper contributes very little novel insight into the evolution of C₄ in this or other plant lineages. Other authors have previously explored the role of gene duplication, differences in expression between C₃ and C₄ species and the role of various transcription factors. The only central and possibly novel finding of the paper involves ratios of transcript and protein abundance. However, this finding is barely significant and the authors largely ignore substantial differences among C₃+C₄ species as well as between them and their C₃ and C₄ relatives. Without significant revision and clarification, I cannot recommend this paper for publication in Nature Communications. That being said, I think it is possible that there are further insights that could really improve the impact of this research. As a single example (though I'm sure there are others), exploring gene regulation and coexpression outside of the canonical C₄ pathway, might yield novel insights into downstream effects of duplication, modification and regulatory rewiring of C₄ genes. I want to reiterate that I think this dataset is impressive and will represent a significant and foundational resource to further study. I think the analyses conducted are relatively standard but sound. The primary weakness of this paper is that it does not seem to generate any novel insights or testable hypotheses for future research that would utilize the basic genomic resource presented here.

Response: We are grateful for the thoughtful and constructive comments on our manuscript. We agree with the reviewer's comment that using broad multi-omics datasets to test specific hypotheses of evolution for a given gene or pathway would be challenging without experimental validation. In this study, we aimed to provide a better understanding of the evolution of C₄ genes by integrating the analyses regarding cis-regulatory elements and gene regulatory networks. This approach helps elucidate the evolution of C₄ genes from both cis-regulatory and gene regulatory perspectives. We admit the limitations of our current work. To provide stronger evidence, we conducted additional experiments, including electrophoretic mobility-shift assay (EMSA) and transient expression. The new results (**Figure R1**) confirmed the proposed transcript regulation of ERF transcription factors to C₄ genes, thus suggesting that ERF TFs may play a role in the evolution of C₄ genes. (Page 20, lines 416-428, Result Section). The details are as following:

We used an electrophoretic mobility-shift assay (EMSA) to verify the predicted regulation of C₄ genes by ERF TFs in the C₄ species *Ftri* (**Figure R1**). Seven ERF TFs that were predicted to regulate *FtriCA1*, *PEPC1.1* and *PEPC-k1.1* were used for experiment. Cognate ERF TF binding site

(TFBS) within 2k bp upstream of their (*CA1*, *PEPC1.1* or *PEPC-k1.1*) start codon was used to perform EMSA. Our EMSA experiments verified the binding of ERF12 to the promoter of *FtriPEPC1.1* and *PEPC-k1.1*, the binding of ERF61, ERF51 and ERF1 to the promoter of *CA1.1* (**Figure R1A**). We then performed transient transcription assay to further verify the regulation of ERF12 and ERF61 to *C₄* genes. We used dual-luciferase reporter plasmids containing the firefly luciferase (*LUC*) gene driven by *CA1.1* promoter (200 bp from the start codon) and *PEPC1.1* promoter (250 bp from the start codon) and the Renilla luciferase (*REN*) gene driven by the constitutive 35S promoter in the analysis (**Figure R1B**). Results showed that ERF12 and ERF61 displayed significantly higher LUC/REN ratio compared with Flag tag (**Figure R1C, D**), suggesting the activation of ERF12 and ERF61 on *CA1.1* and *PEPC1.1*.

Figure R1: Identifying *cis*-regulatory elements and TFs that regulate *C₄* genes. (A) Electrophoretic mobility shift assay (EMSA) was performed with ERF TFs and cy5-labeled partial DNA sequence (probe) of their regulated *C₄* genes' promoter from *Ftri*. Labeled probes were incubated with GST-TF-10xHis protein. GST represents GST-10xHis protein without TF. For competition analysis, the binding reaction was performed

with addition of 200-fold of corresponding unlabeled probe. Bands corresponding to DNA-protein complexes (shift) or free probes are indicated by arrows. (B) Structure of reporter and effector plasmids. For the reporter constructs, 35S promoter, C₄ gene (*CA1.1* or *PEPC1.1*) promoter, firefly luciferase (*LUC*) and Renilla luciferase (*REN*) are indicated. For the effector construct, ERF TF (ERF12 or ERF61) was driven by the 35S promoter. (C) and (D) ERF12 and ERF61 activates the expression of *CA1.1* and *PEPC1.1* (Significance was calculated with t-test, n=3). Leaf epidermal cells of *Nicotiana benthamiana* were transfected with infiltration buffer (Mock), reporter DNA (*pCA1.1* or *pPEPC1.1*) with *Flag*, reporter DNA with *ERF-Flag* (ERF12 or ERF61). This figure has been included in the revised manuscript as Figure 5f-i.

Regarding the comments on the generation of testable hypotheses, we think that many of the observations from these analyses would provide the basis for the further following up studies. For examples:

- (1) We observed a dramatic change in the protein to transcript ratio (PTR), with C₄ species exhibiting a lower PTR. This reduction can be attributed to various factors. While we found that the translation efficiency of C₄ genes is comparable during evolution, other factors such as epigenetic modifications and protein degradation rates are worthy of further investigation.
- (2) We propose that both the ERF cis-regulatory network and ERF transcription factors are associated with the evolution of C₄ photosynthesis. Although we have provided evidence based on *in vitro* experiments to verify the regulation of ERF transcription factors on C₄ genes in *F. trinervia* (C₄), further studies are needed to explore the biological functions of ERF TFs in C₄ photosynthesis. Given that ERF TFs are well-known for their roles in stress responses, studying their impact on C₄ photosynthesis will provide insights into the molecular mechanisms of C₄ evolution during adaptation to environmental stress conditions.

These questions are currently the focus of our ongoing research following this genome analysis study. We believe that our study not only provides a rich resource to support *Flaveria* research but also generates a suite of hypotheses for future investigations.

Detailed review:

1. Are C₄ lineages the same as “C₄-like?”

Response: The terms “C₄ lineages” and “C₄-like” are not synonymous. Within the *Flaveria* genus, *F. brownii* from Clade B is considered a C₄-like species because approximately 12% of CO₂ is still fixed directly by Rubisco. However, a full C₄ pathway was performed in *F. brownii*. In contrast, full C₄ species, such as those found in Clade A, exhibit a complete C₄ photosynthetic cycle where CO₂ fixation by Rubisco is minimized. Therefore, researchers have indicated that Clade B represents an independent lineage of C₄ evolution distinct from Clade A.

2. Line 98 - Is Ftri C₄ or C₄-like?

Response: Ftri is a C₄ species. The photosynthetic type of species when mentioned was given after the name in brackets in the revised manuscript.

3. Fig 1 – Ks peaks for WGD2 do not appear to align. Could this be explained by differences in evolutionary rates? You could easily use KsRates (or just compare Ka/Ks) to test this...

Response: We appreciate reviewer for bringing up this concern. Accordingly, we have conducted a thorough re-analysis of the Ka/Ks ratios using a stricter threshold for both paralog and ortholog identification. In addition to an E-value threshold of 0.001. We now require a minimum protein identity of 60%. The new results reveal that the WGD2 peaks for Ftri and Hann are indeed well-aligned (see **Figure R2** below).

Figure R2: Ks between different species and within species. Ks of Frob vs Hann and Ftri vs Hann are shown to represent a speciation between *Flaveria* and Hann. Ks of Lsat vs Hann, Frob vs Lsat are shown to represent the speciation of Asteraceae and Lsat, and Ks of Hann vs. Atha, Lsat vs Atha are shown to represent the speciation of Asterids II with Atha. Ks of paralogs in Ftri and Hann are shown representing a whole genome duplication event (WGD). Ftri and Hann shared the WGD2 indicated as the second peak in orange “WGD2”. The first peak for Ks curves for either Ftri or Hann represents tandem duplication found in each species. This figure has been included in the revised manuscript as Figure 1d.

Furthermore, we have included the Ks distributions for all *Flaveria* species, with the exception of Flin, whose WGD2 peak is less pronounced. Interestingly, the remaining four *Flaveria* species exhibit a consistent peak at the WGD2 event. Notably, lettuce, which diverged from Hann prior to WGD2, does not display a peak at this event (**Figure R3**).

Figure R3: Distribution of Ks values for paralogous gene pairs among different species. Figure presents the distribution of Ks for paralogous gene pairs within and five *Flaveria* species, lettuce, and sunflower. Red star indicates the WGD2 event and blue star indicates the WGT1 event. (Abbreviations: WGD2: whole genome duplication 2, WGT1: whole genome triplication 1). This figure has been included in the revised manuscript as Extended Data Fig.2.

To address the potential impact of evolutionary rate differences, we have also compared the Ka/Ks ratios between *Flaveria* species and both sunflower and lettuce. Our results showed that there is no statistically significant variation in Ka/Ks ratios across these species (**Figure R4**).

Figure R4: Comparative Analysis of Ka/Ks Ratios among *Flaveria* Species, sunflower, and lettuce. (a) Frequency distribution of the Ka/Ks ratios for paralogous gene pairs across *Flaveria* species, sunflower, and lettuce. (b) P-values resulting from Wilcoxon rank sum test conducted on pair-wise comparisons of Ka/Ks ratios between the *Flaveria* species, sunflower, and lettuce.

4. Lines 167-170: did transcripts have to meet both criteria what if a gene was most highly expressed but did not show increased abundance to C₃ species? Which C₃ species were used? Also, this could probably go in the methods.

Response: To clarify, in our selection process, genes were indeed required to fulfill both criteria: being most highly expressed in leaf tissue in C₄ species and displaying increased abundance compared to the C₃ species. We have not encountered such known C₄ genes that exhibit relative high expression in C₄ species, whereas did not show increased abundance in C₄ than in C₃ species. This is reasonable, as the identification of C₄ enzymes and transports was based on highly expressed in C₄ species compared to

C₃ in leaf tissues. According to Reviewer's suggestion, we have relocated these details to the Methods section of the revised manuscript.

5. Lines 187-192: this section makes it sound like there is a gradual increase in light responsiveness across all (or at least many) C₄ genes. I do not see this in Extended Data Fig 2. While the difference between Frob and Ftri is clear in most comparisons, I think that this section overstates the significance of this trend and makes it sound like it is a smooth transition from C₃ -> C₃+C₄ -> C₄. Some examples:

- PEPC1 – is significantly elevated in at least one comparison in all species. Though the magnitude/abundance does increase in Ftri, it doesn't in the C₃+C₄ species.
- NADP-MDH shows no clear trend to my eye and is barely significant in Ftri comparisons.
- PPDK is significantly higher at 4h in Frob, higher at 0h in two C₃+C₄ species and not significant in Ftri.

Response: We appreciate reviewer for pointing out this issue. In the revised manuscript, we described the light induction of tested seven enzymes in details in lines 176-185 on pages 8-9, based on which, we make the conclusion that light induction of C₄ enzymes was most obvious in C₄ species and largely intermediate in the C₃-C₄ species, which is also shown in the following:

PEPC and *PPDK* showed light induction in both C₃ and C₄ species, but the timing varied: 2 hours upon illumination in C₄ species Ftri, and 4 hours after illumination in the C₃ species Frob. Light induction of *CAI* was observed in C₄ (Ftri) and type II C₃-C₄ species (Fram and Flin) but not in Frob (C₃) and type I C₃-C₄ species (Fson). Light induction of *PDPC-k* and *NADP-ME* was observed in Fram (C₃-C₄) and Ftri (C₄), whereas, that of *NADP-MDH* and *PPDK-RP* was only observed in Ftri. Therefore, the light induction of C₄ enzymes was most obvious in C₄ species and largely intermediate in the C₃-C₄ species (Extended Data Fig. 3a), revealing a gradual gain of light responsiveness during C₄ evolution. (Pages 8-9, lines 176-185)

6. The duplicated copy of PEPC is interesting at first glance. However, it seems strange that you do not see this duplication in any of the C₃-C₄ species, since they also share elevated expression of C₄ PEPC. Isn't it possible that elevated expression (and the transition to C₄) preceded this duplication, especially since increases in copy number are not evident in other C₄ species?

Response: We are grateful for your thoughtful feedback regarding this point. Indeed, it is plausible that the elevation in PEPC expression levels may precede the gene duplication event in the *Flaveria* genus. We found the elevated expression of PEPC genes in C₃-C₄ species, especially in Fram, this may initially have been achieved through regulatory modifications. This initial boost in expression could have provided a selective advantage, facilitating further specialization and optimization of C₄ metabolism.

To further explore these hypotheses, future studies could involve more comprehensive comparative genomics across a broader range of C₃, C₃-C₄ intermediate, and C₄ species, coupled with functional analyses of the duplicated PEPC genes in *Flaveria* species. This would help elucidate the precise sequence of events leading to the emergence and fine-tuning of C₄ traits, including the role of gene duplication in this complex evolutionary transition.

7. Fig 2 – some stylistic comments: the background for the chloroplast in a. is distracting and makes the text (especially arrows) difficult to read. This would read much better with a flat background color? Lines in d. look roughly drawn. They would be easier to read if they were straight or at least smoothed (similar to the synteny plot in b.).

Response: We appreciate reviewer's comment. Accordingly, the background for chloroplast in Fig.2a is revised to a light green color, and the Fig.2b is revised to smoothed lines.

8. Up to editor- should ERF be spelled out the first time?

Response: We appreciate the reviewer's comment. The full name of ERF (Ethylene Response Factor) is provided the first time it is mentioned in the manuscript.

9. Line 286-287 why were these TFs filtered out? More importantly, what proportion of TFs have predicted cognates in this species (or are you using predictions from Arabidopsis?). It seems that this approach might filter out important TFs without a predicted binding site which could significantly bias your results toward TFs based on functional screening in a distantly related species. Likewise, it might

lead you to exclude potentially important ones that have no predicted binding site, which are presumably numerous.

Response: We appreciate reviewer pointing out this issue. To improve the accuracy of the predicted transcription factors (TFs) and their targeted genes, we filtered out TFs without cognate TF binding sites within a TF family predicted using PlantPAN (V3.0), which encompasses 17,230 TFs and 4,703 matrices of TF binding sites across 78 plant species, providing a comprehensive and reliable resource.

10. This may just be my ignorance about ATACseq, but I'm a little confused by Fig 3 b. Is this showing THS within the coding sequences of genes? i.e. downstream of TSS? That seems opposite of what I would expect.

Response: We appreciate reviewer pointing out this issue. We re-analyze ATAC-seq data and redraw the profiles of average ATAC-seq peak intensity for all annotated genes near the transcription start site (TSS). The new results show that the ATAC-seq reads show peaks in the upstream of TSS (**Figure R5**).

Figure R5: Predicted *cis*-regulatory elements in the *C₄* species *Ftri* using ATAC-seq (a) The plots show profiles of average ATAC-seq peak intensity from two independent biological replications for all annotated genes near the transcription start site (TSS). The x-axis represents the distance from the TSS, y-axis depicts the normalized read count per million mapped reads. (b) The pie chart demonstrates the distribution of ATAC-seq peaks in relation to gene positions. (c) Genome browser tracks displaying accessible chromatin regions and transcript abundances (RNA-seq) of examples of photosynthesis genes and *C₄* genes. (d) Enriched *cis*-regulatory elements (CREs) in three types of accessible chromatin regions (ACR-CREs), *i.e.*, genic (gACR-CREs: overlapping a gene), upstream (upACR-CREs: within 3kb upstream of the start codon of a gene) and downstream (down ACRs-CRES: within 3kb downstream of the stop codon of a gene). (e) Enriched ACR-CREs associated with photosynthetic genes and photorespiratory genes. (Abbreviations: ATAC-seq: transposase-

accessible chromatin using sequencing; ACR: accessible chromatin regions; CREs: *cis*-regulatory elements). Several panels of this figure have been included in the Figure 5 of the revised manuscript.

11. Based on your trends in TF abundance (Fig 3 f,g) I can sort of understand why you chose to focus on ERFs. However, the abundance plots seem to be more correlated to species than the presence of C₄, not just in ERFs but other TFs as well. Instead of a gradual increase in ERF (or any other family) from C₃ to C₄, Ftri (C₄) is almost always highest, followed by Fson (Type-I) and Frob(C₃) with the other two C₃+C₄ lineages considerably lower. This seems to conflict with the species phylogenetic relatedness as well as their degree of C₄. On a related note, I feel like almost as strong an argument could be made for focusing on C2H2 TFs as ERF. While there is no doubt that TFs play an important role in the evolution of C₄ lineages, I think the role of expansion of large TF families is likely more nuanced than what is presented here.

Response: We appreciate the reviewer's comments. The reviewer correctly points out that C₃-C₄ species with closer phylogenetic distance to C₄ species do not show more similar TF profiles that regulate C₄ genes. In our earlier study, we reported that Fram exhibited a more divergent C₄ gene regulatory network (GRN) compared to Ftri, and when combined with metabolic flux analysis, we proposed that Fram represents a parallel metabolic pathway to C₄ rather than a transitional state (Lyu et al., DOI: 10.1016/j.xplc.2022.100426). The results presented here also indicate that Flin displays a distinct C₄ GRN compared to C₄, and considering Flin's high drought resistance, this might suggest that Flin represents another parallel evolutionary pathway to C₄.

The reviewer is correct that the number of C2H2 TFs regulating C₄ genes has also increased in Ftri. However, we did not observe an increase in the *cis*-regulatory elements of C2H2 in the C₄ genes of Ftri (C₄). Therefore, we did not extend our analysis to the C2H2 TF family in our present study.

12. Line 347 - "significantly lower" hmm... I suppose this all depends on your definition of "significant." P=0.003 is significant by most standards but P=0.02 may not be. While these cutoffs are entirely arbitrary, I think that at the very least you need to explicitly state your significance threshold

in this statement (e.g. “C₄ genes showed significantly lower PTRs ($P < 0.05$)...”) and provide justification for this cutoff, considering this is one of the main findings of your research.

Response: We appreciate for reviewer’s suggestions. We described in the figure legend of Fig.4 that the statistical significance was determined by a one-way ANOVA procedure followed by a two-tailed Wilcoxon rank sum test and P values have been adjusted using the Benjamini-Hochberg (BH) method. In response to the reviewer’s comment, we have added the details of the statistical analysis and P values in the revised manuscript in result section as follows: “C₄ genes showed significantly lower PTRs in the C₄ species than their orthologs did in the C₃ and C₃-C₄ species (two-tailed Wilcoxon rank sum test, BH-adjusted $P < 0.05$)” (Page 13, line 270).

13. Lines 350-352 – I’m not sure I understand what you’re saying here maybe you mean “... which can be attributed to increased transcriptional abundance”?

Response: We appreciate the reviewer pointing out this unclear description. The sentence has been corrected to clarify the contribution of transcriptional abundance to protein levels: “Therefore, during the evolution of C₄ photosynthesis, C₄ species acquired elevated protein levels for C₄ genes, which might be mainly attributed to increased transcriptional abundances.” (Page 13, lines 273-274).

14. Fig 4 d –Why do P-values appear highly significant for the random resampling? Also, while you mention significant difference in photorespiration the difference between Ftr and Fram also appears significant in photosynthesis related genes. These p-values are barely significant and many of the random comparisons seem highly significant including one between Ftr and Fram. I’m not convinced, based on the results as presented in this figure, that there is evidence of a significant difference in PTR as stated here.

Response: We appreciate the reviewer’s comment on this point. Our analysis revealed that Fram generally exhibited higher PTRs for all genes, which may be specific to Fram for random resampling. This is the reason for the significant differences observed between Fram and the other four species. However, there is no significant difference among the remaining four species.

We observed lower PTRs for C₄ genes in Ftri (C₄) compared to the other four species. Although Fram showed higher PTRs for all genes, the difference in PTRs of C₄ genes between Fram and Ftri may be due to the systematically higher PTR in Fram. Nevertheless, the difference between Ftri and the other three species is consistent and not likely due to chance (**Figure R6**).

Figure R6: The distribution of PTR across five *Flaveria* species. The protein-to-mRNA ratio (PTR) distribution of genes across the five *Flaveria* species. High PTR and low PTR genes are defined as genes with PTR higher than the mean plus one standard deviation (SD) and with PTR values lower than the mean minus one SD respectively. Enriched function of conserved high PTR and low PTR genes across the five *Flaveria* species and their enriched function were shown. This figure has been included in the Figure 3b of the revised manuscript.

15. Lines 376 – 377 – why is this notable? Is there some reason you would expect codon bias to change in C₄ species? Is there some other example of this in other C₄ lineages? I’m just not seeing the link between codon usage and the transition to C₄. Without more justification this could be greatly reduced or moved to the supplement. It seems that you are just verifying that this bias is highly conserved between plant lineages, which does not appear to be a novel finding.

Response: We appreciate the reviewer's suggestion. We have removed the comparison of codon bias between different species, as it did not provide a clear link to the transition to C₄ photosynthesis and lacked sufficient justification. Instead, we focused on using ribosome profiling (Ribo-seq) for Frob

(C₃) and Ftri (C₄) to investigate changes in translation efficiency during evolution. As Ribo-seq captures the positions of ribosomes on mRNAs, it provides a direct measure of translational activity. Our data show that the translation efficiency of C₄ genes is comparable between C₃ and C₄ species (**Figure R7**). This finding further supports the conclusion that the increased protein abundances of C₄ genes in Ftri (C₄) are predominantly attributed to increased transcriptional abundances. (Pages 15-16, lines 296-344)

Figure R7. Translation efficiency comparison between Frob and Ftri. (a) GC₃ comparisons of C₄ genes across five *Flaveria* species. As FramPEPC-*k1* is missing, its paralog (FramPEPC-*k2*, FramNA03444, see Extended Data Fig. 4) was utilized as a comparison for Fram in this context. (b) Abundances of ribosome protected fragment (RPF) of C₄ genes. (c) and (d) Distribution of RPF-to-RNA ratio and protein-to-RPF ratio.

(e) Scatter plots of RPF versus RNA of Frob and Ftri. Low/high/moderate RPF-to-RNA genes are labeled in pink/light blue/grey. (f) Scatter plots of protein versus RPF of Frob and Ftri. Low/high/moderate protein-to-RPF genes were labeled in pink/light blue/grey. Low protein-to-RPF C₄ genes were indicated with red arrows. C₄ and photorespiratory genes are labeled in red and blue, and Pearson correlation (PC) between RPF vs. RNA, protein vs. RPF is shown in the parentheses on top of each panel in (e) and (f). (g) and (h) RPF-to-RNA ratio and protein-RPF ratio for C₄ genes and photorespiratory genes. Statistical significance. This figure has been included in the Figure 4 of the revised manuscript.

16. Extended Data Fig 2 – I assume the Fram PEPC-k is the paralogous copy mentioned in Fig 5. If so it should be stated here

Response: Reviewer's suggestion is adopted.

17. Extended Data Fig 3 – ERFs appear to be extremely common throughout the Ftri genome. While I'm not exactly sure how to interpret this plot in any quantitative sense, They appear to dominant in both ACR categories as well as light responsive genes. I think that the expansion of ERF TFs is potentially an interesting difference between Ftri and the other species but I'm not sure how this links them to the transition to C₄.

Response: We appreciate the reviewer's observation regarding the widespread distribution of ERFs in the Ftri genome. Our hypothesis is that the widespread distribution of ERF ACRs in the genome increases the likelihood of acquiring these ERF *cis*-elements via transposons, particularly under stress conditions. Given that C₄ photosynthesis evolved under environmental stress conditions, such as drought, low CO₂, and high light, it is plausible that genes gained ERF *cis*-elements more frequently, leading to the regulation of C₄ genes by ERF TFs during evolution. This hypothesis was discussed in the Discussion section in the revised manuscript (Pages 24, lines 501-504).

18. Lines 447-462 – As this result seems to confirm previous findings and no difference is found between C₃ and C₄ species I'm not sure it merits its own paragraph in the discussion (or its own results section for that matter).

Response: According to Reviewer's suggestion, the paragraph was deleted from Discussion section.

Minor edits:

Line 46: 'both' is unnecessary.

Response: We have corrected accordingly.

Line 220: "franking" should be "flanking."

Response: We have corrected accordingly.

Fig 3d "Cut" or "CUT"?

Response: The termed has been corrected to peak.

Line 162 "gene" should be "genes."

Response: We have corrected accordingly.

Fig 5 – When you say: "FramPEPC-k2, FramNA03444, see Extended Data Fig. 3" I assume you mean Extended Data Fig. 4.

Response: The reviewer is correct and we corrected this mistake accordingly.

Reviewer #2:

The study by Lyu et al. provides a comprehensive analysis of the genomic and transcriptional changes associated with the evolution of C₄ photosynthesis in the genus *Flaveria*. The integrated approach combining genomic sequencing, phylogenetic analysis, and transcript abundance profiling offers valuable insights into the molecular mechanisms underlying this complex trait. However, there are many mistakes, errors or unclear points in the manuscript. Some concerns/suggestions below need to be addressed to further improve the study.

Response: We thank the reviewer for his/her thoughtful and favorable review, as well as critical questions and suggestions for revision. In this revision, we conducted substantial new analyses and experiments and analyses to address the reviewer's concerns. The new results are fully consistent with and strengthen our findings. Please check the specific response to each comment below.

1. Lines 124-125: Inferring that the other four *Flaveria* species and sunflower share the WGD2 event based solely on Ftri may be biased, considering the substantial genomic disparities observed between Frob and Ftri. Therefore, it is crucial to present additional data that supports this inference and provides a more robust basis for the proposed relationship.

Response: We appreciate for reviewer's suggestion. In response to the reviewer's concern about inferring the shared WGD2 event among the four *Flaveria* species and sunflower based solely on Ftri, we have conducted additional analyses and revised our manuscript accordingly (Extended Data Fig.2 in the revised manuscript).

1). We have now included the distribution of paralogous gene pairs' Ks values between all five *Flaveria* species and both sunflower (Hann) and lettuce (Lsat) (**Figure R1**). This comparison reveals that all five *Flaveria* species exhibit similar peaks as sunflower (Hann) at the WGD2 peak, while lettuce (Lsat) does not show this peak. Instead, lettuce displays an ancient peak corresponding to the whole-genome triplication 1 (WGT1) event in Asterids II, which has been estimated to have occurred around 38-50 million years ago (mya).

2) On the other hand, we extended our synteny analysis by comparing the genomes of Frob and Ftri with those of sunflower. Our results demonstrated high levels of chromosomal conservation between the *Flaveria* species and sunflower, with 12 chromosomes in each *Flaveria* species exhibiting clear

correspondence to the 12 chromosomes in sunflower (**Figure R2**). This finding supports the shared origin of the WGD2 event in the *Flaveria* lineage and sunflower.

Figure R1. Distribution of Ks values for paralogous gene pairs among different species. Figure presents the distribution of Ks for paralogous gene pairs within and five *Flaveria* species, lettuce, and sunflower. Red stars indicate the WGD2 event and blue star indicates the WGT1 event. (Abbreviations: WGD2: whole genome duplication 2, WGT1: whole genome triplication 1). This figure has been included as the Extended Data Fig.2a of the revised manuscript.

Figure R2. Synteny Analysis between *Flaveria* Species and Sunflower. Figures represent the synteny alignment between two *Flaveria* species Froben (a) and Ftri (b) and the sunflower (Hann) genome. Each panel

shows the alignment of homologous regions between the respective *Flaveria* species and sunflower chromosomes. Circles represent syntenic blocks between the *Flaveria* species and sunflower, with purple circles indicating conserved chromosomes, and orange representing rearrangements, corresponding rearranged chromosomes are interconnected with red lines. The x-axis represents the chromosomes of the *Flaveria* species, and the y-axis corresponds to the sunflower chromosomes. This figure has been included as the Extended Data Fig.2b of the revised manuscript.

2. Lines 194-195: The reference to "Fig. 2c" does not adequately demonstrate that CA has extra copies in C₄ species, and "Extended Data Fig. 3" does not clearly show that either CA, PEPC, and PEPC-k have extra copies in C₄ species. The authors should provide more direct evidence or a more detailed explanation for their claims.

Response: We appreciate for reviewer pointing out this point. We are sorry for this mistake. Extended Data Fig.3 should be revised to Extended Data Fig.4, in which the gene phylogenetic tree of CA and PEPC-k were shown (termed as $\beta CA1.1$, $\beta CA1.2$; PEPC-k1.1~ PEPC-k1.6) (**Figure R3**). Fig.2c shows the phylogenetic tree of PEPC, in which the C₄ species showed three PEPC1, termed as PEPC1.1, PEPC1.2 and PEPC1.3 (**Figure R4**).

Figure R3: C₄ version of CA and PEPC-k show more copies in the C₄ species Ftri than in other *Flaveria* species resulting from gene duplications. (a) and (b) illustrate the gene trees of CA and PEPC-k respectively. Gene trees were constructed based alignment of protein sequences. Bootstrap scores were from 100 bootstrap

samplings. Bars show transcript abundances in transcript per kilobase per million mapped reads (TPM). The bars show mean \pm s.d. (n=3). (c) and (d) Integrated Genome Viewer (IGV) of RNA-seq reads and ATAC-seq reads of two *CA1* and four *PEPC-k1* that anchored to chromosomes in Ftri respectively. (Abbreviations: *CA1*: carbonic anhydrase1; *PEPC-k1*: phosphoenolpyruvate carboxylase kinase1). This figure has been included as the Extended Data Fig.4 of the revised manuscript.

Figure R4: The gene tree of *PEPC* orthologs. *PEPCs* from *Arabidopsis thaliana* (Atha) are used as outgroups. *PEPC1* is the functional version according to the highest expression levels among all *PEPCs* (indicated with red circles). *PEPC1* has three copies in the *C₄* species Ftri, showing comparable transcript abundances in transcript per kilobase per million mapped reads (TPM). The bars show mean \pm s.d. (n=3). This figure has been included as Figure 2c of the revised manuscript.

3. Lines 89, 119, 261: What are the abbreviations of CREs, mya, ERF? Please use the full names the first time they appear and check for other abbreviations throughout the text.

Response: We appreciate for reviewer pointing out this point. CREs stands for *cis*-regulatory elements, mya stands for million years ago, and ERF stands for ethylene responsive factor. We have conducted a comprehensive review of the entire document to verify that all abbreviations are properly defined at their initial use.

4. Lines 260-261 How many known motifs were used in the analysis and where are these motifs from? The rationale for studying them cannot simply be based on the identification of three out of nine known CREs in C₄ species. The presence of ERF CREs in C₃ species is not demonstrated in the text, and it is unknown whether ERF CREs are specific to C₄ species or if their existence affects C₄ photosynthesis. The enrichment score should also be provided in Fig 3a.

Response: We thank the reviewer for raising this important issue. To identify enriched *cis*-regulatory elements (CREs) in the promoter sequences (3 kb upstream from the start codon) of C₄ genes, we used the findMotifsGenome.pl tool from the HOMER suite (v4.11.1). The known motifs were integrated within the package ("known.motifs" file). In version 4.11.1, a total of 1,006 known motifs were included.

We appreciate the reviewer's comment regarding the rationale for studying ERF CREs. It is worth pointing out here out of so many motifs and corresponding TFs in C₄ *Flaveria*, only nine were enriched in the promoter regions of the C₄ genes. Out of these nine CREs, three were ERF CREs, which is highly statistically significant. We also investigated the enriched CREs in the promoters of orthologous C₄ genes in C₃ and C₃-C₄ species. Our results showed that the promoter regions of orthologs of C₄ genes in Frob (C₃) are enriched with three known motifs, including one ERF motif. In contrast, Fson (C₃-C₄) and Fram (C₃-C₄) do not show enrichment in ERF motifs. The counterparts of C₄ genes in Flin showed no enrichment in any motifs (the enrichment scores (q-values) are provided alongside the enriched motifs (**Figure R5**). In addition, we have also applied ATAC-seq to predict CREs in open chromatin regions of Ftri. Our findings again indicate that ERF CREs are enriched in these open chromatin regions. Additionally, we have identified a higher abundance of ERF transcription factors predicted to regulate the expression of C₄ genes. Based on all these observations, we hypothesize that ERF *cis*-regulatory elements and ERF TFs may play roles in the evolution of C₄ photosynthesis. In the revised manuscript, we have also included *in vitro* experiments to validate the regulation of C₄ genes by ERF TFs in Ftri (**Figure R6**).

Frob (C ₃)	Motif name	Ftri (C ₄)	Motif name
	Ppas4 (q=0.05)		PHV(HB) (q=0.007)
	Hnf1 (q=0.049)		bZIP (q=0.006)
	ERF104 (ERF) (q=0.04)		CRF10 (ERF) (q=0.05)
Fson (C₃-C₄)			MYB65 (q=0.05)
	ANAC062 (NAC) (q=0.05)		PRDM15 (q=0.05)
	Cdx2 (homeobox) (q=0.04)		RAP26 (ERF) (q=0.05)
Fram (C₃-C₄)			MYB3R4 (q=0.05)
	GSC (Homeobox) (q=0.05)		KLF6 (q=0.05)
	PR(NR) (q=0.05)		At1G77640 (ERF) (q=0.05)

Figure R5: Enriched motif of protein sequences of C₄ genes and their orthologous. This figure presents the enriched *cis*-regulatory elements (CREs) of C₄ genes in Ftri (C₄) and their orthologous from C₃ and C₃-C₄ species. The q-values associated with each motif are shown. ERF motifs are highlighted in pink. Note that counterparts of C₄ genes in Flin do not exhibit enriched motifs. This figure has been included as Extended Data Fig.8 of the revised manuscript.

Figure R6: Identifying *cis*-regulatory elements and TFs that regulate C₄ genes. (A) Electrophoretic mobility shift assay (EMSA) was performed with ERF TFs and cy5-labeled partial DNA sequence (probe) of their regulated C₄ genes' promoter from *F. tri*. Labeled probes were incubated with GST-TF-10xHis protein. GST represents GST-10xHis protein without TF. For competition analysis, the binding reaction was performed with addition of 200-fold of corresponding unlabeled probe. Bands corresponding to DNA-protein complexes (shift) or free probes are indicated by arrows. (B) Structure of reporter and effector plasmids. For the reporter constructs, 35S promoter, C₄ gene (*CA1.1* or *PEPC1.1*) promoter, firefly luciferase (*LUC*) and Renilla luciferase (*REN*) are indicated. For the effector construct, ERF TF (ERF12 or ERF61) was driven by the 35S promoter. (C) and (D) ERF12 and ERF61 activates the expression of *CA1.1* and *PEPC1.1* (Significance was calculated with t-test, n=3). Leaf epidermal cells of *Nicotiana benthamiana* were transfected with infiltration buffer (Mock), reporter DNA (*pCA1.1* or *pPEPC1.1*) with *Flag*, reporter DNA with *ERF-Flag* (ERF12 or ERF61). This figure has been included in the revised manuscript as Figure 5f-i.

5. Line 280: "Extended Data Fig. 3" does not depict the chromatin accessibility of CA1 and PEPC-k1. Please recheck the numbering sequence of the Extended Data Figures in the manuscript.

Response: We apologize for any confusion caused by the incorrect reference in the text. The intended figure for this information is Extended Data Fig. 4. To rectify this oversight, we have verified the numbering sequence of the Figures and Extended Data Figures throughout the manuscript

6. Line 282: there is some missing word after “previous”

Response: Thank you for pointing out the typographical error. The phrase now correctly reads as "previously published."

7. Lines 277-278: The three copies of the FtriPEPC1 gene have identical sequences and, as described in lines 213-214, share the same sequences from approximately 2500 base pairs upstream to about 2000 base pairs downstream. Thus, assuming the preservation of multi-mapped reads, their RNA-Seq and ATAC-Seq results would necessarily be the same, which does not elucidate any biological issue.

Response: We appreciate your critical perspective and concur with your comments. In light of the reviewer’s comment, the result related to the three copies of FtriPEPC1 gene are rephrased as following:

Given the complete sequence identity in the upstream regions of the three FtriPEPC1 paralogs, the analyses of chromatin accessibility unsurprisingly showed consistent patterns across these genes (Fig.5a). (Page 18, lines 383-385)

8. Lines 327-328: How to compare ortholog transcript and protein abundances from different species? How are the data normalized for comparison? It’s unclear from the text.

Response: We appreciate reviewer’s feedback. The detailed method for comparing transcript abundances and protein abundances among different species were given in the revised manuscript, which was also shown in the following:

For the inter-species comparison across different *Flaveria* species, orthologous gene pairs between the remaining four *Flaveria* species and *Frob* were predicted through blast (v2.2.31+) (Camacho et al., 2009), identifying the top hits with the E value threshold of $1e^{-5}$ and a sequence identity requirement of at least 60%. A K-means clustering analysis was then performed on the transcript abundances and protein abundance data separately, leveraging the unified *Frob* annotation. Differential protein abundance analysis between different *Flaveria* species was carried out using a two-sample t-test with a p value threshold of 0.05, and foldchange of 1.2. (Pages 32-33, lines 729-733)

With respect to normalization, we compared the abundance of orthologous genes across species after normalizing gene expression with total mapped reads and gene length as the transcript per kilobase of exon model million mapped reads (TPM) values in each sample. TPM is a well-established normalization method, which allowed us to quantify transcript abundance relative to the library composition of each species independently, thereby facilitating a biologically relevant comparison.

9. Line 376: The authors state that "The codon usage patterns of C₄ genes were conserved," yet in line 383, the term used is "all protein-coding genes." Is there a conflict here?

Response: We appreciate your meticulous reading and thoughtful inquiry regarding the description of our findings. Our analyses revealed that not only the C₄ genes but the general codon usage patterns across all genes display a high degree of conservation across different *Flaveria* species. In the original manuscript, we use this subtitle "The codon usage patterns of C₄ genes were conserved," to highlight the conservation of codon usage patterns within the C₄ gene set across different *Flaveria* species.

In the revised manuscript, we have removed the comparison of codon bias across species since no difference of codon usage patterns was identified for C₄ genes. Instead, we conducted new experiments using ribosome profiling (Ribo-seq) for Frob (C₃) and Ftri (C₄) to investigate changes in translation efficiency during evolution. Our new results showed that the translation efficiency was comparable between C₃ and C₄ species (**Figure R7**). This finding further supports the conclusion that the increased protein abundances of C₄ genes in Ftri (C₄) are predominantly attributed to increased transcriptional abundances. (Pages 15-16, lines 296-344)

Figure R7: Translation efficiency comparison between Frob and Ftri

(a) GC₃ comparisons of C₄ genes across five *Flaveria* species. As FramPEPC-*k1* is missing, its paralog (FramPEPC-*k2*, FramNA03444, see Extended Data Fig. 4) was utilized as a comparison for Fram in this context. (b) Abundances of ribosome protected fragment (RPF) of C₄ genes. (c) and (d) Distribution of RPF-to-RNA ratio and protein-to-RPF ratio. (e) Scatter plots of RPF versus RNA of Frob and Ftri. Low/high/moderate RPF-to-RNA genes are labeled in pink/light blue/grey. (f) Scatter plots of protein versus RPF of Frob and Ftri. Low/high/moderate protein-to-RPF genes were labeled in pink/light blue/grey. Low protein-to-RPF C₄ genes were indicated with red arrows. C₄ and photorespiratory genes are labeled in red and blue, and Pearson correlation (PC) between RPF vs. RNA, protein vs. RPF is shown in the parentheses on top of each panel in (e) and (f). (g) and (h) RPF-to-RNA ratio and protein-RPF ratio for C₄ genes and photorespiratory genes. Statistical significance was determined by two-tailed Wilcoxon test (* p < 0.05, ** p < 0.01, *** p < 0.001). This figure has been included as Figure 4 of the revised manuscript.

10. Lines 391-395: The text attempts to illustrate nuanced codon usage variations between different species, but the consequences of these differences are not demonstrated in the manuscript.

Response: We are grateful for your insightful suggestion. One noteworthy aspect influencing both transcription and translation is the frequencies of G+C in third positions of codons (GC_3). We compared the GC_3 of C_4 orthologous genes in the five *Flaveria* species and found no significant differences in GC_3 among these C_4 orthologous genes. Therefore, In the revised manuscript, we focused on investigating translational efficiency between C_3 and C_4 species based on Ribo-seq data for Frob (C_3) and Ftri (C_4). The comparison of codon usage variations between different species was removed in the revised manuscript.

11. Line 139: "*Flaveria*" should be italicized, and the formatting throughout the text should be checked for consistency.

Response: We appreciate your attention to detail, all the "*Flaveria*" is consistently italicized throughout the text.

12. Figure 1a: The time axis should not have negative values.

Response: We deeply appreciate for reviewer pointing out this mistake for us. The time axis was corrected.

13. Figure 2d: The colinearity plot between genes should be presented in the same manner as Figure 2b, without manual connections.

Response: We appreciate for reviewer's comment, the collinearity plot was re-drawn using JCVI package as following (**Figure R8**):

Figure R8: Collinearity plot was re-drawn using JCVI package. This figure has been included as Figure 2d of the revised manuscript.

14. The provided link (https://db.cngb.org/codeplot/datasets/public_dataset?id=Flaveria) doesn't include gene annotation GTF files.

Response: We appreciate reviewer pointing out this issue. The gene annotation files for genome assemblies have been uploaded to China National GeneBank (CNGB) under same project [CNP0003058]. A quick link to access the gene annotation files for *Flaveria robusta* is available at <https://ftp.cngb.org/pub/CNSA/data2/CNP0003058/CNS0557296/CNA0050661/>. Please find a snapshot of the FTP directory structure below (**Figure R9**):

名称	大小	更新时间	Aspera 高速下载	复制下载命令	直接下载
Frob_chr_transcript_order.gff3	38.64MB	2024-07-30 06:19:40			
Frobusta_geome.fa.gz	139.08MB	2022-05-28 02:28:37			
Frobusta_geome.fa.gz.md5	53B	2024-04-17 16:58:54			

Figure R9: A snapshot of genome and genome annotation file for Frob in CNGB.

Reviewer #3:

The convergent evolution of C₄ photosynthesis is an important question with clear applied implications in plant science. The authors generated five chromosome-scale assemblies of *Flaveria* species that represent the spectrum of C₃-C₄ photosynthetic biology. The authors find through analysis of RNA-seq and ATAC-seq data that cis-regulatory evolution played a major role in the acquisition of C₄ photosynthesis in *F. trinervia*. The chromosome-assemblies appear to be of high-quality and will be useful to the plant science and photosynthesis community. The manuscript is well written, and the figures are nicely prepared. While I do think that this study will be of general interest, I have concerns regarding the ATAC-seq data quality and analysis that will require attention before publication can be justified. More details below.

Response: We thank the reviewer for his/her thoughtful and favorable review, as well as critical questions and suggestions for revision. In this revision, we performed new analyses and experiments to address the reviewer's concerns and revised the whole manuscript according to the reviewer's suggestions. The new results are fully consistent with and strengthen our findings. Please check the specific response to each comment below.

Major comments:

I could not find how ATAC-seq replicates were assessed for reproducibility in the methods or main text. Therefore, I cannot evaluate the quality of these data (however, I do have concerns regarding data quality given the strange ATAC profile of Figure 3b, low signal/noise ratios observed in Figure 3d, and extremely low unique mapping rates). At a minimum, peaks should be called for each replicate, peaks merged by taking the union across all replicates, read counts scored against all peaks across replicates, and correlations taken between replicates. The fraction of reads in reproducible peaks (FRiPs) should also be reported for each sample. Although the methods section does not specify how correlations were estimated, please be aware that using counts taken from uniform bins across the genome represents a zero-inflated distribution and will result in highly inflated correlation estimates. Ideally, the authors should perform IDR analysis (from the ENCODE project) to assess reproducibility and keep peaks reliably identified in multiple samples. Note, please pay special attention to the parameters for peak calling with ATAC-seq data. More on this below.

Line 732: The parameters used for THS peak calling are inappropriate. I see this mistake all the time. MACS2 is a general usage peak caller that creates pseudofragments based on PE data when used in PE-mode (as is done here), which would be appropriate for ChIP-seq-like data. The result is that MACS2 is using the pile-up of the inferred fragment. For ATAC-seq data, you want the pile-up to happen on the integration sites – the ends of the fragments, not the center. As a result of using these parameters, the authors are likely calling THS that are in between accessible chromatin, which will obviously create artefacts/issues for downstream interpretation. Moreover, many of these THS peaks will be nucleosomal (the opposite of accessible chromatin), because Tn5 can insert adapters in linkers flanking nucleosomes. This is the basis of nucleosomal banding patterns seen in ATAC-seq libraries from a TapeStation or Fragment Analyzer. The correct way to call peaks with ATAC-seq data is to isolate the integration site (5' end of reads at single-bp resolution) and create pseudofragments that are centered on the integration site (rather than in the center of the PE coordinates) using the `–shift` and `–extsize` arguments. The larger issue is that because some proportion of THS represent in-accessible closed chromatin, it's hard to know how much of the observed CRE enrichments are reflecting biology versus technical artefacts (especially considering the ERF TFs bind highly degenerate GC-rich motifs). This analysis needs to be redone correctly before it is possible to evaluate the biological implications of this experiment.

As mentioned above, the meta ATAC-seq profile from Figure 3b looks strange to me. In every species I've seen, the peak of accessible chromatin is upstream of the TSS, not within the gene body. This suggests to me incorrect analysis parameters, an artefact of data processing, or bad data quality. If the authors can show the data is of good quality (see my first comment), I would take a careful look at how reads were processed by deeptools. Typically, you would want to use the Tn5 integration sites (again, not the PE reads), to create bigwigs for such analysis. MACS2 can output the coverages of Tn5 integration sites as bedgraphs if you give it a single-bp resolution Tn5 insertion site.

Response: Thank you for your insightful comments regarding our ATAC-seq data analysis. MACS2 has been applied for ATAC-seq data analysis for many studies (xxxxxx). We have carefully considered your suggestions and have reanalyzed the ATAC-seq data to address your concerns. Here are the steps we have taken:

1. Quality control and reporting:

We performed additional quality control checks on our ATAC-seq data to ensure its reliability. Adapters were removed using "cutadapt," and we reported the quality of clean reads (**Figure R1**).

These preliminary assessments support the suitability of these ATAC-seq data for downstream processing and analysis.

2. Reporting mapping statistics.

We used Bowtie2 to align ATAC-seq reads to the Ftri genome. Regarding the low uniquely mapped reads for ATAC-seq (34% for Ftri1 and 29% for Ftri2), we think this may be due to recent tandem duplications in the Ftri genome. For example, the unique mapping ratio of DNaseI sequencing in maize has been reported as 6-20% (Burgess et al., Plant Cell, 2019, doi: 10.1105/tpc.19.00078, Supplemental Dataset 1). Both uniquely and multiply mapped reads were utilized for peak prediction.

3. Filtering Duplicate Reads from PCR

Specifically, we first used “`samtools view -q 10`” to remove low-quality mapped reads, and then used “`samtools collate`” to group reads with the same names, and “`samtools fixmate -m`” to fill in mate coordinates and add mate score tags, and “`samtools markdup -r`” to remove duplicate reads. Previously, we used “`samtools rmdup`”, which is not recommended according to the SAMtools documentation.

4. Adjusting the start position of ATAC-seq mapping

We adjusted the start position of each read based on strand information. Reads on the forward strand were shifted by +4 bp, and reads on the reverse strand were shifted by -5 bp, using “`alignmentSieve -ATACshift`”.

5. Correcting peak calling parameters:

We applied MACS2 to call peaks for Ftri1 and Ftri2 using the updated command: `macs2 callpeak -f BAMPE --bdg -g 1.7e9 -q 0.05 --broad --nomodel --keep-dup all --nolambda --shift -100 --extsize 200`

6. Performing IDR analysis

We used “idr” to perform irreproducibility discovery rate analysis to assess the consistency of the two biological replicates of ATAC-seq data from Ftri. As a result, we obtained 14,333 peaks at the IDR threshold of 0.05 (**Figure R2**).

7. Calculating FRiPs

We calculated the Fraction of Reads in Reproducible Peaks (FRiPs) from the IDR analysis. The results show that the FRiPs were 15% for replicate 1 (Ftri1) and 16% for replicate 2 (Ftri2).

8. Calculating of correlation between two replicates

We calculated the correlation of mapped read counts from conserved peak regions across two distinct biological replicates of Ftri. Results showed a high Pearson correlation of 0.98 between the two replicates (**Figure R3**).

9. Generating bigwig files

We generated BigWig files from the bedGraph files output by MACS2 using "bedGraphToBigWig". We then drew the intensity of ATAC-seq against the bed file of all genes. The updated result showed that the peaks were on the upstream of the TSS site (**Figure R4a**). We apologize for the mistake in the previous result where we used the bed file of called peaks, which resulted in peaks downstream of the peak start site (wrong termed as TSS in our previous result).

10. IGV tracks of examples of highly expressed genes in Ftri leaves

We updated the IGV tracks using the new BigWig files. We provided examples of several photosynthesis genes and C₄ genes, which are expected to be highly expressed in leaves. The results displayed accessible chromatin regions upstream of the TSS of those genes (**Figure R4c**).

11. Updating accessible cis-regulatory elements

accessible cis-regulatory elements have been updated based on the new results (**Figure R4d**).

We believe these revisions significantly strengthen our manuscript and address the concerns raised. We appreciate your valuable comments, which has helped us improve the quality of our analysis.

a General Statistics

Copy table | Configure Columns | Plot | Showing 4/4 rows and 3/5 columns.

Sample Name	% Dups	% GC	M Seqs
FtriR1_fastp_R1	52.2%	43%	25.3
FtriR1_fastp_R2	45.4%	43%	25.3
FtriR2_fastp_R1	55.5%	42%	33.9
FtriR2_fastp_R2	48.3%	42%	33.9

b Adapter Content

4

Help

The cumulative percentage count of the proportion of your library which has seen each of the adapter sequences at each position.

No samples found with any adapter contamination > 0.1%

c

Per Sequence Quality Scores

4

Help

The number of reads with average quality scores. Shows if a subset of reads has poor quality.

d

Sequence Quality Histograms

4

Help

The mean quality value across each base position in the read.

Y-Limits: on

e

Per Sequence GC Content

4

Help

The average GC content of reads. Normal random library typically have a roughly normal distribution of GC content.

Y-Limits: on

f

Per Base N Content

4

Help

The percentage of base calls at each position for which an N was called.

Y-Limits: on

Figure R1: Quality Control Assessment of ATAC-seq Data for Ftri. This figure presents the results of FastQC and MultiQC analyses for two biological replicates of ATAC-seq data from Ftri, including read counts and duplicate percentages (a), adapter content (b), per-sequence quality scores (c), sequence quality histograms (d), per-base GC content (e), and per-base N content (f).

Figure R2: Replicate consistency of ATAC-seq data for Ftri using IDR analysis. This figure presents the results of the Irreproducible Discovery Rate (IDR) analysis applied to assess the consistency of two biological replicates of ATAC-seq data from Ftri. The upper left panel displays a scatterplot of ranks versus ranks, and the upper right panel shows log10 scores versus ranks, again with red designating regions surpassing the IDR threshold. Regions having an IDR value greater than or equal to 0.05 are indicated in red. The lower panels depict rank distribution plots for each sample against the negative logarithm of IDR values, with black bars representing the number of peaks falling within each rank-IDR bin. This figure has been included as Figure 4 of the revised manuscript. This figure has been included in the Supplementary Note 13 of the revised manuscript.

Figure R3: Replicate consistency analysis of ATAC-seq data in Ftri. These figures illustrate the correlation of mapped read counts from conserved peak regions across two distinct biological replicates of Ftri. The assessment focuses on conserved peak regions, which were identified using IDR (Irreproducible Discovery Rate) with a stringent false discovery rate (FDR) threshold of 0.05. The region with counts within 1000 is highlighted for clarity. This figure has been included in the Supplementary Note 13 of the revised manuscript.

Figure R4: Predicted *cis*-regulatory elements in the *C₄* species *Ftri* using ATAC-seq. (a) The plots show profiles of average ATAC-seq peak intensity from two independent biological replicates for all annotated genes near the transcription start sites (TSS). The x-axis represents the distance from the TSS, y-axis depicts the normalized read count per million mapped reads. (b) The pie chart demonstrates the distribution of ATAC-seq peaks in relation to gene positions. (c) Genome browser tracks displaying accessible chromatin regions and transcript abundances (RNA-seq) of examples of

photosynthesis genes and C₄ genes. (d) Enriched *cis*-regulatory elements (CREs) in three types of accessible chromatin regions (ACR-CREs), *i.e.*, genic (gACR-CREs: overlapping a gene), upstream (upACR-CREs: within 3kb upstream of the start codon of a gene) and downstream (down ACRs-CREs: within 3kb downstream of the stop codon of a gene). (e) Enriched ACR-CREs associated with photosynthetic genes and photorespiratory genes. (Abbreviations: *RUBSC1b*: *Rubisco small subunit 1b*; *Lhca1b*: *light-harvesting complex a 1b*; *PGR5-like*: *proton gradient regulation 5-like*; *CA1*: *carbonic anhydrase 1*; *PPDK*: *pyruvate orthophosphate dikinase*; *PPDK-RP*: *PPDK regulatory protein*; *NADP-ME4*: *NADP-dependent malic enzyme 4*; *PPT1*: *phosphate/phosphoenolpyruvate translocator 1*; *NHD1*: *sodium: hydrogen antiporter 1*; ATAC-seq: transposase-accessible chromatin using sequencing; ACR: accessible chromatin regions; CREs: *cis*-regulatory elements.) This figure has been included as Extended Data Fig.9 in the revised manuscript.

Line 222: "... all three FtriPEPC1's were upregulated ..." – Although the authors note a significant difference between low and normal CO₂ conditions, the differences appear trivial. Moreover, in the methods, the authors report using a P-value threshold of 0.05. Did the authors perform multiple test correction? If not, then the false-discovery rate will be incredible high and many detected differences are likely to be technical rather than biological in origin.

Response: In the revised manuscript, we only include the RNA-seq data for plants grown under normal conditions to compare the transcript abundance between different orthologs and paralogs. In the revised manuscript, we removed the RNA-seq data under low CO₂ conditions since this dataset was not the focus of the paper and did not add value to the text. The revised figure is as following (**Figure R5**):

Figure R5: The gene tree *PEPC*. This figure has been included as Fig. 2c in the revised manuscript.

Minor points:

Line 100: "... genome size has gradually increased..."

Response: Thanks for pointing out this. The sentence was revised to "The assembled genome size has gradually increased during the evolution of C₄ photosynthesis in this genus".

Figure 3d,e: How are the tracks scaled? There are no axes or scale bars.

Response: We appreciate the reviewer for pointing out this issue. In the revised figures, the Y-axes for RNA-seq (TPM) and ATAC-seq (CPM) are now clearly labeled. Additionally, the chromosome regions for each IGV track are indicated at the top of each track. For example, the legend for the track of RuBSC1B is "RuBSC1B (Ftri1G19684) Chr1:7,368-7,375 kb."

Figure 3c: How does the ERF enrichment in C₄ genes compare the genome background across all ACRs?

Response: To identify enriched ACRs for C₄ genes (as well as for photosynthetic genes and photorespiratory genes), we compared the frequency of ACRs associated with C₄ genes against the frequency of all predicted ACRs. Specifically, we identified 257 gene associated ACRs in total (background), and 111 ACRs associated with C₄ genes. In order to determine enrichment, we calculated the FDR (false discovery rate) for each ACR relative to the background. Specifically, we randomly selected 111 motifs from 257 ACRs and performed 1,000 re-samplings. The FDR for a specific ACR was calculated as the average frequency of this ACR from the 1,000 re-samplings divided by the frequency of the ACR observed in C₄ genes. As a result, 10 ACRs were found to be enriched in C₄ genes, with an FDR threshold of <0.001. The FDR for the ERF motif was <0.0001.

Why does the GRN analysis include CREs 3kb upstream of TSS instead of leveraging THSs?

Response: We appreciate for reviewer's comment. As we have not yet obtained ATAC-seq data for the C₃ or C₃-C₄ species at this stage, we have used CREs located 3 kb upstream of the start codon for all species. More studies in comparisons between open chromatin features of different *Flaveria* species will be conducted in our future study.

Reviewer #4:

In their manuscript, Lyu et al. conducted genome assembly for five *Flaveria* species, demonstrating superior quality compared to prior genome annotations. By harnessing this genomic data, the authors identified genes encoding C₄ enzymes and transporters, discovering duplication events in C₄ genes within C₄ species. Subsequently, ATAC-seq analysis revealed a prevalence of ERF cis-regulatory elements and ERF TFs in Ftri. Lastly, comparing protein-to-mRNA ratios (PTR) among these species, the study suggests that transcriptional regulation, rather than translational regulation, drives the protein levels of C₄ genes in Ftri species. Notably, the study highlights the significant role of transcriptional regulation in C₄ photosynthesis across *Flaveria* species. However, additional evidence is required to bolster the conclusions drawn by the authors.

Response: We thank the reviewer for his/her thoughtful and favorable review, as well as critical questions and suggestions for revision. In this revision, we performed new analyses and experiments to address the reviewer's concerns and revised the whole manuscript according to the reviewer's suggestions. The new results are fully consistent with and strengthen our findings. Please check the specific response to each comment below.

p293 - p305, the manuscript lists the five most abundant TF families across the five *Flaveria* species. Besides ERF, other TF types, such as bHLH and MYB, are notably prevalent in Ftri. However, it's challenging to definitively establish the direct correlation between these TFs and C₄ genes solely from the C₄GRNs. Additional experimental assays should be devised to thoroughly investigate this relationship.

Response: We appreciate reviewer's comments. Accordingly, we performed electrophoretic mobility-shift assay (EMSA) and transient expression to verify the binding of predicted ERF TFs to the promoter of C₄ genes in Ftri (C₄) (**Figure R1**). The details are as following: Seven ERF TFs that were predicted to regulate Ftri*CA1*, *PEPC1.1* and *PEPC-k1.1* were used for experiment. Cognate ERF TF binding site (TFBS) within 2k bp upstream of their (*CA1*, *PEPC1.1* or *PEPC-k1.1*) start codon to perform EMSA. Our EMSA experiments verified the binding of ERF12 to the promoter of Ftri*PEPC1.1* and *PEPC-k1.1*, the binding of ERF61, ERF51 and ERF1 to the promoter of *CA1.1* (**Figure R1A**). We then performed transient transcription assay to further verify the regulation of ERF12 and ERF61 to C₄ genes. We used dual-luciferase reporter plasmids containing the firefly luciferase (*LUC*) gene driven by *CA1.1* promoter (200 bp from the start codon) and *PEPC1.1* promoter

(250 bp from the start codon) and the Renilla luciferase (*REN*) gene driven by the constitutive 35S promoter in the analysis (**Figure R1B**). Results showed that ERF12 and ERF61 displayed significantly higher LUC/REN ratio compared with Flag tag (**Figure R1C, D**), suggesting the activation of ERF12 and ERF61 on *CA1.1* and *PEPC1.1*. (Page 20, lines 416-428)

Figure R1: Identifying cis-regulatory elements and TFs that regulate *C₄* genes. (A) Electrophoretic mobility shift assay (EMSA) was performed with ERF TFs and cy5-labeled partial DNA sequence (probe) of their regulated *C₄* genes' promoter from *F. tritici*. Labeled probes were incubated with GST-TF-10xHis protein. GST represents GST-10xHis protein without TF. For competition analysis, the binding reaction was performed with addition of 200-fold of corresponding unlabeled probe. Bands corresponding to DNA-protein complexes (shift) or free probes are indicated by arrows. (B) Structure of reporter and effector plasmids. For the reporter constructs, 35S promoter, *C₄* gene (*CA1.1* or *PEPC1.1*) promoter, firefly luciferase (*LUC*) and Renilla luciferase (*REN*) are indicated. For the effector construct, ERF TF (ERF12 or ERF61) was driven by the 35S promoter. (C) and (D) ERF12 and ERF61 activates the expression of *CA1.1* and *PEPC1.1* (Significance was calculated with t-test, n=3). Leaf epidermal cells of *Nicotiana benthamiana* were transfected with infiltration buffer

(Mock), reporter DNA (*pCAI.1* or *pPEPC1.1*) with *Flag*, reporter DNA with *ERF-Flag* (ERF12 or ERF61). This figure has been included in the revised manuscript as Figure 5f-i.

p689 - p707, the authors did not provide a clear explanation of the proteomics assay. Details such as the quantity of samples (e.g., ?g) and protein extraction buffer (e.g., ?mL) utilized for total protein extraction, as well as the number of technical or biological replicates employed, remain unspecified. The observation of low PTR for certain C₄ proteins in Ftri, as depicted in figures 4c and 4d, led to the conclusion that transcriptional regulation significantly impacts C₄ photosynthesis. However, it's crucial to acknowledge that other factors, such as epigenetic regulation or post-transcriptional/translational processes, could influence protein abundance. Moreover, it's pertinent to assess the functionality of these C₄ enzymes in these species. Have the authors conducted additional experiments to further validate the significance of transcriptional regulation in Ftri?

Response: We appreciate reviewer pointing out these issues. We have provided detailed information regarding the total protein extraction procedure in the revised manuscript. Specifically, 0.1 g of mature leaves were collected from one-month-old plants and immediately placed into liquid nitrogen. Six biological replicates were used for each species. The frozen leaf samples were thoroughly ground and then incubated in 0.6 mL of lysis buffer.

The revised methods for protein extraction are as following:

Approximately 0.1g mature leaves were collected from one-month old plants as depicted above, and leaves were put into liquid nitrogen quickly. Six biological replicates were used for each species. Frozen leaf samples were ground thoroughly and then incubated in 0.6ml lysis buffer (100 mM Tris-Base, 100 mM EDTA, 50 mM Borax, 50 mM Ascorbic Acid, 30% (m/v) Sucrose, Triton X-100 (final concentration 1%), 10 mM TCEP, 1mM PMSF, complete EDTA-free protease inhibitor cocktail (PIC) (Roche)). The lysis buffer was prepared fresh, and the pH was adjusted to 8.2 using ammonium hydroxide (NH₄OH). After the addition of TCEP, the pH was readjusted to 8.0. The buffer was stored at -80°C until needed and was thawed at room temperature before use. (Page 32, lines 708-718)

Although our result suggests a significant impact of transcriptional regulation on C₄ photosynthesis, we fully acknowledge that other factors, such as epigenetic regulation and post-transcriptional/post-translational processes, could also influence protein abundance. To address these

concerns, we have expanded our discussion to include a more comprehensive consideration of these regulatory mechanisms.

“However, it is important to acknowledge that other factors, such as epigenetic regulation and post-transcriptional/post-translational processes, could also play significant roles in influencing protein abundance” (Page 23, lines 489-491)

To assess the functionality of these C₄ enzymes in the five *Flaveria* species, we first performed Western blot experiments of three key enzymes. The results showed that the protein abundances are consistent with those obtained from proteomics (**Figure R2**), in which C₄ species Ftri showed the highest protein abundances among the five species. Result showed that Flin (C₃-C₄) and Fram (C₃-C₄) showed intermediate protein abundances between C₃ and C₄ species. Fson (C₃-C₄) showed comparable protein abundances with Frob (C₃).

We also conducted *in vitro* enzyme activities analysis of four key C₄ enzymes, in leaf tissues of five *Flaveria* species. Results showed that the enzymatic activities were consistent with the observed protein abundances (**Figure R3**). For PEPC and NADP-MDH, C₃ and C₃-C₄ species showed comparable enzyme activity, whereas Fram showed higher enzyme active than C₃ species for NADP-ME and PPDK.

To further validate the significance of transcriptional regulation of C₄ enzymes, we performed ribosome profiling (Ribo-seq) for Frob (C₃) and Ftri (C₄) to investigate changes in translation efficiency during evolution. Our data show that the translation efficiency is comparable between C₃ and C₄ species (**Figure R4**). This finding further supports the conclusion that the increased protein abundances of C₄ genes in *F. trinervia* (C₄) are predominantly attributed to increased transcriptional abundances.

Figure R2: Protein immunodetection in leaf of five *Flaveria* species. Protein immunoblot analysis was conducted on leaf extracts prepared with the loading amount standardized per unit of fresh leaf weight, three plants from species were analysed. Actin was used as loading control. (Abbreviations: PEPC: phosphoenolpyruvate carboxylase, NADP-ME: NADP-dependent malic enzyme, PPDK: pyruvate orthophosphate *dikinase*). This figure has been included as Extended Data Fig.3c in the revised manuscript.

Figure R3: Enzyme activity of four key C₄ enzymes. Enzyme activities of four key C₄ enzymes PEPC, NADP-MDH, NADP-ME and PPDK were measured in leaf tissues of five *Flaveria* species. The activities are normalized to fresh weight and presented as mean ± SD (n ≧ 3). Significance was determined using two-tailed Wilcoxon rank sum tests, P-values were adjusted with “BH” method (* 0.01 ≧ P < 0.05, ** 0.001 ≧ P < 0.01, *** P < 0.001). The activation of PPDK in Frob was unmeasurable due to very low protein abundances. (Abbreviations: PEPC, PEP carboxylase; MDH, malate dehydrogenase; NADP-ME, NADP-malic enzyme; and

PPDK, pyruvate, orthophosphate dikinase). This figure has been included as Extended Data Fig.3b in the revised manuscript.

Figure R4: Translation efficiency comparison between Frob and Ftri. (a) GC₃ comparisons of C₄ genes across five *Flaveria* species. As FramPEPC-k1 is missing, its paralog (FramPEPC-k2, FramNA03444, see Extended Data Fig. 4) was utilized as a comparison for Fram in this context. (b) Abundances of ribosome protected fragment (RPF) of C₄ genes. (c) and (d) Distribution of RPF-to-RNA ratio and protein-to-RPF ratio. (e) Scatter plots of RPF versus RNA of Frob and Ftri. Low/high/moderate RPF-to-RNA genes are labeled in pink/light blue/grey. (f) Scatter plots of protein versus RPF of Frob and Ftri. Low/high/moderate protein-to-RPF genes were labeled in pink/light blue/grey. Low protein-to-RPF C₄ genes were indicated with red arrows.

C₄ and photorespiratory genes are labeled in red and blue, and Pearson correlation (PC) between RPF vs. RNA, protein vs. RPF is shown in the parentheses on top of each panel in (e) and (f). (g) and (h) RPF-to-RNA ratio and protein-RPF ratio for C₄ genes and photorespiratory genes. Statistical significance was determined by two-tailed Wilcoxon test (* p < 0.05, ** p < 0.01, *** p < 0.001). This figure has been included as Figure 4 in the revised manuscript.

Other minors in the manuscript:

In Figure 1 and the method section (pages 150 and 521), the authors conducted FISH analysis to confirm the chromosome numbers of *Flaveria* species. However, they did not specify which DNA probes were used for this assay.

Response: We appreciate the reviewer's attention to detail. We used centromeric and telomeric repeats sequences probes to count the chromosome numbers of *Flaveria* species. This information has been included in the Supplementary Note 2 of revised manuscript.

The cell suspension (5–8 µL) was placed onto glass slides within a moist chamber and allowed to dry. The slides were then cross-linked using an ultraviolet cross-linking apparatus. Centromeric and telomeric repetitive sequences were used as probes for chromosome identification and counting. The slides were hybridized with these probes, followed by washing and counterstaining with DAPI (NPE Analyzer, NPE 731085, USA). Chromosome numbers of the three *Flaveria* species were determined by visualizing the slides under a microscope (Leica DM2500). (Supplementary Note, Page 5, Supplementary Note 2)

In Figure 2c, the authors labeled PEPC1-4 for the first four clades but omitted labels for their sub-clades. Additionally, in Figure 2d, it would be beneficial to include gene IDs for PEPC1.1 to 1.3. Additionally, the labels of PEPC1.1 to 1.3 could be removed from Figure 2c.

Response: We appreciate the reviewer's comments on our figures. Regarding the classification of the PEPC gene family into four clades (PEPC1 to PEPC4), as each clade typically contains only one gene copy per species, except in cases where tandem duplication events occur. Therefore, we did not label sub-clades. Regarding to Fig. 2d, we included gene IDs for three PEPC1s in the revised Fig. 2 (**Figure R5**).

Figure R5: Tandem duplication of *PEPC1* in *Ftri*. The gene IDs were given in for PEPC1.1, PEPC1.2 and PEPC1.3. This figure has been included as Figure 2d in the revised manuscript.

In Figure 4c, it should be noted that the red dots and blue dots represent proteins, not genes. Dotted lines should be added to segregate high PTR and low PTR in the scatter plot. Furthermore, the labeling of some C_4 proteins by arrows does not align with some low and high PTR, necessitating double-checking in the figure. For Figure 4d, asterisks indicating the p-values should be added to the figure. The conclusion regarding the conserved codon usage patterns in the five *Flaveria* species seems to have minimal contribution. It may be more appropriate to move this information to another supplementary figure.

Response: We appreciate reviewer's suggestion.

In the revised manuscript, we corrected “ C_4 genes” to “ C_4 proteins” and “photorespiratory genes” to “photorespiratory proteins”.

We add a dotted lines to segregate high PTR and low PTR genes with moderate PTR genes in the revised figures.

In the revised figure, only high PTR and low PTR were indicated by arrows.

For Figure 4d, we have included asterisks to indicate the significance levels for the comparisons, providing a clearer visualization of the statistically significant differences (Fig.3d in the revised manuscript)

Regarding the conserved codon usage patterns in the five *Flaveria* species, we have removed the comparison of codon bias between different species, as it did not provide a clear link to the transition to C₄ photosynthesis and lacked sufficient justification. Instead, we focused on using ribosome profiling (Ribo-seq) for *F. robusta* (C₃) and *F. trinervia* (C₄) to investigate changes in translation efficiency during evolution. Our data show that the translation efficiency is comparable between C₃ and C₄ species. This finding further supports the conclusion that the increased protein abundances of C₄ genes in *F. trinervia* (C₄) are predominantly attributed to increased transcriptional abundances. The revised results were shown in Fig. 4 in the revised manuscript.

Reviewer #5:

Lyu and co-authors describe an extensive genomics, transcriptomics and proteomics analysis of five *Flaveria* species to study C₃ to C₄ photosynthesis evolution. Draft genomes for four *Flaveria* species were published before in the context of C₄ evolution (Taniguchi, Y. Y. et al. (2021). *Plant Genome*, 14:e20095), though higher-quality genomes at pseudo-chromosome scale were acquired in this work. As also indicated in the discussion, the increased mRNA and protein levels of C₄ enzymes is known and confirmed in this work. As described more elaborately in the comments below, I have some questions regarding the protein quantification values used and the comparison of them between species. This is important as the calculation of the proposed translational efficiency depends on these. It should be noted that methods such as ribosome profiling are important in such studying mRNA translation, for instance within the field translation efficiency is defined as ribosome footprints divided by mRNA-seq expression. There is thus quite a leap from mRNA to protein intensities to conclude on translational efficiency, which warrants caution with the statements made from these data.

Response: We thank the reviewer for his/her thoughtful and favorable review, as well as critical questions and suggestions for revision. In this revision, we performed new analyses and experiments to address the reviewer's concerns and revised the whole manuscript according to the reviewer's suggestions. The new results are fully consistent with and strengthen our findings.

We appreciate reviewer's comments and suggestion for perform ribosome profiling (Ribo-seq) to compare the translation efficiency of C₄ genes between different species. In the revised manuscript, we conducted Ribo-seq for Frob (C₃) and Ftri (C₄) with two biological replicates for each, to investigate changes in translation efficiency during evolution. Our results showed that the translation efficiency is comparable between C₃ and C₄ species (**Figure R1**). This finding further supports the conclusion that the increased protein abundances of C₄ genes in Ftri (C₄) are predominantly attributed to increased transcriptional abundances.

The details are showed in result section of "The translation efficiency in the C₄ genes between C₃ and C₄ *Flaveria* species" (Pages 15-16, lines 296-344), which is as following:

The translation efficiency in the C₄ genes between C₃ and C₄ *Flaveria* species

In addition to transcriptional regulation, various factors such as RNA stability, translational efficiency, and protein stability contribute to the increased protein abundance. One noteworthy aspect influencing both transcription and translation is the frequencies of G+C in third positions of codons (GC₃)^{42,43}.

We compared the GC₃ of C₄ orthologous in the five *Flaveria* species and found no significant differences in GC₃ among these C₄ orthologous genes (Fig.4a).

To further examine whether there are significant differences in translation efficiency during evolution, we performed ribosome profiling (Ribo-seq) on two representative species, i.e. the C₃ species Frob and the C₄ species Ftri with two biological replicates for each. As Ribo-seq captures the positions of ribosomes on mRNAs, providing a direct measure of translational activity. Simultaneously, we conducted RNA-seq from the same samples for Frob and Ftri to compile translational efficiency (Extended Data Fig. 7a). After filtering out rRNA sequences, around 35% and 25% of the reads mapped to the coding sequences of the corresponding genomes of Frob and Ftri, respectively (Extended Data Fig. 7a). Despite the high amount of ribosomal RNAs, the mapping ratio in our study is relatively low but comparable to those reported in other species, such as 12% in maize⁴⁴ and 16% in *Saccharomyces cerevisiae*⁴⁵. The Ribo-seq data showed a clear triplet periodicity on codons in the reference transcriptome (Extended Data Fig. 7b), and the read length distribution exhibited a peak at 27 to 32 nucleotides, with 94% of fragments mapped to the coding sequence (Extended Data Figs.7 c and d).

Principal component analysis (PCA) based on transcripts per million (TPM) of Ribo-seq data showed that samples from Frob were well separated from those of Ftri, with the first component explaining 65% of the total variance. (Extended Data Fig. 7e). Consistent with result from RNA-seq data, C₄ genes from Ftri exhibited higher transcript abundances in TPM compared to their counterparts in Frob based on Ribo-seq (Fig.4b). To compare the transcriptional efficiency in each species, we calculated the translation efficiency as the ratio of ribosome-protected fragment (RPF) abundances to RNA abundances. To ensure comparability, the translation efficiency was normalized by the mean translation efficiency values of all photosynthesis genes (without C₄ genes). The distribution of translation efficiencies was comparable between Frob and Ftri after normalization (Fig. 4c). Similarly, the distribution of the ratio of protein to RPF abundances was also similar (Fig. 4d).

There was a relatively high correlation between RPF and RNA abundances, with Pearson correlations of 0.81 and 0.88 in Frob and Ftri, respectively (Fig. 4 e). The correlation between protein and RPF abundances fell between the RPF vs. RNA and protein vs. RNA correlations, with Pearson correlations of 0.57 and 0.58 in Frob and Ftri, respectively (Fig. 4f). We defined high and low translation-efficiency genes as the top 5% and bottom 5% of genes ranked by translation efficiency, resulting in 995/743 high and 965/664 low translation-efficiency genes in Frob and Ftri, respectively. Notably, all C₄ genes and a large proportion of photorespiratory genes were among those with

moderate translation efficiency in both species (Fig. 4e). Consistently, the translation efficiency showed no significant difference between Frob and Ftri for both C₄ and photorespiratory genes.

We also defined high and low protein-RPF-ratio genes the same way as protein-RNA-ratio, resulting in 254/250 high and 250/253 low protein-RPF-ratio genes in Frob and Ftri, respectively (Fig. 4 f). In line with the protein and RNA comparisons, more C₄ genes fell into the category of low protein-RPF-ratio genes, and C₄ proteins showed significantly lower protein-RPF ratios in Ftri compared to those in Frob (T-test, $P < 0.05$) (Fig. 4h). In contrast, photorespiratory proteins showed comparable protein-RPF ratios between the two species. This suggests that the observed decreased protein-RPF ratios for C₄ genes may largely be due to the increased transcriptional abundances rather than changes in translation efficiency.

Figure R1: Translation efficiency comparison between Frob and Ftri. (a) GC₃ comparisons of C₄ genes across five *Flaveria* species. As FramPEPC-*k1* is missing, its paralog (FramPEPC-*k2*, FramNA03444, see Extended Data Fig. 4) was utilized as a comparison for Fram in this context. (b) Abundances of ribosome protected fragment (RPF) of C₄ genes. (c) and (d) Distribution of RPF-to-RNA ratio and protein-to-RPF ratio. (e) Scatter plots of RPF versus RNA of Frob and Ftri. Low/high/moderate RPF-to-RNA genes are labeled in pink/light blue/grey. (f) Scatter plots of protein versus RPF of Frob and Ftri. Low/high/moderate protein-to-RPF genes were labeled in pink/light blue/grey. Low protein-to-RPF C₄ genes were indicated with red arrows. C₄ and photorespiratory genes are labeled in red and blue, and Pearson correlation (PC) between RPF vs. RNA, protein vs. RPF is shown in the parentheses on top of each panel in (e) and (f). (g) and (h) RPF-to-RNA ratio and protein-RPF ratio for C₄ genes and photorespiratory genes. Statistical significance was determined by two-tailed Wilcoxon test (* p < 0.05, ** p < 0.01, *** p < 0.001). This figure has been included as Figure 4 in the revised manuscript.

- The authors also performed long-read sequencing before in *Flaveria* species and studied C₄ evolution, which I believe is a necessary work to be considered in the introduction. Also here GRNs were made and associated with TF families. Paper: Lyu, A. et al. (2023). Evolution of gene regulatory network of C₄ photosynthesis in *Flaveria*. *Plant Commun.*, 4(1), 100426.

Response: We appreciate the reviewer's suggestion. We have incorporated a reference to our previous study in the introduction section of the manuscript. We have highlighted the importance of this work in understanding the regulatory mechanisms underlying C₄ photosynthesis and its evolutionary trajectory in *Flaveria* species.

Previous analysis of gene regulatory networks (GRNs) using long-read transcriptomic sequencing has provided important insights into the evolution of C₄ photosynthesis in the genus *Flaveria* (Lyu et al., 2022), which highlighted the significance of regulatory mechanisms in shaping the C₄ pathway across different *Flaveria* species. (Page 3, lines 77-80.)

Additionally, we have conducted a systematic comparison of gene regulatory networks (GRNs) constructed in our previous study and those updated in the current work. The GRNs reconstructed here (for *F. robusta*, *F. sonorensis*, *F. ramosissima*, and *F. trinervia*) were largely consistent with previously constructed GRNs that were annotated based on long-read sequencing (Lyu et al., 2022).

However, the number of regulated transcription factors (TFs) has increased due to improved TF annotations based on the latest genome assemblies (Supplementary note 15 in the revised manuscript).

- There are five *Flaveria* species sequenced, it could be of interest to specifically mention which are compared to the published genome assemblies (from line 132). It is written in the supplementary notes that a BLASTP E-value threshold of 0.001 was used to determine proteins to be common or not. This is I believe not a very stringent criterion, i.e. would it not be better so to set a high % identity of your protein sequence of your assembly? For instance if your BLASTP alignment only covers 50% of your protein but fulfilling your E value criterion – you consider this as a common protein?

Response: We appreciate the reviewer's thoughtful suggestions.

Our comparison of assembled genes was carried out between the *F. robusta* (C₃) genome assembly generated by Taniguchi et al. (2019) and our own assembly for the same species. We understand the concern regarding the BLASTP E-value threshold of 0.001, which might not be stringent enough to identify true orthologous relationships.

To address this issue, we have implemented an additional criterion based on coverage, where coverage is defined as the alignment length divided by the query sequence length, ensuring that the aligned regions cover a minimum percentage of the query sequence. Using this combined approach, we found that:

- 1) Under a coverage threshold of 70%, 94.5% of proteins from Taniguchi's assembly have counterparts in our assembly, and 85.1% of our assembly proteins have counterparts in Taniguchi's assembly.
- 2) Under a coverage threshold of 80%, 93.1% of proteins from Taniguchi's assembly have counterparts in our assembly, and 80% of our assembly proteins have counterparts in Taniguchi's assembly.
- 3) Under a coverage threshold of 90%, 90% of proteins from Taniguchi's assembly have counterparts in our assembly, and only 74.6% of our assembly proteins have counterparts in Taniguchi's assembly. (**Figure R2a**, the red frame)

Therefore, the new results supported that the annotated protein-coding genes in this study were improved compared to Taniguchi's assembly.

We also updated the protein length distribution in Fig.s4 b and c in Supplementary Note 4, considering a coverage threshold of 80% (**Figure R2** below)

Figure R2: Comparison of the annotated Frob protein from Taniguchi's assembly and our assembly.

Genes with counterparts in another assembly (either ours or Taniguchi's) were predicted. (a) statistics on the number of annotated protein-coding genes and those with counterparts in the other assembly. The table includes data for both assemblies, with the number of genes having ≥ 100 amino acids and the corresponding percentages of genes with counterparts in the other assembly at coverage thresholds of 70%, 80%, and 90%. (b) shows the distribution of protein lengths for genes from Taniguchi's assembly with (pink) and without (grey) counterparts in our assembly when considering a coverage threshold of 80%. (c) presents the distribution of protein lengths for genes from our assembly with (pink) and without (grey) orthologs in Taniguchi's assembly when applying a coverage threshold of 80%. (Note: [@] The number of protein-coding gene reported originally in Taniguchi's assembly (Taniguchi et al., 2021) is 46,138.). This figure has been included as Fig. s4 in Supplementary Note 4 in the revised manuscript.

- From line 185 it was described how high light induction was used to further verify the identified C₄ enzymes. Within Supplementary note 5, that was used to describe the high expression of these in the C₄ species, there was also a CO₂ treatment whose rationale is not really properly explained in the manuscript itself (although also included in Figure 2c). Could you not integrate the high light expression together with the CO₂ treatment here in a single plot, and explain these results better within the results section? You could describe the high light induction as a third criterion, i.e. (i) highest expression among paralogues, (ii) higher transcript abundance in C₄ species vs. C₃, and (iii) HL induction.

Response: We appreciate the reviewer's suggestion to integrate the high light induction data with the CO₂ treatment and to better explain these results within the results section. While we agree that integrating these data would provide a more cohesive narrative, we would like to clarify our approach and rationale for the experimental design.

In this study, we first predicted the functional copies of C₄ genes in each gene family, and then used high light induction to verify the predicted functional copy. As shown in our data, the functional copy of C₄ genes, such as *PEPC1*, gained greater light responsiveness during evolution, we also agree that it might be a potential criterion to identify novel C₄ related genes. In the revised manuscript, we added one sentence to describe this new criterion.

Our results also indicate that light responsiveness might be a potential criterion for identifying novel C₄-related genes. (Page 9, lines 184-185)

- Line 256 and onwards: There is a sudden strong focus on the role of ERF transcription factors in this chapter. The introduction does not mention ERFs, is there any known role of ERF in regulating C₄ enzymes? Did the expression of important ERF in the GRNs also increase from C₃ to C₄ species?

Response: We appreciate the reviewer bringing up this issue.

The rationale for focusing on ERF *cis*-regulatory elements (CREs) and ERF transcription factors (TFs) is based on the following observations:

- 1) The promoter sequences of C₄ genes from *Ftri*(C₄) showed enrichment in ERF *cis*-regulatory elements when compared to the whole genome sequence as a background. This enrichment was not observed in C₃-C₄ species (**Figure R3**).
- 2) The CREs in accessible chromatin regions associated with C₄ genes in *Ftri* were enriched in

ERF motifs.

- 3) More ERF TFs were involved in the regulation of C_4 genes in *Flaveria trinervia* (C_4) compared to C_3 and C_3 - C_4 species.
- 4) ERF CREs are widely enriched in plant genomes (Burgess et al., Plant Cell, 2019, doi: 10.1105/tpc.19.00078). Given that ERF TFs are involved in stress responses, and C_4 photosynthesis is an adaptive phenotype to environmental stresses such as low CO_2 , high light, and drought, the widespread presence of ERF *cis*-regulatory elements may provide a basis for rewiring gene regulation in response to environmental stress.

To our knowledge, there is no genetic evidence of the regulation of ERF transcription factors (TFs) on C_4 genes. However, recent work from one of our collaborations has shown that AP2/EREBP transcription factors, to which ERFs belong, play a major role in regulating the expression of C_4 genes in the grass C_4 species sugarcane (Hua et al., Plant Physiology, 2024, DOI: 10.1093/plphys/kiae455).

Our results suggest that the recruitment of ERF TFs to regulate C_4 genes is a consequence of acquiring ERF CREs in the promoter regions of C_4 genes. In addition, we observed that seven ERF TFs within the C_4 gene regulatory network (C_4 GRN) showed increased transcript abundance in Ftri (C_4) compared to Frob (C_3) (**Figure R4**). Although the remaining ERF TFs did not show increased expression in Ftri, they may still play a role in the regulation of C_4 genes, especially for those that already have high expression levels.

Frob (C_3)	Motif name	Ftri (C_4)	Motif name
	Ppas4 (q=0.05)		PHV(HB) (q=0.007)
	Hnf1 (q=0.049)		bZIP (q=0.006)
	ERF104 (ERF) (q=0.04)		CRF10 (ERF) (q=0.05)
Fson (C_3-C_4)			MYB65 (q=0.05)
	ANAC062 (NAC) (q=0.05)		PRDM15 (q=0.05)
	Cdx2 (homeobox) (q=0.04)		RAP26 (ERF) (q=0.05)
Fram (C_3-C_4)			MYB3R4 (q=0.05)
	GSC (Homeobox) (q=0.05)		KLF6 (q=0.05)
	PR(NR) (q=0.05)		At1G77640 (ERF) (q=0.05)

Figure R3: Enriched motif of protein sequences of C_4 genes and their orthologous. This figure presents the enriched *cis*-regulatory elements (CREs) of C_4 genes in Ftri (C_4) and their orthologous from C_3 and C_3 - C_4

species. The q-values associated with each motif were shown. ERF motifs are highlighted in pink. Note that counterparts of C₄ genes in Flin do not exhibit enriched motifs. This figure has been included as Extended Data Fig. 8 in the revised manuscript.

Figure R4: The change of transcript abundances of C₄TF between Ftri and Frob. Volcano plot shows differentially expressed C₄TFs of Ftri (C₄) between Ftri and Frob (C₃). Down-regulated and up-regulated TFs in Ftri compared to Frob are indicated in blue and red, respectively.

- Line 327: This jumps straight in a result. It would be beneficial to know that what sort of proteomics set-up was performed here in the text. It is in fact a thorough quantitative analysis, with six replicates per species – leading up to species-specific DDA spectral libraries and Spectronaut DIA quantitative analysis. It would be of value to showcase your experimental set-up, potentially with a small graphic included.

Response: We thank reviewer's suggestion. We give the brief description of experimental design for RNA-seq and proteomics study as following:

The increased transcript abundance of C₄ genes in C₄ species were well reported, we than investigated how the protein abundances were modified along evolution using proteomics. We performed proteomics measurement for the five species with 6 biological replicates for each (Extended

Data Fig.5a and b). To compare the protein level and transcript level of paralogs genes across different *Flaveria* species, we included the RNA-seq data from an previous work, in which RNA-seq data grown in the normal condition in low CO₂ experiments¹⁹, with 6 replicates for each species. We found that replicates from a same species showed higher correlations than those between species both based on transcript abundances of detected 27,684 genes and protein abundances of 4,908 detected proteins (Extended Data Fig.6a and b), implying the reliability of RNA and protein quantifications. (Page 12, line 243-251)

Besides, we add and illustrated graphic to show the process of proteomics in the revised manuscript, which is shown in **Figure R5**.

Figure R5: Workflow for protein quantification of five *Flaveria* species. Total proteins were extracted from leaf tissues of five *Flaveria* species and subjected to tryptic digestion. Six biological replicates were used for each species. The resulting peptides were fractionated via high-pH HPLC, followed by LC-MS/MS analysis. For data-dependent acquisition (DDA), a reference library comprising the protein sequences of the five *Flaveria* species was used. Data-independent acquisition (DIA) scans were set to a resolution of 120,000. DIA data were analyzed using Spectronaut software. False discovery rate (FDR) was controlled at 1% for both peptide and protein levels. iBAQ values, quantified based on each species' own protein reference, were utilized for quantitative analysis. Protein-to-RNA ratios (PTR) were calculated for each species. To compare the PTRs between different orthologs across the five *Flaveria* species, genes from the other four species were assigned to the orthologs of Frob. This figure has been included as Extended Data Fig.5 in the revised manuscript.

- It is highly advisable to use a more absolute quantitative measure for protein levels. Instead of protein LFQ values, it would be better to use iBAQ quantification values – see “Global quantification of mammalian gene expression control” (Schwanhäusser et al.). Spectronaut 14 contains iBAQ quantitation values as a feature. Protein LFQ values are definitely suited for relative protein quantification, i.e. the same protein across different conditions (treatments/conditions). But you are here in fact comparing the levels between proteins and species. Hence, comparison of iBAQ values (proteomics) to TPM values (RNA-seq) seems more sensible.

Response: We appreciate the reviewer’s comment on this matter. We have revised our quantification using iBAQ for all proteomics analyses. Generally, the iBAQ values are relatively smaller than LFQ values; however, the patterns across the five *Flaveria* species remain consistent based on both types of values (**Figure R6a**).

Furthermore, clustering analysis confirms that most C₄ genes are grouped in clusters where C₄ species show higher transcript abundances compared to C₃ and C₃-C₄ species.

The results of the protein-to-mRNA ratio (PTR) analysis are largely consistent, though there are slight changes in genes with low PTR and high PTR. Nonetheless, the C₄ species still exhibit significantly lower PTR for C₄ genes compared to C₃ and C₃-C₄ species.

We have revised Fig.3 (**Figure R6**) accordingly, and the corresponding supplementary figures have been revised in Supplementary Notes 11 and 12.

Figure R6: The C₄ species showed increased transcript abundances of C₄ genes. (a) RNA-seq and proteomics data for the C₄ genes in the five *Flaveria* species show increased transcript and protein abundances of C₄ genes in the C₄ species Ftri. *PEPC-k1* was excluded as the protein level of C₄ version of *PEPC-k1* could not be detected in any of these species. (b) The protein-to-mRNA ratio (PTR) distribution of genes across the five *Flaveria* species. High PTR and low PTR genes are defined as genes with PTR higher than the mean plus one standard deviation (SD) and with PTR values lower than the mean minus one SD respectively. Enriched function of conserved high PTR and low PTR genes across the five *Flaveria* species and their enriched function were shown. (c) Scatter plots of protein versus transcript abundance of the five *Flaveria* species. Low PTR and high PRT C₄ genes were indicated with arrows. Pearson correlation (PC) between protein abundance and

transcript abundance is shown in the parentheses on top of each panel. (d) PTR values for the C₄ gene set in the five *Flaveria* species, showing that C₄ genes have significantly lower PTRs in C₄ species *Ftri* than in the non-C₄ species. Note that no such decrease is shown for photorespiratory genes (except *Fson*), photosynthesis genes, or 100 times of resampling dataset (randomly choosing 14 genes from each species for each resampling). The statistical significance was determined by a one-way ANOVA procedure followed by a two-tailed Wilcoxon test, *P* values were adjusted with “BH” (* *p*<0.05, **0.05<*P*<0.01, ***0.01<*P*<0.001). (Abbreviations for the C₄ gene are the same as Fig.2). This figure has been included as Figure 3 in the revised manuscript.

- I am unsure how the final DIA quantifications were obtained. It mentions in the methods “As a result, five peptide libraries were obtained. Finally, data-independent acquisition (DIA) was performed using Spectronaut (version 14.7, Biognosys, Zurich, Switzerland).”. Does this mean the data of each species was analysed separately with Spectronaut (with its respective spectral library) and later on quantitative protein matrices were merged? This is not an easy task, as you would have to do this based on orthologous relationships between proteins and perhaps define certain orthologous protein groups.

Response: We appreciate the reviewer’s comments. The data for each species were analyzed separately using Spectronaut with their respective spectral libraries, and the quantitative protein matrices were later merged. To ensure accurate inter-species comparisons, we predicted orthologous relationships between *Frob* (C₃) and the other *Flaveria* species. The detailed method for inter-species comparisons has been added to the Methods section of the revised manuscript as follows:

For the inter-species comparison across different *Flaveria* species, orthologous gene pairs between the remaining four *Flaveria* species and *Frob* were predicted through blast (v2.2.31+) (Camacho et al., 2009), identifying the top hits with the E value threshold of 1e-5 and a sequence identity requirement of at least 60%. A K-means clustering analysis was then performed on the transcript abundances and protein abundance data separately, leveraging the unified *Frob* annotation. (Pages 32-33, lines 729-733)

- Related to this, how exactly did you generate Figure S29 as you are dealing with species-specific spectral libraries? Also for instance, in Fig S31 we see *Frob* protein identifiers (right of the heatmap) with quantifications in non-*Frob* species. Hence, this gives me the feeling a single species library was used for all species? If you search every species separately with Spectronaut (which seems the most correct to me), you could compare iBAQ values of the respective orthologs.

Response: We appreciate the reviewer's comments. To clarify, the data for each species were analyzed separately using Spectronaut with their respective spectral libraries. To compare protein abundances of orthologous genes, we obtained the orthologous relationship between Frob (C₃) and the other four *Flaveria* species (as mentioned above). The comparison leverages the unified Frob (C₃) annotation. Regarding Figure S29 and similar figures, such as Figure S31, where Frob protein identifiers appear with quantifications in non-Frob species, this is due to the orthologous mapping process. We identified orthologous genes between Frob (C₃) and the other species, and then mapped the protein abundances (iBAQ values) onto the annotation of Frob, allowing for inter-species comparisons.

- The data was submitted to PRIDE but no accession, citation and reviewer access is provided. With such a data-rich article it would be good to have an accessible datafile providing TPM and protein levels across species. I did see there was a figshare link but I had no access, I apologize if I overlooked something.

Response: We appreciate the reviewer's comments on this matter.

The mass spectrometry proteomics data have been deposited to the ProteomeXchange Consortium via the PRIDE (Perez-Riverol et al., 2019) partner repository with the dataset identifier PXD024720. Reviewer can access the dataset through username: reviewer_pxd024720@ebi.ac.uk, and password: M6E7Wz1M.

Additionally, the genome assembly, gene annotation, protein sequences, CDS sequences, gene expression data in TPM, Ribo-seq in TPM, and protein abundances in iBAQ, are available at the following Figshare link: <https://figshare.com/s/0a1a8f4fab6ae5315a14>

We apologize for any inconvenience caused by the previous lack of accessibility and hope these resources will facilitate further analysis and validation of our findings.

- Figure 4a: Heatmaps of z-scored quantitative values are displayed, it is not mentioned what these were in the legend. Is this TPM for RNA-seq and LFQ protein intensity for proteins? Please see my comment above on iBAQ values.

Response: We appreciate reviewer's comments. In the revised Figure, we have provided the information in the figure legend that the heatmaps display z-score normalized TPM values for RNA-seq and iBAQ values for protein. Please see revised figure in **Figure R6** above.

- Figure 4b: I strongly advise against such complex Venn diagrams, a more elegant solution could be UpSet plots, e.g. see the paper entitled “ ” by Lex et al., there are many interactive tools and R packages to do this.

Response: We appreciate the reviewer's advice. We have replaced the complex Venn diagram in Figure 4b with an UpSet plot, which is shown as followed as **Figure R7**.

Figure R7: Intersection of low PTR and high PTR protein across the five *Flaveria*. Upset plots illustrates the intersection of low/high PTR genes among the five *Flaveria* species. The x-axis represents the individual species, with the bar height indicating the number of low/high PTR genes in each species. The y-axis shows the number of intersected genes between different species combinations. Genes were assigned to orthologous to Frob for the other four species for inter-species comparison. This figure has been included as Extended Data Figs. 6d and e in the revised manuscript.

Minor comments

Line 80: Missing point.

Response: Thanks for pointing out this and the period is added in the revised manuscript.

Line 89: CRE not spelled out within text at first mention.

Response: We appreciate reviewer pointing out this issue. The full name of CRE was given at first mentioned.

.....which compromises their potential application in looking for C₄-related *cis*-regulate elements (CREs) (Page 4, line 83)

Figure 1: Panel c text font is too small to be readable.

Response: We appreciate reviewer for pointing out this issue. The font size in Fig1 c has been enlarged in the revised manuscript to ensure readability.

Supplementary Note 4: “exhibited no annotations sequences in the”: simply replace to were not annotated?

Response: Reviewer’s suggestion is adopted. The sentence was revised as following:

.....such as CA1, PEPC1, and NADP-ME4, were not annotated in the C₄ species *F. bidentis* (Supplementary note 4)

Supplementary Note 5: I deduce from the graphs that TPM values were used to estimate transcript abundance, which tool did you use to map/quantify? The methods do not state this. The figure quality of the alignments is highly variable in these Supplementary notes. The meaning of the statistics (*) that are not explained in the legend – these is differential expression according CO₂ treatment as in Figure 2 I assume?

Response: We appreciate reviewer for bringing these points to our attention. The method for RNA-seq quantification and differentially expressed gene identification were described in the Methods section, which was also shown as follows:

To quantify the expression level of *Flaveria* genes, raw reads were trimmed applying fastp (v0.20.0) (Chen et al., 2018) in default parameters, where reads were filtered if 40% bases were unqualified (phred quality<15). Transcript abundance of genes were calculated by mapping RNA-seq reads to the assembly genome sequence of corresponding species using RSEM (v1.3.3) (Li and Dewey,

2011), where STAR (v2.7.3a) (Dobin et al., 2013) was selected as the mapping tool. (Pages 31-32, lines 701-705)

We removed the RNA-seq data under low CO₂ conditions in the revised manuscript since this dataset was not the focus of the paper and did not add value to the text.

In the revised manuscript, we have improved the figure quality, ensuring consistent high-quality visuals throughout, particularly for the font of the axis labels. Additionally, we have clarified the meaning of the statistical symbols in the figure legends. Such as for Fig.3 in the revised manuscript:

The statistical significance was determined by a one-way ANOVA procedure followed by a two-tailed Wilcoxon rank sum test, *P* values were adjusted with “BH” (* *P* < 0.05, ** *P* < 0.01, *** *P* < 0.001) (Page 15, lines 291-293)

Line 265: 265 (ATAC-seq) experiments, also Tn5 hypersensitive site might need some more background for readers.

Response: We appreciate the reviewer’s input.

The sentence has been revised to provide additional context for Tn5 hypersensitivity in ATAC-seq experiments:

During ATAC-seq experiments, Tn5 transposase enzyme preferentially binds to nucleosomes with accessible DNA, generating sequencing tags that correspond to open chromatin. Therefore, Tn5 transposase-sensitive sites often exhibit peaks at gene transcription start sites. (Pages 18, lines 373-375)

Figure 3d and e: Currently many IGV genome views are provided, from genome to chromosome scale to million to scale (panel d) and for three genes of interest (e). Instead of these many visualizations, I feel like it would be better to dedicate more space simply to the three genome views of the individual genes – preferably with their tracks at the same y-axis scale.

Response: We appreciate for reviewer’s suggestion.

In the revised manuscript, we focused more on the genome views of individual genes of interest, ensuring that their tracks are at the same y-axis scale. We have selected representative photosynthetic genes for visualization, including: *RUBSC1b* (*Rubisco small subunit 1b*), *Lhca1b* (*light-harvesting complex 1 b*), *PGR5-like* (*proton gradient regulation 5-like*), *CA1* (*carbonic anhydrase 1*), *PPDK*

(*pyruvate orthophosphate dikinase*), *PPDK-RP* (*PPDK regulatory protein*), *PPT1* (*phosphate/phosphoenolpyruvate translocator 1*) and *NHDI* (*sodium: hydrogen antiporter*).

These revised visualizations are now presented in Extended Data Figure 9c of the revised manuscript.

Figure 3h: This means the CRE of the C₄ enzymes were connected to all possible matching TFs with ERFs in pink? The legend could give some more detailed explanation.

Response: We appreciate reviewer pointing out this issue.

To clarify, the lines in the figure represent predicted regulatory interactions between transcription factors (TFs) and C₄ genes, with ERFs highlighted in pink. We have revised the figure legend in the revised manuscript to provide a more detailed explanation:

Lines represent predicted regulatory interactions between TFs and C₄ genes. ERFs are highlighted in pink. (Revised legend of Fig. 5e)

Overall many typos in the supplementary notes, e.g. ‘vacumm-dried’, ‘CAN’ instead of ‘ACN’,..

Response: Thank you for pointing these out. We apologize for any confusion caused by the typographical errors in the supplementary notes. We have carefully reviewed and corrected all instances of “vacumm-dried” to “vacuum-dried”, “CAN” to “acetonitrile”, and other typos throughout the document.

Point-by-Point Responses

We are grateful to the reviewers for their insightful comments and constructive suggestions on our manuscript entitled “A dominant role of transcriptional regulation during the evolution of C₄ photosynthesis in *Flaveria* species” (Manuscript ID: NCOMMS-24-16595A). We have addressed all questions and concerns raised by the reviewers with new analyses and explanations, and believe the manuscript to be significantly improved, especially for the method for motif analysis.

Detailed responses to reviewers’ comments:

Reviewer #1: Comments and point-by-point responses (Pages 2~4)

Reviewer #2: Comments and point-by-point responses (Page 5)

Reviewer #3: Comments and point-by-point responses (Pages 6~27)

Reviewer #4: Comments and point-by-point responses (Page 28~29)

Reviewer #5: Comments and point-by-point responses (Page 30)

Reviewer #1:

Overall, I find the revised manuscript to be much improved. The conclusions are clearer and have sufficient support from the data. Likewise, additional figures and modifications to existing figures make the results easier to understand to a broader range of readers. Specifically, my concerns regarding the role of gene duplication and Ks analysis have been dealt with, as have my concerns about claims regarding the effects of gene duplication. Revisions to results associated with light response in C₄ and C₃-C₄ species is also satisfactory. The results for ATACseq analysis are much clearer as are methods relating to TF analysis and gene expression. In general, I believe that this version is suitable for publication in Nature Communications pending some basic editing for grammar.

Response: We are deeply grateful to the reviewer for his/her considerate and positive review, along with insightful questions and suggestions for these two rounds of revisions. These inputs are immensely valuable to us in enhancing our manuscript. Please find our Point-by-Point Responses below.

Q1: “The reason we do not delve deeply into C₃-C₄ species is that we have found that C₃-C₄ species may not necessarily be transitional between C₃ and C₄ species but rather alternative outcomes.” I agree that viewing C₃+C₄ as a transitional state is problematic and that in most cases it more likely represents an alternative outcome to C₄. I think now that my primary confusion emerged from the description of C₃+C₄ plants as intermediate but realize that “intermediate” is not necessarily equivalent to transitional. I appreciate the authors’ well-considered reply to my comment in their response but I still feel that this could be made clearer in the introduction. Statements such as “facilitating the progression from C₃-C₄ intermediate to a full C₄ state.” And “Therefore, the transition from C₄-like species to C₄ ones is regarded as a fine-tuning process” still give the impression that extant C₃+C₄ lineages are potentially species in transition unless it is clearly stated otherwise.

Response: We are grateful for reviewer’s positive assessment and constructive feedback on our revised manuscript. Regarding the reviewer’s comment on the description of C₃-C₄ species, we acknowledge the concern that certain phrases might unintentionally imply that extant C₃-C₄ lineages are transitional forms. To address this, we have modified the Introduction (Page 3, lines 71-73) and Discussion (Page 23, lines 482-483) to explicitly state that C₃-C₄ species should not be viewed as necessarily representing an evolutionary intermediate stage between C₃ and C₄ photosynthesis. Instead, we emphasize that these species might represent alternative evolutionary outcomes within the diversity of photosynthetic pathways.

Specifically, the revised Introduction (Page 3, lines 71–73) now states: “Note that the term intermediate species does not necessarily refer to transitional forms, but may instead represent alternative evolutionary outcomes within the spectrum of photosynthetic strategies”.

In the Discussion, the original statement — “facilitating the progression from C₃-C₄ intermediate to a full C₄ state” has been revised to “We propose that it might be a beneficial event facilitating the progression from the C₃-C₄ intermediate to a full C₄ state, without implying an inevitable progression towards a full C₄ photosynthetic pathway.” (Page 23, lines 480-483)

In addition, we have removed the original statement of “Therefore, the transition from C₄-like species to C₄ ones is regarded as a fine-tuning process” in the Introduction to avoid potential misinterpretation.

Q2: The authors state in their response: “The terms “C₄ lineages” and “C₄-like” are not synonymous. Within the *Flaveria* genus, *F. brownii* from Clade B is considered a C₄-like species because approximately 12% of CO₂ is still fixed directly by Rubisco.” I think my confusion arose from the fact while C₄-like plants are described in the intro they are not included in this study.

Response: We thank reviewer’s comments. We appreciate the opportunity to clarify any confusion regarding our terminology. The statement “Therefore, the transition from C₄-like species to C₄ ones is regarded as a fine-tuning process” has been removed from the revised manuscript to avoid any potential misinterpretation implying that C₄-like species represent transitional forms.

Q3: Also, if the authors plan to conduct follow-up targeted studies on C₃ + C₄ species (or that further study is merited in these species) this could be indicated in the discussion.

Response: We thank the reviewer for the recommendation to include potential follow-up studies on C₃-C₄ species in the Discussion. Accordingly, we have proposed potential avenues for further research on C₃-C₄ species in the Discussion (Page 25, lines 538-544), which was also shown as follows:

“Moreover, this study provides a wealth of data that can serve as a foundation to explore the genomic features and evolutionary stages of different intermediate species within the *Flaveria* genus. For instance, our comprehensive dataset allows detailed comparisons between C₃-C₄ species from clade A (Fson and Fram) and clade B (Flin) of this genus. Such analyses may uncover the mechanisms underlying the absence of true C₄ photosynthesis in clade B, thereby providing deeper insights into the evolutionary dynamics and genetic factors that influence photosynthetic pathway development.” (Page 25, lines 538-544)

Q4: Still some typos:

Line 83: cis-regulate should be cis-regulatory

Response: We appreciate reviewer pointing out this typo and have corrected accordingly in the revised manuscript.

Line 361: “Precited” should be “predicted”

Response: We appreciate reviewer pointing out this typo and have corrected accordingly in the revised manuscript.

Line 376 - 377: “exhibited a peak at the upstream of gene transcription” can remove “at the”

Response: We appreciate reviewer pointing out this error. The sentence was corrected accordingly in the revised manuscript.

In addition to addressing the above typos, we have thoroughly reviewed the manuscript, corrected previously overlooked mislabeling and typographical errors, and revised the entire text to enhance the clarity of all results.

Reviewer #2:

Thank you to the authors for thoroughly addressing my comments. However, I noticed an additional issue: the genome annotation GFF3 file is not available at https://db.cngb.org/codeplot/datasets/public_dataset?id=flaveria

Response: We deeply appreciate the reviewer's thoughtful and positive review, as well as the insightful questions and suggestions provided throughout these two rounds of revisions. These inputs have been invaluable in improving the quality of our manuscript. We sincerely thank the reviewer for the thorough review and for identifying the issue with the genome annotation GFF3 file. The CNSA repository has been updated under the same project ID to include both GFF3 and GTF format files for genome annotations.

These files are now readily accessible to facilitate further research utilizing the valuable genomic and genome resources of the *Flaveria* genus (**Table R1**). For the reviewer's convenience, we have included a snapshot of the updated page for *F. robusta*, as shown in **Figure R1**.

Table R1. The specific links for accessing genome assembly and gene annotation

Species	Data link:
F. robusta	https://ftp.cngb.org/pub/CNSA/data2/CNP0003058/CNS0557296/CNA0050661/
F. sonorensi	https://ftp.cngb.org/pub/CNSA/data2/CNP0003058/CNS0557297/CNA0050662/
F. linearis	https://ftp.cngb.org/pub/CNSA/data2/CNP0003058/CNS0557298/CNA0050663/
F. ramosissima	https://ftp.cngb.org/pub/CNSA/data2/CNP0003058/CNS0557299/CNA0050664/
F. trinervia	https://ftp.cngb.org/pub/CNSA/data2/CNP0003058/CNS0557300/CNA0050665/

CNSA FTP公开服务					download
当前文件夹: / pub / CNSA / data2 / CNP0003058 / CNS0557296 / #CNA0050661					
名称	大小	更新时间	Aspera 高速下载	复制下载命令	直接下载
Frob.chr_transcript.order.gff3	38.64MB	2024-07-30 06:19:40			
Frob.chr_transcript.order.gtf	48.65MB	2024-11-22 07:23:36			
Frobusta_geome.fa.gz	139.08MB	2022-05-28 02:28:37			
Frobusta_geome.fa.gz.md5	53B	2024-04-17 16:58:54			

Figure R1. A snapshot of genome assembly and gene annotation files for *F. robusta* in CNGB

Reviewer #3:

Q1: While the authors have addressed some of my concerns, the revised manuscript contains language that reveals a worrying lack of expertise in this field. For example, in line 373, the authors state "... the Tn5 transposase enzyme preferentially binds nucleosomes with accessible DNA...". This is patently incorrect. At the concentration used for most ATAC-seq experiments, Tn5 has an incredibly strong preference for nucleosome-free regions, and cannot bind nucleosomes or nucleosome-bound DNA. An additional concern is that the results are seemingly unchanged following the reanalysis of a totally new set of peaks. To confirm that the results in the present manuscript are from the new peak set, I ask that the authors contrast motif enrichment scores in (1) all ACRs and (2) ACRs near C_4 genes compared to background/control sequences (non ACRs) for both the original peak set and the updated peak set for all tested motifs (highlighting the scores for ERF family TFs). Please indicate how the background/control sequences were selected, as well as their composition (GC content and genomic distribution) and include this information in the methods sections. For motifs that are enriched in ACRs close to C_4 genes, the background for this analysis should be the remaining set of ACRs (non- C_4 gene ACRs).

Response: We sincerely thank the reviewer for his/her thoughtful and favorable review, as well as critical questions and suggestions for revision. In this revision, we conducted additional analyses to address the reviewer's concerns. Please find our Point-by-Point Responses below.

We apologize for the misleading statement regarding the binding preferences of the Tn5 transposase enzyme. The relevant sentences have been corrected in the revised manuscript (page 18, lines 384-385), which was also shown as follows:

"During ATAC-seq experiments, the Tn5 transposase enzyme shows a strong preferential binding to nucleosome-free DNA regions".

We also apologize for the unclear description of the motif analysis, which may have caused confusion for the reviewer. To clarify this process, we have created a graph to illustrate the steps of motif analysis as shown in **Figure R1**. Obtaining high-quality ATAC-seq data is challenging, particularly for non-model species such as those in the *Flaveria* genus. In this study, we only obtain ATAC-seq data from C_4 species (Ftri). Significant efforts were made to conduct ATAC-seq experiments on other *Flaveria* species, but we were unable to obtain data of comparable quality to that of the C_4 species *F. trinervia*. Therefore, we initially analyzed the enriched motifs in the promoters of C_4 genes (or their counterparts) compared to non- C_4 gene promoters (background) in each species using the HOMER package. This analysis revealed that the promoters of C_4 genes were enriched with more motifs belonging to ERF TF families (annotated as AP2EREB in HOMER) in the C_4 species *F. trinervia* compared to their counterparts in C_3 and C_3 - C_4 species, suggesting a potential role for ERF TF families in regulating C_4 genes. We then utilized ATAC-seq data from the C_4 species *F. trinervia* to investigate whether open chromatin regions also contained ERF motifs in this species. After peak calling using ATAC-seq data, we scanned known plant motifs in the peak regions to determine whether ERF motifs were associated with C_4 genes. The plant-specific motifs

were obtained from the PlantPAN 3.0 database. A total of 1,471,751 occurrences of 277 motifs (from 28 families) were predicted in all peaks using FIMO with a q-value threshold of 0.05. Among these, 1,858 occurrences of 117 motifs (from 14 families) were identified in the peak regions near C₄ genes (including the regions 3 kb upstream start codons, 3kb downstream of stop codons, and the gene bodies). To further identify motifs associated with C₄ genes beyond random expectation, we conducted a Monte Carlo permutation test. Specifically, for each of the 117 motifs, we compared the observed occurrences with the expected occurrences derived from 1,000 permutations. In each permutation, 1,858 motif occurrences were randomly selected from the total 1,471,751 occurrences of 277 motifs, and the expected occurrences of each motif were documented. After 1,000 permutations, the enrichment p-value for each motif was calculated based on the number of permutations in which the motif occurrences exceeded the observed occurrences. The Benjamini and Hochberg correction was further applied to obtain the FDR value of each motif. As a result, 22 motifs from 10 families were identified with FDR <0.05, with the ERF family dominating the enriched motifs associated with C₄ genes.

Figure R1. Pipeline for enriched motif analysis in ACRs near C₄ genes. Left Panel: Predict enriched motifs in the promoters of C₄ genes in each of the five *Flaveria* species using HOMER. The promoters of non-C₄ genes were used as the background in each species. Right Panel: Based on ATAC-seq for the C₄ species *Flaveria trinervia* (Ftri), after peaks calling, known plant motifs within peaks were obtained using FIMO with a q-value threshold of 0.05. The enriched motifs associated with C₄ genes were identified using 1,000 permutation tests under a false discovery rate (FDR) 0.05.

In response to reviewer’s concern on whether the motif analysis was from new peaks, we performed comparison of enriched motifs between updated and original peaks.

First of all, by comparing the genomic distributions of updated and original peaks, we found a higher percentage of peaks associated with gene promoters in the updated peaks compared to original peaks (**Figure R1**), indicating an improvement in peak quality.

Figure R1. Genomic distribution of updated and original peaks

The pie charts illustrate the distribution of peaks across different genomic regions of (a) 14,333 peaks from updated peak analysis, and (b) 166,785 peaks from previous peak analysis.

Regarding the enriched motifs within all ACRs, we predicted enriched motifs of ACRs based on both updated peaks and original peaks using HOMER package. We used all ACRs as the input, and all the chromosomes and scaffolds of genomes were used as background for motif enrichment analysis. We agree with the reviewer that the background sequence is of importance for motif analysis. We selected HOMER package for motif enrichment finding analysis because its algorithm applies two types of normalization to account for the sequence composition bias between the input and background sequences. Regarding GC composition bias, HOMER package automatically selects background regions from the provided background sequences that match the GC-content distribution of the input sequences (in 5% increments), as described in the HOMER manual (<http://homer.ucsd.edu/homer/ngs/peakMotifs.html>). Specifically, if the input is highly GC-rich, HOMER selects random regions from GC-rich regions of the background as a control. In addition to accounting for GC-content bias, HOMER package also applies “autonormalization of sequence bias” to remove bias introduced by lower-order oligo sequences. HOMER package works by assuming the input and background should not have an imbalance in 1-mers, 2-mers, 3-mers, etc, and after calculating the imbalances for each oligo, it adjusts the weights of each background sequence by a small amount to help normalize any imbalance. This analytical procedure ensures that the motif enrichment analysis accounts for potential sequence composition biases between the input and background.

We predicted 416 enriched motifs from the updated peaks (14,333), and 480 motifs from the original peaks (166,785) using a “BH” adjusted P-value threshold of 0.05 (**Table R1**). Among these, 328 motifs were shared between updated peaks and original peaks (**Table R2**), and 30 of them are ERF motifs (annotated as AP2EREB in HOMER). The shared motifs were showed in **Table R2**,

motifs uniquely enriched in the updated peaks are shown in **Table R3**, and those uniquely enriched in the original peaks are presented in **Table R4**.

In addition to the analysis based on HOMER package, we also compared the enriched motifs between updated peaks and original peaks using dinucleotide shuffled background sequences. For this analysis, we generated dinucleotide-shuffled background sequences for both the updated and original peaks independently using BiasAway [1]. For each sequence, we generated five dinucleotide shuffled sequences, resulting in a background dataset five times of the ACRs set (shuffled ACR sequences). Based on the known plant motifs downloaded from Plantpan 3.0, we used FIMO to identify the occurrences of those motifs with a q-value threshold of 0.05 on ACR sequences and shuffled ACR sequences, respectively. We then used Fisher’s exact test to identify enriched motifs on ACR sequences by using the occurrences of motifs from the shuffled ACR sequences as the background. The results revealed 173 enriched motifs (including 104 ERF motifs) were identified from the updated peaks (**Table R5**), and 153 enriched motifs (including 80 ERF motifs) were identified from the original peaks (**Table R6**).

Regarding the enriched motifs from ACRs near C_4 genes, we used the remaining set of ACRs as the background as required by the reviewer. Based on the known motifs of plants downloaded from Plantpan 3.0, we used FIMO to identify the occurrences of motifs of all ACRs with a q-value threshold of 0.05, and then used Fisher’s exact test to identify enriched motifs near C_4 genes (within 3 kb of the gene body region) by using the occurrences of motifs from the remaining set of ACRs as the background. The results revealed 19 enriched motifs from 8 TF families identified from the updated peaks, including 10 enriched motifs from ERF TF families, and 15 enriched motifs from 6 TF families identified from the original peaks, including 7 enriched motifs from ERF TF families. All 6 enriched motif families from original peaks, except for LBD TF family, were shared in the enriched motif families from updated peaks (**Table R7** and **Table R8**). Note that 8 out of top 10 enriched motifs on updated peaks were from ERF TF family, and the top 5 enriched motifs were all ERF motifs (**Table R7**). Since the updated peaks exhibited improved peak quality, these results further underscored the importance of ERF TF families in regulating C_4 genes.

Although less C_4 -genes associated peaks were observed in updated peaks than in original peaks, the length of C_4 -genes associated peaks is longer in updated peaks than in original peaks (**Table R9**), this may explain more C_4 -genes associated motifs were observed in updated peaks than in original peaks (**Table R7** and **Table R8**).

Altogether, the observed results showed the differences of enriched motifs between updated peaks and original peaks and demonstrated that enriched motifs from the ERF family were identified in both the updated and original peaks for all ACRs and ACRs near C_4 genes.

Table R1. Statistics of comparison of enriched motifs based on HOMER from updated peaks and original peaks

	No. All motifs	No. ERF (AP2EREB) motifs
Shared enriched motifs	328	30
Specific in updated peaks (14,333 peaks)	88	0
Specific in original peaks (166,785 peaks)	152	28

Table R5. Enriched motifs based on updated peaks using Fisher's exact test (shuffled sequences of original peak regions as background)

Index	Motif	TF family	Total background (5-fold shuffled peak sequence)	# Motif in back-ground	# Total motif in all peaks	# Motif in all peak	Pvalue (Fisher.test)	Oddratio	Adjusted Pvalue(BH)
1	TfmatrixID_0656	ERF	248949	0	1471751	9221	0.00E+00	Inf	0.00E+00
2	TfmatrixID_0663	ERF	248949	0	1471751	15859	0.00E+00	Inf	0.00E+00
3	TfmatrixID_0671	ERF	248949	0	1471751	13795	0.00E+00	Inf	0.00E+00
4	TfmatrixID_0677	ERF	248949	0	1471751	23425	0.00E+00	Inf	0.00E+00
5	TfmatrixID_0686	ERF	248949	0	1471751	17829	0.00E+00	Inf	0.00E+00
6	TfmatrixID_0692	ERF	248949	0	1471751	57091	0.00E+00	Inf	0.00E+00
7	TfmatrixID_0698	ERF	248949	0	1471751	24122	0.00E+00	Inf	0.00E+00
8	TfmatrixID_0704	ERF	248949	0	1471751	9574	0.00E+00	Inf	0.00E+00
9	TfmatrixID_0707	ERF	248949	0	1471751	5788	0.00E+00	Inf	0.00E+00
10	TfmatrixID_0715	ERF	248949	0	1471751	4849	0.00E+00	Inf	0.00E+00
11	TfmatrixID_0724	ERF	248949	0	1471751	15570	0.00E+00	Inf	0.00E+00
12	TfmatrixID_0725	ERF	248949	0	1471751	16861	0.00E+00	Inf	0.00E+00
13	TfmatrixID_0730	ERF	248949	0	1471751	5703	0.00E+00	Inf	0.00E+00
14	TfmatrixID_0731	ERF	248949	0	1471751	6689	0.00E+00	Inf	0.00E+00
15	TfmatrixID_0733	ERF	248949	0	1471751	6954	0.00E+00	Inf	0.00E+00
16	TfmatrixID_0737	ERF	248949	0	1471751	11606	0.00E+00	Inf	0.00E+00
17	TfmatrixID_0742	ERF	248949	0	1471751	16793	0.00E+00	Inf	0.00E+00
18	TfmatrixID_0748	ERF	248949	0	1471751	36710	0.00E+00	Inf	0.00E+00
19	TfmatrixID_0753	ERF	248949	0	1471751	16180	0.00E+00	Inf	0.00E+00
20	TfmatrixID_0759	ERF	248949	0	1471751	9251	0.00E+00	Inf	0.00E+00
21	TfmatrixID_0765	ERF	248949	0	1471751	38752	0.00E+00	Inf	0.00E+00
22	TfmatrixID_0766	ERF	248949	0	1471751	10286	0.00E+00	Inf	0.00E+00
23	TfmatrixID_0767	ERF	248949	0	1471751	17819	0.00E+00	Inf	0.00E+00
24	TfmatrixID_0770	ERF	248949	0	1471751	44012	0.00E+00	Inf	0.00E+00
25	TfmatrixID_0775	ERF	248949	0	1471751	22956	0.00E+00	Inf	0.00E+00
26	TfmatrixID_0777	ERF	248949	0	1471751	34203	0.00E+00	Inf	0.00E+00
27	TfmatrixID_0782	ERF	248949	0	1471751	24802	0.00E+00	Inf	0.00E+00
28	TfmatrixID_1122	LBD	248949	0	1471751	11608	0.00E+00	Inf	0.00E+00
29	TfmatrixID_1124	LBD	248949	0	1471751	5307	0.00E+00	Inf	0.00E+00
30	TfmatrixID_1126	LBD	248949	0	1471751	8200	0.00E+00	Inf	0.00E+00
31	TfmatrixID_1215	MYB	248949	0	1471751	6191	0.00E+00	Inf	0.00E+00
32	TfmatrixID_0664	ERF	248949	1	1471751	51305	0.00E+00	7208.01	0.00E+00
33	TfmatrixID_0659	ERF	248949	1	1471751	31393	0.00E+00	6129.49	0.00E+00
34	TfmatrixID_0660	ERF	248949	1	1471751	25454	0.00E+00	4661.72	0.00E+00
35	TfmatrixID_0743	ERF	248949	2	1471751	47391	0.00E+00	4189.73	0.00E+00
36	TfmatrixID_0719	ERF	248949	2	1471751	40625	0.00E+00	3481.23	0.00E+00
37	TfmatrixID_0771	ERF	248949	1	1471751	16826	0.00E+00	2846.11	0.00E+00
38	TfmatrixID_0734	ERF	248949	1	1471751	16675	0.00E+00	2820.57	0.00E+00
39	TfmatrixID_0756	ERF	248949	1	1471751	15698	0.00E+00	2655.31	0.00E+00
40	TfmatrixID_0726	ERF	248949	1	1471751	15649	0.00E+00	2647.02	0.00E+00
41	TfmatrixID_0739	ERF	248949	1	1471751	14269	0.00E+00	2413.59	0.00E+00
42	TfmatrixID_0750	ERF	248949	2	1471751	20780	0.00E+00	1757.45	0.00E+00
43	TfmatrixID_0760	ERF	248949	3	1471751	27671	0.00E+00	1560.16	0.00E+00
44	TfmatrixID_0717	ERF	248949	1	1471751	8835	0.00E+00	1494.42	0.00E+00
45	TfmatrixID_0745	ERF	248949	1	1471751	7170	0.00E+00	1212.79	0.00E+00
46	TfmatrixID_0705	ERF	248949	3	1471751	20948	0.00E+00	1181.09	0.00E+00
47	TfmatrixID_0772	ERF	248949	4	1471751	27689	0.00E+00	1170.87	0.00E+00
48	TfmatrixID_0735	ERF	248949	1	1471751	6690	0.00E+00	1131.59	0.00E+00
49	TfmatrixID_0713	ERF	248949	3	1471751	18776	0.00E+00	1058.63	0.00E+00
50	TfmatrixID_0716	ERF	248949	2	1471751	10598	0.00E+00	896.30	0.00E+00
51	TfmatrixID_0973	C2H2	248949	7	1471751	33763	0.00E+00	815.83	0.00E+00
52	TfmatrixID_0755	ERF	248949	2	1471751	8466	0.00E+00	715.99	0.00E+00
53	TfmatrixID_0728	ERF	248949	3	1471751	9549	0.00E+00	538.38	0.00E+00
54	TfmatrixID_0749	ERF	248949	2	1471751	5435	0.00E+00	459.64	0.00E+00
55	TfmatrixID_0757	ERF	248949	6	1471751	15590	0.00E+00	439.48	0.00E+00
56	TfmatrixID_1214	MYB	248949	14	1471751	28474	0.00E+00	343.99	0.00E+00
57	TfmatrixID_0894	Dof	248949	4	1471751	7485	0.00E+00	316.49	0.00E+00
58	TfmatrixID_0901	Dof	248949	11	1471751	6016	0.00E+00	92.48	0.00E+00
59	TfmatrixID_0912	Dof	248949	25	1471751	12476	0.00E+00	84.38	0.00E+00
60	TfmatrixID_0915	Dof	248949	61	1471751	28903	0.00E+00	80.11	0.00E+00
61	TfmatrixID_1199	MYB	248949	4	1471751	4640	2.91E-305	196.19	1.32E-304
62	TfmatrixID_0764	ERF	248949	2	1471751	4288	4.42E-286	362.63	1.97E-285
63	TfmatrixID_0741	ERF	248949	2	1471751	4150	9.09E-277	350.96	4.00E-276
64	TfmatrixID_0706	ERF	248949	0	1471751	3956	6.86E-269	Inf	2.97E-268
65	TfmatrixID_0761	ERF	248949	0	1471751	3708	4.25E-252	Inf	1.81E-251
66	TfmatrixID_0774	ERF	248949	0	1471751	3589	4.86E-244	Inf	2.04E-243
67	TfmatrixID_0781	ERF	248949	0	1471751	3557	7.13E-242	Inf	2.95E-241
68	TfmatrixID_0699	ERF	248949	0	1471751	3463	1.66E-235	Inf	6.75E-235
69	TfmatrixID_1197	MYB	248949	3	1471751	3369	7.44E-222	189.93	2.99E-221
70	TfmatrixID_0654	ERF	248949	2	1471751	3265	4.75E-217	276.11	1.88E-216
71	TfmatrixID_0672	ERF	248949	0	1471751	2894	5.76E-197	Inf	2.25E-196
72	TfmatrixID_0722	ERF	248949	0	1471751	2860	1.16E-194	Inf	4.46E-194
73	TfmatrixID_0066	ERF	248949	0	1471751	2734	3.99E-186	Inf	1.51E-185
74	TfmatrixID_0685	ERF	248949	0	1471751	2731	6.37E-186	Inf	2.38E-185
75	TfmatrixID_0083	ERF	248949	0	1471751	2717	5.66E-185	Inf	2.09E-184
76	TfmatrixID_0723	ERF	248949	2	1471751	2270	5.98E-150	191.96	2.18E-149
77	TfmatrixID_0888	Dof	248949	9	1471751	2447	2.76E-149	45.96	9.94E-149
78	TfmatrixID_0676	ERF	248949	0	1471751	2157	5.03E-147	Inf	1.79E-146
79	TfmatrixID_0675	ERF	248949	1	1471751	2183	2.75E-146	369.23	9.66E-146
80	TfmatrixID_0688	ERF	248949	0	1471751	2041	3.67E-139	Inf	1.27E-138

81	TFmatrixID_1146	MYB	161720	4	2508171	3103	3.72E-77	50.01	1.02E-76
82	TFmatrixID_1094	WOX	161720	0	2508171	2814	4.78E-77	Inf	1.30E-76
83	TFmatrixID_0715	ERF	161720	0	2508171	2804	8.93E-77	Inf	2.40E-76
84	TFmatrixID_1144	MYB	161720	3	2508171	2962	4.55E-75	63.66	1.21E-74
85	TFmatrixID_0706	ERF	161720	1	2508171	2777	8.14E-74	179.05	2.14E-73
86	TFmatrixID_0774	ERF	161720	0	2508171	2666	4.92E-73	Inf	1.27E-72
87	TFmatrixID_1148	MYB	161720	0	2508171	2623	7.20E-72	Inf	1.85E-71
88	TFmatrixID_1183	MYB	161720	2	2508171	2770	1.06E-71	89.30	2.69E-71
89	TFmatrixID_0983	C2H2	161720	0	2508171	2560	3.67E-70	Inf	9.21E-70
90	TFmatrixID_0895	Dof	161720	3	2508171	2219	2.67E-55	47.69	6.62E-55
91	TFmatrixID_0648	B3	161720	174	2508171	7331	1.30E-53	2.72	3.19E-53
92	TFmatrixID_1156	MYB	161720	2	2508171	1958	5.51E-50	63.12	1.34E-49
93	TFmatrixID_0708	ERF	161720	1	2508171	1889	6.61E-50	121.79	1.58E-49
94	TFmatrixID_0722	ERF	161720	2	2508171	1909	1.12E-48	61.54	2.65E-48
95	TFmatrixID_1196	MYB	161720	3	2508171	1883	2.11E-46	40.47	4.96E-46
96	TFmatrixID_0909	Dof	161720	4	2508171	1929	3.79E-46	31.09	8.81E-46
97	TFmatrixID_1205	MYB	161720	0	2508171	1627	7.29E-45	Inf	1.68E-44
98	TFmatrixID_0657	ERF	161720	0	2508171	1604	3.07E-44	Inf	6.98E-44
99	TFmatrixID_0687	ERF	161720	2	2508171	1728	7.42E-44	55.70	1.67E-43
100	TFmatrixID_0735	ERF	161720	0	2508171	1497	2.45E-41	Inf	5.46E-41
101	TFmatrixID_1299	NAC	161720	0	2508171	1494	2.95E-41	Inf	6.52E-41
102	TFmatrixID_0685	ERF	161720	1	2508171	1545	1.15E-40	99.61	2.52E-40
103	TFmatrixID_0679	ERF	161720	0	2508171	1385	2.67E-38	Inf	5.78E-38
104	TFmatrixID_0721	ERF	161720	2	2508171	1518	2.85E-38	48.93	6.10E-38
105	TFmatrixID_1210	MYB	161720	0	2508171	1356	1.63E-37	Inf	3.47E-37
106	TFmatrixID_0916	Dof	161720	0	2508171	1355	1.74E-37	Inf	3.65E-37
107	TFmatrixID_0889	Dof	161720	6	2508171	1668	8.89E-37	17.92	1.85E-36
108	TFmatrixID_1378	B3	161720	45	2508171	2941	4.94E-35	4.21	1.02E-34
109	TFmatrixID_0919	Dof	161720	3	2508171	1447	6.45E-35	31.10	1.32E-34
110	TFmatrixID_0672	ERF	161720	1	2508171	1284	1.15E-33	82.78	2.34E-33
111	TFmatrixID_1124	LBD	161720	0	2508171	1148	7.15E-32	Inf	1.44E-31
112	TFmatrixID_1259	MYB_related	161720	0	2508171	1133	1.82E-31	Inf	3.63E-31
113	TFmatrixID_0918	Dof	161720	3	2508171	1312	2.22E-31	28.19	4.37E-31
114	TFmatrixID_1397	SRS	161720	1	2508171	1051	1.98E-27	67.76	3.87E-27
115	TFmatrixID_0680	ERF	161720	1	2508171	1010	2.46E-26	65.12	4.77E-26
116	TFmatrixID_0893	Dof	161720	0	2508171	883	1.10E-24	Inf	2.12E-24
117	TFmatrixID_0999	CPP	161720	0	2508171	855	6.34E-24	Inf	1.21E-23
118	TFmatrixID_0676	ERF	161720	0	2508171	777	8.28E-22	Inf	1.57E-21
119	TFmatrixID_1192	MYB	161720	4	2508171	911	8.02E-20	14.68	1.50E-19
120	TFmatrixID_1189	MYB	161720	0	2508171	702	8.97E-20	Inf	1.67E-19
121	TFmatrixID_1245	MYB_related	161720	0	2508171	679	3.77E-19	Inf	6.96E-19
122	TFmatrixID_0690	ERF	161720	0	2508171	625	1.10E-17	Inf	2.01E-17
123	TFmatrixID_0658	ERF	161720	0	2508171	621	1.41E-17	Inf	2.56E-17
124	TFmatrixID_0744	ERF	161720	2	2508171	680	3.16E-16	21.92	5.68E-16
125	TFmatrixID_0688	ERF	161720	0	2508171	549	1.27E-15	Inf	2.27E-15
126	TFmatrixID_1238	MYB_related	161720	0	2508171	512	1.28E-14	Inf	2.27E-14
127	TFmatrixID_1346	NAC	161720	0	2508171	479	1.01E-13	Inf	1.77E-13
128	TFmatrixID_1265	MYB_related	161720	0	2508171	476	1.21E-13	Inf	2.12E-13
129	TFmatrixID_1370	Nin-like	161720	722	2508171	14638	1.69E-13	1.31	2.92E-13
130	TFmatrixID_1187	MYB	161720	0	2508171	460	3.30E-13	Inf	5.66E-13
131	TFmatrixID_0675	ERF	161720	1	2508171	513	3.86E-13	33.07	6.57E-13
132	TFmatrixID_0763	ERF	161720	2	2508171	509	7.85E-12	16.41	1.33E-11
133	TFmatrixID_1276	NAC	161720	0	2508171	375	6.68E-11	Inf	1.12E-10
134	TFmatrixID_0674	ERF	161720	0	2508171	276	3.24E-08	Inf	5.39E-08
135	TFmatrixID_0796	BBR-BPC	161720	805	2508171	15036	7.68E-08	1.20	1.27E-07
136	TFmatrixID_0681	ERF	161720	3	2508171	387	7.77E-08	8.32	1.27E-07
137	TFmatrixID_1398	SRS	161720	0	2508171	261	8.27E-08	Inf	1.35E-07
138	TFmatrixID_0699	ERF	161720	2	2508171	342	1.24E-07	11.03	2.01E-07
139	TFmatrixID_1271	MYB_related	161720	0	2508171	246	2.11E-07	Inf	3.39E-07
140	TFmatrixID_0712	ERF	161720	2	2508171	327	2.91E-07	10.55	4.64E-07
141	TFmatrixID_0711	ERF	161720	0	2508171	231	5.39E-07	Inf	8.53E-07
142	TFmatrixID_0693	ERF	161720	2	2508171	309	8.06E-07	9.96	1.27E-06
143	TFmatrixID_1197	MYB	161720	0	2508171	206	2.57E-06	Inf	4.01E-06
144	TFmatrixID_1286	NAC	161720	0	2508171	199	3.98E-06	Inf	6.17E-06
145	TFmatrixID_1250	MYB_related	161720	0	2508171	172	2.15E-05	Inf	3.31E-05
146	TFmatrixID_1291	NAC	161720	2	2508171	244	3.00E-05	7.87	4.59E-05
147	TFmatrixID_1050	GeBP	161720	0	2508171	162	4.02E-05	Inf	6.10E-05
148	TFmatrixID_1504	ZF-HD	161720	1	2508171	192	7.79E-05	12.38	1.17E-04
149	TFmatrixID_1200	MYB	161720	0	2508171	147	1.03E-04	Inf	1.54E-04
150	TFmatrixID_0963	C2H2	161720	0	2508171	115	7.58E-04	Inf	1.13E-03
151	TFmatrixID_0948	C2H2	161720	0	2508171	114	8.06E-04	Inf	1.19E-03
152	TFmatrixID_0902	Dof	161720	7	2508171	289	2.43E-03	2.66	3.56E-03
153	TFmatrixID_0710	ERF	161720	0	2508171	81	6.34E-03	Inf	9.24E-03

Note: P-values were adjusted with Benjamini-Hochberg method. ERF motifs were marked in red font.

Table R7. Enriched motifs on ACRs near C₄ genes based on updated peaks

Index	Motif	TF family	#Total occurrence in background	# Motif in background	# Total motif C ₄ gene	# Motif in C ₄	Pvalue (Fisher.test)	Odds Ratio
1	TFmatrixID_0771	ERF	1471751	16826	1858	49	1.6E-07	2.31
2	TFmatrixID_0748	ERF	1471751	36710	1858	76	4.1E-05	1.64
3	TFmatrixID_0782	ERF	1471751	24802	1858	54	1.1E-04	1.72
4	TFmatrixID_0777	ERF	1471751	34203	1858	67	4.7E-04	1.55
5	TFmatrixID_0664	ERF	1471751	51305	1858	92	8.2E-04	1.42
6	TFmatrixID_0499	MIKC_MADS	1471751	3557	1858	11	3.4E-03	2.45
7	TFmatrixID_0719	ERF	1471751	40625	1858	72	3.6E-03	1.40
8	TFmatrixID_0724	ERF	1471751	15570	1858	32	4.5E-03	1.63
9	TFmatrixID_0772	ERF	1471751	27689	1858	50	8.4E-03	1.43
10	TFmatrixID_1259	MYB_related	1471751	448	1858	3	1.0E-02	5.31
11	TFmatrixID_0023	ERF	1471751	57091	1858	91	1.8E-02	1.26
12	TFmatrixID_1378	B3	1471751	979	1858	4	1.9E-02	3.24
13	TFmatrixID_1238	MYB_related	1471751	573	1858	3	1.9E-02	4.15
14	TFmatrixID_1270	MYB_related	1471751	18663	1858	34	1.9E-02	1.44
15	TFmatrixID_0892	Dof	1471751	9574	1858	19	2.9E-02	1.57
16	TFmatrixID_0798	BBR-BPC	1471751	1172	1858	4	3.2E-02	2.70
17	TFmatrixID_1004	CPP	1471751	55	1858	1	3.4E-02	14.40
18	TFmatrixID_0973	C2H2	1471751	22956	1858	39	3.7E-02	1.35
19	TFmatrixID_0055	ERF	1471751	421	1858	2	5.0E-02	3.76

Note: 117 motifs associated with C₄ genes were predicted, 19 of them show P-value less than 0.05. ERF motifs were marked in red font.

Table R8. Enriched motifs on ACRs related to C₄ genes from original peaks

Index	Motif	TF family	#Total motif in background	# Motif in background	# Total motif C ₄ gene	# Motif in C ₄	Pvalue (Fisher.test)	Odds Ratio
1	TFmatrixID_0796	BBR-BPC	4160630	32286	1379	35	2.09E-09	3.27
2	TFmatrixID_0797	BBR-BPC	4160630	52522	1379	39	5.14E-06	2.24
3	TFmatrixID_1122	LBD	4160630	67458	1379	42	9.43E-05	1.88
4	TFmatrixID_0795	BBR-BPC	4160630	67079	1379	37	1.93E-03	1.66
5	TFmatrixID_0742	ERF	4160630	85386	1379	43	5.16E-03	1.52
6	TFmatrixID_1238	MYB_related	4160630	512	1379	2	6.48E-03	11.79
7	TFmatrixID_1377	B3	4160630	37978	1379	22	7.56E-03	1.75
8	TFmatrixID_0761	ERF	4160630	31334	1379	19	8.99E-03	1.83
9	TFmatrixID_0685	ERF	4160630	17411	1379	12	9.35E-03	2.08
10	TFmatrixID_1126	LBD	4160630	25532	1379	15	1.81E-02	1.77
11	TFmatrixID_0775	ERF	4160630	51236	1379	26	1.87E-02	1.53
12	TFmatrixID_0719	ERF	4160630	37048	1379	20	2.14E-02	1.63
13	TFmatrixID_0717	ERF	4160630	29387	1379	16	2.65E-02	1.64
14	TFmatrixID_0765	ERF	4160630	25106	1379	14	2.78E-02	1.68
15	TFmatrixID_0957	C2H2	4160630	4367	1379	4	2.98E-02	2.76

Note: 106 motifs associated with C₄ genes were predicted, 15 of them show P-value less than 0.05 (orange background), ERF motifs were marked in red font.

Table R9. Comparison of C₄-gene associated peaks in updated peaks and original peaks

Chr. peak	Peak start	Peak end	Peak length	Chr. gene	Gene start	Gene end	GeneID	Strand	Name
Updated peaks									
chr18	94257279	94257717	438	chr18	94259412	94274641	Ftri18G25078	+	CA1.2
chr3	12813853	12814715	862	chr3	12804741	12811443	Ftri3G16655	-	PEPC1.1
chr14	6754533	6755514	981	chr14	6744728	6753862	Ftri14G29727	-	PPDK
chr8	6716981	6718451	1470	chr8	6711821	6716590	Ftri8G07792	+	AlaAT1
chr3	2072080	2072629	549	chr3	2063337	2072319	Ftri3G17815	-	NHD1
chr12	120751623	120752094	471	chr12	120751662	120754595	Ftri12G28389	+	OMT
Original peaks									
chr18	94257425	94257722	297	chr18	94259412	94274641	Ftri18G25078	+	CA1.2
chr3	12814288	12814688	400	chr3	12804741	12811443	Ftri3G16655	-	PEPC1.1
chr14	6754636	6755301	665	chr14	6744728	6753862	Ftri14G29727	-	PPDK
chr8	6716977	6717999	1022	chr8	6711821	6716590	Ftri8G07792	+	AlaAT1
chr3	2072086	2072582	496	chr3	2063337	2072319	Ftri3G17815	-	NHD1
chr12	120751783	120751906	123	chr12	120751662	120754595	Ftri12G28389	+	OMT
chr3	11542191	11542355	164	chr3	11535740	11542489	Ftri3G30452	+	PEPC1.3
chr15	713158	714274	1116	chr15	712896	715061	Ftri15G31095	+	PPDK-RP1
chr3	2062905	2063131	226	chr3	2063337	2072319	Ftri3G17815	-	NHD1
chr3	2064585	2064909	324	chr3	2063337	2072319	Ftri3G17815	-	NHD1
chr7	17893573	17893693	120	chr7	17889689	17892775	Ftri7G08542	+	DCT
chr18	94219992	94220128	136	chr18	94216923	94219051	Ftri18G21063	-	CA1.1

Related reference:

[1] BiasAway: command-line and web server to generate nucleotide composition-matched DNA background sequences, *Bioinformatics*, 2021 [PMID: 33135764]

Q2: Additionally, I could not find a description of how the enrichment scores were calculated in extended data fig 8 in the revised manuscript. What were the background sequences used to determine enrichment? ERF motifs are pretty degenerate (GGC repeats) and frequent in the genome by chance. Please include this information in the methods section. In general, background sequences should be matched by GC content, genome distribution (i.e. same fraction of intergenic, promoter, genic overlap as ACRs), and by length (same length distribution as ACRs).

Response: We thank the reviewer for highlighting this important detail regarding the calculation of enrichment scores in Extended Data Figure 8. We apologize for the oversight in omitting this information in the initial submission. Since we only succeed in the ATAC-Seq experiment for a C₄ species (Ftri), the motif enrichment analysis was conducted based on putative promoter region of C₄ genes in each of the five *Flaveria* species. To determine the enriched motifs within the promoters of C₄ genes, we utilized the HOMER package [1]. Specifically, for each species, the promoter sequences of C₄ genes were taken as the input. We used the promoter sequences of non-C₄ genes as the background to match the sequence genomic distribution. We agree with the reviewer that background sequence is of importance for motif finding. We chose HOMER package for motif enrichment finding analysis because this algorithm conducted two types of normalization to account

for the sequence composition bias between the input and background sequences. Regarding GC composition bias, HOMER package automatically selects background regions from the promoter sequences of non-C₄ genes that match the GC-content distribution of the promoter sequences of C₄ genes (in 5% increments) according to the Manu of HOMER package (<http://homer.ucsd.edu/homer/ngs/peakMotifs.html>). Specifically, if the promoter sequences of C₄ genes (Input) are extremely GC-rich, HOMER will select random regions from GC-rich regions of the promoter sequences of non-C₄ genes (background) as a control. In addition to accounting for GC-content bias, HOMER package also applies “autonormalization of sequence bias” to remove bias introduced by lower-order oligo sequences associated with the promoter sequences of C₄ genes. HOMER package works by assuming the promoter sequences of C₄ genes (input) and the promoter sequences of non-C₄ genes (background) should not have an imbalance in 1-mers, 2-mers, 3-mers, etc, and after calculating the imbalances for each oligo, it adjusts the weights of each background sequence by a small amount to help normalize any imbalance. The above analysis procedure ensured that the motif enrichment analysis accounted for potential biases in sequence composition and genomic distribution between C₄ and non-C₄ gene promoters.

In response to the reviewer’s comments, we also compared sequence composition of promoter between C₄ genes and non-C₄ genes in each species. This comparison included the mononucleotide, dinucleotide GC, and trinucleotide GGC frequencies. The result showed no significant sequence composition differences for all mononucleotides, dinucleotide GC and trinucleotide GGC except that the GC and GGC content was significantly lower in C₄ gene promoters compared to non-C₄ gene promoters in a C₃-C₄ species *Fram* (Wilcoxon test, adjusted P < 0.05) (**Table R10**). It should be noted that the C₄ species (*Ftri*) showed no significant sequence composition differences for all mononucleotide, dinucleotide GC, and trinucleotide GGC frequencies between C₄ gene promoters and non-C₄ gene promoters. These results suggest that differences in sequence composition would have minimal impact on motif enrichment analysis. Any potential biases could be further corrected using the sequence composition bias normalization algorithm implemented in the HOMER package.

In the revised manuscript, we have included the description of the enrichment score calculation at lines 852-870 on Page 37, which was also shown as follows:

“We employed the HOMER package [1] to identify enriched motifs within the promoters (3 kb upstream of the start codons) of C₄ genes and their orthologous counterparts in each *Flaveria* species. For each species, the promoter sequences (3 kb upstream of the start codon) of non-C₄ genes were used as the background to account for potential genomic distribution bias. Regarding sequence composition bias, the HOMER package automatically selects background regions from the promoter sequences of non-C₄ genes that match the GC-content distribution of the promoter sequences of C₄

genes (in 5% increments), as detailed in the HOMER manual (<http://homer.ucsd.edu/homer/ngs/peakMotifs.html>). Specifically, if the promoter sequences of C₄ genes (Input) are highly GC-rich, HOMER selects random regions from GC-rich regions of the promoter sequences of non-C₄ genes (background) as a control. In addition to accounting for GC-content bias, HOMER package also applies “autonormalization of sequence bias” to eliminate bias introduced by lower-order oligo sequences associated with the promoter sequences of C₄ genes. The HOMER package operates under the assumption that the promoter sequences of C₄ genes (input) and non-C₄ genes (background) should not exhibit imbalances in 1-mers, 2-mers, 3-mers, etc. After calculating these imbalances for each oligonucleotide, HOMER adjusts the weights of background sequences slightly to normalize the imbalances. This analytical procedure ensured that the enrichment analysis accounted for potential biases in sequence composition and genomic distribution between the promoters of C₄ and non-C₄ genes.”

Related reference:

[1] Simple combinations of lineage-determining transcription factors prime *cis*-regulatory elements required for macrophage and B cell identities, *Mol Cell*, 2010 [PMID: 20513432]

Table R10. Comparison of promoter sequence compositions between C₄ gene promoters and non-C₄ gene promoters.

Type	Frob (C ₃)		Pvalue	Padj. ("BH")	Fson (C ₃ -C ₄)		Pvalue	Padj. ("BH")	Flin (C ₃ -C ₄)		Pvalue	Padj. ("BH")	Fram (C ₃ -C ₄)		Pvalue	Padj. ("BH")	Ftri (C ₄)		Pvalue	Padj. ("BH")
	C ₄	non-C ₄			C ₄	non-C ₄			C ₄	non-C ₄			C ₄	non-C ₄			C ₄	non-C ₄		
A	0.36	0.34	0.051	0.107	0.34	0.34	0.265	0.499	0.33	0.33	0.736	0.935	0.34	0.32	0.085	0.085	0.34	0.34	0.404	0.658
T	0.35	0.34	0.388	0.466	0.34	0.34	0.97	0.97	0.33	0.33	0.612	0.935	0.33	0.31	0.042	0.06	0.34	0.34	0.548	0.658
G	0.15	0.16	0.183	0.275	0.16	0.16	0.639	0.766	0.17	0.17	0.805	0.935	0.16	0.19	0.036	0.06	0.15	0.16	0.482	0.658
C	0.15	0.16	0.053	0.107	0.16	0.16	0.333	0.499	0.17	0.17	0.762	0.935	0.17	0.18	0.05	0.06	0.16	0.16	0.742	0.742
GC	0.02	0.02	0.047	0.107	0.02	0.02	0.037	0.224	0.03	0.03	0.935	0.935	0.02	0.03	0.003	0.019	0.02	0.02	0.271	0.658
GGC	0.004	0.005	0.633	0.633	0.004	0.005	0.158	0.473	0.006	0.006	0.739	0.935	0.004	0.006	0.007	0.02	0.004	0.005	0.295	0.658

Note: statistical significance was assessed using the Wilcoxon rank-sum test, and p-values were adjusted for multiple comparisons using the Benjamini-Hochberg (BH) method. Significant differences are highlighted in yellow.

Q3: Related to the above comments. I could not understand the statistical rationale for enrichment tests in the authors response (sounds like a Monte Carlo permutation test, but the way the description is worded lacks clarity and conciseness). The authors move between descriptions of motifs and ACRs as the focus of the test? It doesn't make any sense. The choice of the background is not justified and was not what I asked for (the specific enrichment test requested was ACRs near C₄ genes versus the total set of ACRs). Something like Fisher's exact test or a Chi-square test would be sufficient for enrichment tests, provided the background/control regions are well reasoned. Also,

how is FDR calculated? Which specific method? I did not see this information in the response or in the methods of the revised manuscript. The apparent lack of rigor is a concern. Although, this could be alleviated by more comprehensive reporting in the methods.

Response: We appreciate reviewer for pointing out this issue. we apologize for omitting detailed method for finding enriched motifs within the ATAC-seq peak regions of C₄ genes. We applied the following steps to obtain the enriched motifs. Firstly, FIMO of MEME package (v5.0.2) was used to identify the occurrences of known motifs of plants under a q-value threshold of 0.05 within the total set of ACRs of ATAC-seq data. The known plant motifs were annotated in Plantpan 3.0. This analysis resulted in 1,471,751 occurrences of 277 motifs. Among these, 1,858 occurrences of 117 motifs were from the ACRs near C₄ genes (including the regions 3 kb upstream start codons, 3kb downstream of stop codons, and the gene bodies). To further determine which motif was associated with the C₄ genes more than expected by chance, we performed a Monte Carlo permutation test. Specifically, for each of these 117 motifs, we compared its observed occurrence and the distribution of the expected occurrences that were estimated based on 1,000 permutations. In each permutation, 1,858 motif occurrences were randomly selected from the total 1,471,751 occurrences, and the expected occurrence of each motif was recorded. After 1,000 permutations, p-value for each motif was estimated as ratio between the number of permutations where the occurrence of motifs exceeding the observed occurrence and the number of total permutations (1,000). The Benjamini and Hochberg correction was further applied to obtain the FDR value of each motif.

We agree with the reviewer that instead of using permutation test, Fisher's exact test is also sufficient for the enrichment tests. Accordingly, using 1,471,751 occurrences of 277 motifs from the total set of ACRs as background, we applied Fisher's exact test to calculate enrichment for each motif within ACRs near C₄ genes. The motif enrichment results were largely consistent between these two methods. The union of enriched motifs associated with C₄ genes identified using these two methods based on updated peaks are presented in **Table R11**. The table was also included to the Table S11 in Supplementary Note 13 in the updated Supplementary note file.

In the revised manuscript, we revised the method for identifying enriched motif associated with C₄ genes by including both permutation-based method and Fisher's exact test-based method (Pages 36-37, lines 829-851), which was also shown as follows:

“Based on accessible chromatin regions (ACRs) from ATAC-seq data of C₄ species (Ftri), we employed both a permutation-based method and a Fisher's exact test-based method to predict the enriched CREs associated with C₄ genes (including the regions 3 kb upstream of start codons, 3kb downstream of stop codons, and the gene bodies). FIMO of MEME suite (v5.0.2) 114 was used to identify the occurrences of known CREs of plants within the entire set of ACRs, applying a q-value threshold of 0.05. The CRE annotations were sourced from PlantPAN 3.0 ⁹⁵ (<https://plantpan.itps.ncku.edu.tw/plantpan3/download/home.php>). This analysis identified 1,471,751

occurrences of 277 distinct CREs, with 1,858 occurrences of 117 CREs were associated with C₄ genes. To assess whether specific CREs were overrepresented near C₄ genes beyond random chance, we conducted a Monte Carlo permutation test. For each of the 117 CREs, observed occurrences were compared against a distribution of expected occurrences estimated from 1,000 permutations. In each permutation, 1,858 CRE occurrences were randomly selected from the total pool, and the frequency of each CRE was recorded. Following the completion of all permutations, the p-value for each CRE was calculated as the proportion of permutations where CRE occurrences surpassed the observed value. To control for multiple testing, we applied the Benjamini-Hochberg procedure to adjust for the false discovery rate (FDR). For the Fisher’s exact test-based method, we evaluated the enrichment of each CRE associated with C₄ genes against the background of 1,471,751 total CRE occurrences. The CRE enrichment results were largely consistent between these two methods (Supplementary Note 13).

Furthermore, to predict enriched CREs in ACRs in various genomic contexts, including within gene bodies, upstream and downstream of genes, as well as those associated with photosynthetic and photorespiratory genes, we employed the Monte Carlo permutation test as described above.” (Pages 36-37, lines 829-851)

Table R11. The union of enriched motifs on ACRS near C₄ genes using permutation-based method or Fisher’s exact test-based method.

Motif	TF family	#Total motif in background	# Motif in background	# Total motif C ₄ gene	Motif in C ₄	FDR (1000 times of Permutation)	Pvalue (Fisher’s exact test)	Odds Ratio
TFmatrixID_0771	ERF	1471751	16826	1858	49	0	1.6E-07	2.31
TFmatrixID_0748	ERF	1471751	36710	1858	76	0	4.1E-05	1.64
TFmatrixID_0777	ERF	1471751	34203	1858	67	0.001	4.7E-04	1.55
TFmatrixID_0499	MIKC_MADS	1471751	3557	1858	11	0.001	3.4E-03	2.45
TFmatrixID_0782	ERF	1471751	24802	1858	54	0.001	1.1E-04	1.72
TFmatrixID_0664	ERF	1471751	51305	1858	92	0.001	8.2E-04	1.42
TFmatrixID_0719	ERF	1471751	40625	1858	72	0.007	3.6E-03	1.40
TFmatrixID_0772	ERF	1471751	27689	1858	50	0.007	8.4E-03	1.43
TFmatrixID_0724	ERF	1471751	15570	1858	32	0.008	4.5E-03	1.63
TFmatrixID_1270	MYB_related	1471751	18663	1858	34	0.022	1.9E-02	1.44
TFmatrixID_1259	MYB_related	1471751	448	1858	3	0.029	1.0E-02	5.31
TFmatrixID_0798	BBR-BPC	1471751	1172	1858	4	0.029	3.2E-02	2.70
TFmatrixID_0973	C2H2	1471751	22956	1858	39	0.034	3.7E-02	1.35
TFmatrixID_0892	Dof	1471751	9574	1858	19	0.042	2.9E-02	1.57
TFmatrixID_0023	ERF	1471751	57091	1858	91	0.043	1.8E-02	1.26
TFmatrixID_1238	MYB_related	1471751	573	1858	3	0.047	1.9E-02	4.15
TFmatrixID_1378	B3	1471751	979	1858	4	0.048	1.9E-02	3.24
TFmatrixID_1214	MYB	1471751	28474	1858	41	0.048	2.0E-01	1.14
TFmatrixID_0055	ERF	1471751	421	1858	2	0.048	5.0E-02	3.76
TFmatrixID_1370	Nin-like	1471751	5510	1858	10	0.049	1.2E-01	1.44

TFmatrixID_1126	LBD	1471751	3265	1858	7	0.049	7. 0E-02	1.70
TFmatrixID_0743	ERF	1471751	47391	1858	72	0.049	6. 6E-02	1.20
TFmatrixID_1004	CPP	1471751	55	1858	1	0.062	3. 4E-02	14.40

Note: FDR were adjusted with Benjamini-Hochberg method.

Q4: I wish to note that the addition of the EMS and transient transcription assays is a strength. However, I do wonder why these results weren't included in the original submission, given the quick turnaround time of the revised manuscript (seems unlikely that the review was the prompt for these new experiments). I hope that the computational analyses were not cherry-picked to support the experimental results. To enable reanalysis of the ATAC-seq data by other laboratories, please upload the raw fastq files from the ATAC-seq replicates to a public repository (NCBI GEO), similar to the other data sets.

Response: We appreciate reviewer's comments on the addition of EMSA and transient transcriptional assay. We also thank the reviewer for recognizing the value that these experimental results bring to our study. Initially, the EMSA experimental results were included in a version submitted to another journal. Later, we realized that the current manuscript would be more focused on genome assembly and genomic analysis, and therefore did not include experimental validation results in the original submission to *Nature Communications*. However, following the suggestion of Reviewer #4, we reintroduced the EMSA results and conducted new transient expression experiments to further validate the activation of ERF TFs in regulating the expression of C₄ genes during the last revision.

We assure the reviewer that our computational analyses were not selectively designed to support the experimental results. Please note that our observations regarding the regulatory function of ERF TFs on C₄ genes were based on gene regulatory network analysis, promoter sequence motif analysis, and ATAC-seq-based analysis. We acknowledge that the quality of ATAC-seq data may not be as high as that obtained from model species such as *Arabidopsis* and *maize*. However, this represents the best ATAC-seq data we could obtain at this time. We made significant efforts to conduct ATAC-seq experiments in other *Flaveria* species but were unable to obtain data of comparable quality to that of the C₄ species *Ftri*. The ATAC-seq data for other *Flaveria* species can be accessed via the following link: <https://pan.baidu.com/s/1f42EcAthuFHgr5-FF72rwQ?pwd=1234> (password: 1234). We are continuing to refine our experimental protocols to obtain high-quality ATAC-seq data for *Flaveria* species. However, at present, the ATAC-seq data for the C₄ species *Ftri* is the only dataset available. We also mentioned the issue of ATAC-seq data as a limitation in the Discussion (Page 24, lines 518-529), which is shown as following:

“ATAC-seq is an important genomic approach for facilitating the genome-wide identification of *cis*-regulatory elements⁶²⁻⁶⁴. However, obtaining high-quality ATAC-seq data remains challenging, especially for non-model species, including those in *Flaveria* genus. In this study, we obtained ATAC-seq data only from C₄ species (*Ftri*). Although considerable effort has been devoted to ATAC-seq experiments in other *Flaveria* species, we were unable to obtain ATAC-seq data of

comparable quality to that of the C₄ species (Ftri). Based on ATAC-seq data from C₄ species (Ftri), we provided evidence that ERF CREs were enriched in the open chromatin regions near C₄ genes (Fig. 5b). Importantly, the electrophoretic mobility-shift assay (EMSA) and transient transcription assay further verified the regulation of ERF TFs on the expression of C₄ genes. Nevertheless, high-quality ATAC-seq data from species of *Flaveria* genus other than C₄ species (Ftri) are critical for further deepening our understanding of the regulatory and evolutionary mechanisms underlying the formation of C₄-specific photosynthesis, which requires further exploration.”

The raw data of ATAC-seq have been uploaded in CNGB Nucleotide Sequence Archive (CNSA) with accession numbers CNR0676372 and CNR0676373, which can be accessed via the following links:

<https://db.cngb.org/search/run/CNR0676372>

<https://db.cngb.org/search/run/CNR0676373>

Minor comments:

1. The manuscript is still littered with grammatical and nonsensical errors. For example, line 377-380 “... from the two replicates Apply Irreducible Discovery Rate...” should probably read “... from the two replicates after applying Irreproducible Discovery Rate (IDR) analysis...”.

Response: We appreciate reviewer for pointing out this typo. The sentence was corrected in the revised manuscript (Page 18, line 389). Furthermore, we have conducted a thorough review of the entire manuscript to correct any remaining typos and grammatical errors. We have made every effort to ensure that the revised manuscript is clear, accurate, and free from such issues.

2. For the correlation analysis between replicates (using the IDR peaks threshold at 0.05), of course the peaks are highly correlated, this is the point of IDR. Please report the correlation between replicates using the union of the raw peaks identified in rep 1 and rep 2.

Response: We appreciate the reviewer’s comments regarding the correlation analysis between replicates. Accordingly, we computed the correlation of union peaks from two replicates, which resulted in 204,265 peaks. The correlation coefficient for these union peaks was 0.90 (**Figure R2**). We also included this result in the revised manuscript as Supplementary Note 13, Fig. S26.

Figure R2. Replicate consistency analysis of ATAC-seq data in Ftri

Plots illustrate the correlation of mapped read counts from union peak regions across two biological replicates of Ftri. The regions with counts within 2000 are highlighted for clarity.

Reviewer #4:

Q1: The authors have addressed these questions and provided more experimental information for the conclusions. Lines 184-187, The C₄ species showed significantly higher enzyme activity and light responsiveness, which needs more explanation and discussion.

Response: We are deeply grateful to the reviewer for his/her considerate and positive review, along with insightful questions and suggestions for these two rounds of revisions. These inputs are immensely valuable to us in enhancing our manuscript. The observation of significantly higher enzyme activity and light responsiveness in C₄ species aligns with previous studies [1, 2, 3]. We have cited relevant publications in the Result (Page 9, lines 184-189 for the observation of light responsiveness, and Page 9, lines 195-198 for the observation of enzyme activity) of the revised manuscript, which was also shown as follows:

“These findings align with previous reports [1, 2], indicating that C₄ genes have evolved to become light-responsive over time. Given that orthologs of these C₄ genes play roles in primary metabolism within C₃ species³⁶, the acquisition of light responsiveness during the evolution of C₄ photosynthesis enables these genes to better synchronize their activities with those of other photosynthetic genes, which predominantly exhibit light responsiveness.” (Page 9, lines 184-189)

“Specifically, the C₄ species displayed approximately 10-fold higher enzyme activities for PEPC and NADP-MDH compared to the C₃ species. For NADP-ME, the increase was even more pronounced, aligning with recent observations of enzyme activity in *Flaveria* species [3]” (Page 9, lines 195-198)

Related references:

[1] Ancestral light and chloroplast regulation form the foundations for C₄ gene expression, *Nature Plants*, 2016 [PMID: 27748771]

[2] What Matters for C₄ Transporters: Evolutionary Changes of Phosphoenolpyruvate Transporter for C₄ Photosynthesis, *Front Plant Sci*, 2020 [PMID: 32695130]

[3] The Evolution of C₄ Photosynthesis in *Flaveria* (Asteraceae): Insights from the *Flaveria linearis* Complex, *Plant Physiol*, 2023 [PMID: 36200882]

Some minor points need to be modified in the methods.

Q2: Line 810 and 811: “0.1 IPTG” should be “0.1 mM IPTG”? “ortholog in is Arabidopsis” should be “ortholog in Arabidopsis is” ?

Response: We appreciate reviewer pointing out this error. The sentence was corrected in the revised manuscript.

Q3: Some spaces need to be added in the text for lines 711-712, 811.

Response: We appreciate reviewer pointing out this error and have corrected accordingly.

Reviewer #5:

I thank the authors for addressing all my comments. I appreciate the inclusion of ribosome footprinting data in the study of translational efficiency. Questions/concerns regarding the orthologous protein relationships and spectral library searches are now clarified, also the proteomics experiment is now properly introduced. Taken together, the methodology and drawn conclusions from these data seem appropriate to me. Lyu and co-authors addressed all my comments. I appreciate the extensive effort to incorporate the iBAQ quantification and especially the inclusion of ribosome footprinting data to strengthen the claims on translational efficiency. Also the proteomics analyses were now clarified. All together, this present an extensive work, and I believe this to be of general interest to the plant genomics community.

Response: We are deeply grateful to the reviewer for his/her considerate and positive review, along with insightful questions and suggestions for these two rounds of revisions. These inputs are immensely valuable to us in enhancing our manuscript. We hope the high-quality chromosome-level reference genomes and the comprehensive genomic data provided in this study would facilitate further researches regarding evolution and regulatory mechanisms underlying C₄ photosynthesis.